# A Statistical Theory of Contrastive Learning via Approximate Sufficient Statistics

**Licong Lin**
UC Berkeley
liconglin@berkeley.edu

**Song Mei**
UC Berkeley
songmei@berkeley.edu

## Abstract

Contrastive learning—a modern approach to extract useful representations from unlabeled data by training models to distinguish similar samples from dissimilar ones—has driven significant progress in foundation models. In this work, we develop a new theoretical framework for analyzing data augmentation-based contrastive learning, with a focus on SimCLR as a representative example. Our approach is based on the concept of *approximate sufficient statistics*, which we extend beyond its original definition in Oko et al. [28] for contrastive language-image pretraining (CLIP) using KL-divergence. We generalize it to equivalent forms and general f-divergences, and show that minimizing SimCLR and other contrastive losses yields encoders that are approximately sufficient. Furthermore, we demonstrate that these near-sufficient encoders can be effectively adapted to downstream regression and classification tasks, with performance depending on their sufficiency and the error induced by data augmentation in contrastive learning. Concrete examples in linear regression and topic classification are provided to illustrate the broad applicability of our results.

## 1 Introduction

Leveraging massive unlabeled data to learn useful representations has played a central role in recent advances in foundation models. A prominent approach of this kind is contrastive learning, which has driven significant progress in visual representation learning [5, 14], large-scale speech processing [3], and multimodal AI [31, 21].

In short, contrastive learning finds useful representations of the data by maximizing similarity between paired samples while minimizing it for non-paired samples. Consider SimCLR [5] for visual representation learning as an illustrative example. Given a dataset of images $x \in \mathcal{X}$, SimCLR generates two augmented views $(z^{(1)}, z^{(2)}) \in \mathcal{X} \times \mathcal{X}$ for each image $x$ using random transformations (i.e., data augmentations) such as random cropping, random color distortions, and random Gaussian blur, etc. It then trains an encoder $f$ that aligns the paired views and separates the non-paired views through minimizing the loss in Eq. (2). The learned representation $f(x)$ (or $f(z^{(1)})$) can then be adapted to downstream tasks with few labeled samples and minimal fine-tuning.

Despite its remarkable empirical performance, the theoretical aspects of contrastive learning remain an active area of study [32, 28]. In this work, we present a theoretical analysis of data augmentation-based contrastive learning, with a specific focus on the SimCLR framework [5] as a representative example. Notably, recent work by Oko et al. [28] has introduced new theoretical insights into contrastive language-image pretraining (CLIP). They first introduced the concept of approximate sufficient statistics, showing that the image and text encoders obtained from the empirical risk minimizer of CLIP are approximately sufficient. Additionally, under the joint graphical hierarchical model (JGHM) assumption for image and text data, they demonstrated that such encoders can be efficiently adapted to various downstream multimodal tasks.

39th Conference on Neural Information Processing Systems (NeurIPS 2025).

Our work complements and extends the work by Oko et al. [28] in two key ways.

(1) We extend the concept of approximate sufficient statistics, which was originally defined for CLIP in a specific form based on KL-divergence, to three equivalent forms and general f-divergences. Based on the equivalent forms of the definition, we establish that minimizing the contrastive loss (e.g., the InfoNCE loss [29]) is essentially finding approximate sufficient statistics that are adaptable to downstream tasks.

(2) We focus on data augmentation-based contrastive learning following the SimCLR framework. In contrast to CLIP, the random transformations in SimCLR introduce additional challenges for theoretical analysis. We show that the downstream performance of the learned encoder depends on its sufficiency and the error induced by the random transformations. Furthermore, motivated by the generalized definition of approximate sufficient statistics, we theoretically demonstrate that encoders trained using alternative contrastive losses can achieve similar downstream performance to those trained using standard SimCLR.

The remainder of this work is organized as follows. In Section 2, we introduce the concept of approximate sufficient statistics. Sections 3.1–3.2 present the setup of data augmentation-based contrastive learning and analyze the downstream performance of the SimCLR-trained encoder. In Section 3.3, we extend our analysis to general f-contrastive losses. Examples in linear regression and topic classification are presented in Section 4. We also conduct synthetic experiments to compare contrastive learning losses in Section 5. Discussion of related works is deferred to Appendix A.

## 2 Approximate sufficient statistics

Before diving into the analysis of contrastive learning, we first introduce the concept of approximate sufficient statistics, which provides a novel viewpoint for characterizing the quality of encoders $f$ used in contrastive learning. Let $f : \mathbb{R}_+ \mapsto \mathbb{R}$ be a convex function such that $f(1) = 0$. For random variables $(X, Y)$ on $\mathcal{X} \times \mathcal{Y}$ with joint density $\mathbb{P}(x, y)$ with respect to some measure $\boldsymbol{\mu}$ [1], we define the f-mutual information (f-MI) as

$$I_{\mathrm{f}}(X, Y) = \int \mathrm{f}\Big(\frac{\mathbb{P}(x, y)}{\mathbb{P}(x)\mathbb{P}(y)}\Big)\mathbb{P}(x)\mathbb{P}(y)d\boldsymbol{\mu}.$$

Note that the f-MI is essentially the f-divergence between the joint distribution and the product of marginal distributions. It is non-negative and symmetric in $X$ and $Y$. Moreover, provided that f is strictly convex, $I_{\mathrm{f}}(X, Y) = 0$ if and only if $X$ and $Y$ are independent. Let $(X, Y)$ be random variables that have the joint density $\mathbb{P}(X, Y)$ ($Y$ could be thought as the parameter $\theta$ in Bayesian statistics). For any statistic $T : \mathcal{X} \mapsto T(\mathcal{X})$, to characterize the information loss of using $T(X)$ instead of $X$ for predicting $Y$, we introduce the following definition of the sufficiency of $T(X)$.

**Definition 1** (Approximate sufficiency). *Let $T : \mathcal{X} \to T(\mathcal{X})$ be a mapping (i.e., a statistic). We define three forms of the sufficiency of $T$, which will be shown to be equivalent:*

- **Information Loss Sufficiency (ILS):** *The information loss sufficiency of $T$ is defined as*

$$\mathrm{Suff}_{\mathrm{il,f}}(T) = I_{\mathrm{f}}(X, Y) - I_{\mathrm{f}}(T(X), Y).$$

- **Variational Form Sufficiency (VFS):** *The variational form sufficiency of $T$ is given by*

$$\mathrm{Suff}_{\mathrm{vf,f}}(T) = \inf_{\mathsf{S}:T(\mathcal{X})\times\mathcal{Y}\mapsto\mathbb{R}} R_{\mathrm{f}}(\mathsf{S} \circ T) - \inf_{\mathsf{S}:\mathcal{X}\times\mathcal{Y}\mapsto\mathbb{R}} R_{\mathrm{f}}(\mathsf{S}),$$

*where $\mathsf{S} \circ T(x, y) := \mathsf{S}(T(x), y)$, and the f-contrastive loss*

$$R_{\mathrm{f}}(\mathsf{S}) := \mathbb{E}_{\mathbb{P}(x,y)}[-\mathsf{S}(x, y)] + \inf_{\mathsf{S}_x:\mathcal{X}\mapsto\mathbb{R}} \mathbb{E}_{\mathbb{P}(x)\mathbb{P}(y)}[\mathrm{f}^*(\mathsf{S}(x, y) - \mathsf{S}_x(x)) + \mathsf{S}_x(x)], \quad (1)$$

*where $\mathrm{f}^*$ is the Fenchel-dual of f.*

---

[1] For example, $\boldsymbol{\mu}$ can be the Lebesgue measure on Euclidean spaces, or the counting measure on discrete spaces.

- **_Conditional Bregman Sufficiency (CBS):_** _The conditional Bregman sufficiency of $T$ is defined as_

$$\mathrm{Suff}_{\mathrm{cb,f}}(T) = \mathbb{E}_{\mathbb{P}(x) \times \mathbb{P}(y)} \Big[ B_{\mathrm{f}} \Big( \frac{\mathbb{P}(y|x)}{\mathbb{P}(y)}, \frac{\mathbb{P}(y|T(x))}{\mathbb{P}(y)} \Big) \Big],$$

_where $B_{\mathrm{f}}(a, b) := \mathrm{f}(a) - \mathrm{f}(b) - (a - b)\mathrm{f}'(b)$ is the Bregman divergence of $\mathrm{f}$._

_Indeed, these definitions will be shown to be equivalent (Lemma 1), i.e.,_

$$\mathrm{Suff}_{\mathrm{il,f}}(T) = \mathrm{Suff}_{\mathrm{vf,f}}(T) = \mathrm{Suff}_{\mathrm{cb,f}}(T) =: \mathrm{Suff}_{\mathrm{f}}(T).$$

_We say $T(X)$ is an $\varepsilon$-approximate sufficient statistic if $\mathrm{Suff}_{\mathrm{f}}(T) \leqslant \varepsilon$._

The Information Loss Sufficiency (ILS) is closely linked to the InfoMax principle [22, 15], which finds a statistic $T$ that maximizes mutual information $I(T(X), Y)$ under certain constraints. The equivalence between ILS and CBS suggests that the loss in mutual information can be represented as a divergence between the conditional probabilities $\mathbb{P}(Y|X)$ and $\mathbb{P}(Y|T(X))$. This provides a concrete measure for interpreting the information loss.

In VFS, by definition, the excess risk $R_{\mathrm{f}}(\mathsf{S} \circ T) - \inf_{\widetilde{\mathsf{S}}} R_{\mathrm{f}}(\widetilde{\mathsf{S}})$ serves as an upper bound on the sufficiency $\mathrm{Suff}_{\mathrm{f}}(T)$, and they are nearly equal when $\mathsf{S}$ is obtained by minimizing $R_{\mathrm{f}}(\mathsf{S} \circ T)$ over a sufficiently rich space $\mathcal{S}$. Consequently, VFS provides a loss minimization framework for finding $T$ with low sufficiency by minimizing the $\mathrm{f}$-contrastive loss $R_{\mathrm{f}}(\mathsf{S})$ over $\mathsf{S}$ in some space $\mathcal{S}$ and extracting $T$ from $\mathsf{S}$. Moreover, an extension of approximate sufficiency to similarity scores $\mathsf{S}$ is introduced in Appendix B.3.

The concept of approximate sufficient statistics was first proposed in Oko et al. [28], but only in the CBS form for KL divergence (i.e., $\mathrm{f}(x) = x \log x$). In this work, we extend the definition to general $\mathrm{f}$-divergences and establish the equivalence among three forms of sufficiency. Notably, for $\mathrm{f}$ that is strictly convex, we have $\mathrm{Suff}_{\mathrm{f}}(T) = 0$ if and only if $Y \perp\!\!\!\perp X | T(X)$ from the CBS form, aligning with the classic definition of sufficient statistics (see e.g., [19]). We will mainly consider two special cases of $\mathrm{f}$: $\mathrm{f}(x) = x \log x$ (KL-divergence) and $\mathrm{f}(x) = (x - 1)^2/2$ ($\chi^2$-divergence), with the corresponding sufficiency denoted by $\mathrm{Suff}_{\mathrm{kl}}$ and $\mathrm{Suff}_{\chi^2}$. For more examples and properties regarding approximate sufficient statistics, we refer the readers to Appendix B.

In the context of data augmentation-based contrastive learning, we may choose $X$ and $Y$ as two augmented views of the sample, and $T$ as the encoder $f$. The sufficiency $\mathrm{Suff}_{\mathrm{f}}(f)$ then quantifies the loss of recovering augmented views from the encoder representation. We will show that the downstream performance of $f$ can be controlled by its sufficiency (in the CBS form) and the error induced by data augmentation. Specifically, for any downstream task, a small risk can be achieved using $f$ if it is near-sufficient and the random transformations in contrastive learning do not significantly change the downstream outcomes. As a preview of the results, we have

**Theorem (Informal).** *The risk on a downstream task using encoder $f$ (denoted by $\mathcal{R}(f)$) satisfies*

$$\mathcal{R}(f) \leqslant c \cdot \left( \sqrt{\mathrm{Suff}_{\mathrm{f}}(f)} + \epsilon_{\mathcal{G}} \right)$$

*for some constant $c > 0$, where $\mathrm{Suff}_{\mathrm{f}}(f)$ is the $\mathrm{f}$-sufficiency of $f$ and $\epsilon_{\mathcal{G}}$ denotes the error on the downstream task induced by data augmentation.*

Contrastive learning with general $\mathrm{f}$-divergence was also studied in [23, 48], but the loss functions considered in these works differ from the variational form in (1). In particular, while Lu et al. [23] considered a variational form similar to (1), they set $\mathsf{S}_x = 0$ instead of taking the infimum over $\mathsf{S}_x$.

## 3   Statistical properties of contrastive learning

In this section, we demonstrate that data augmentation-based contrastive learning can find near-sufficient encoders that are effectively adaptable to downstream tasks. We focus on the SimCLR framework in Section 3.1–3.2, and extend the results to general $\mathrm{f}$-contrastive losses in Section 3.3.

### 3.1 Setup and the ERM estimator

Let $\boldsymbol{x} \in \mathcal{X}$ be a random sample drawn from a distribution $\mathbb{P}_{\mathcal{X}}$ on $\mathcal{X}$. Consider a set of transformations $\mathcal{G}$ in which each transformation $g : \mathcal{X} \to \mathcal{X}$ maps $\mathcal{X}$ to itself.[2] Let $\mathbb{P}_{\mathcal{G}}$ denote a distribution over the transformations in $\mathcal{G}$. Given a sample $\boldsymbol{x}$ and two transformations $g^{(1)}, g^{(2)} \sim_{iid} \mathbb{P}_{\mathcal{G}}$, we generate two augmented views of $\boldsymbol{x}$, denoted as $\boldsymbol{z}^{(1)} = g^{(1)}(\boldsymbol{x})$ and $\boldsymbol{z}^{(2)} = g^{(2)}(\boldsymbol{x})$. The marginal distribution of $\boldsymbol{z}^{(1)}$ (or equivalently $\boldsymbol{z}^{(2)}$) is denoted by $\mathbb{P}_{\boldsymbol{z}}$. Often, we will omit the superscripts and let $\boldsymbol{z} = g(\boldsymbol{x})$ denote a single augmented view generated by a transformation $g \sim \mathbb{P}_{\mathcal{G}}$.

Throughout the remainder of this work, unless otherwise specified, we set $(X, Y) \overset{d}{=} (\boldsymbol{z}^{(1)}, \boldsymbol{z}^{(2)})$ in Definition 1, i.e., we define the sufficiency $\mathrm{Suff}_{\mathrm{f}}(T) = I_{\mathrm{f}}(\boldsymbol{z}^{(1)}, \boldsymbol{z}^{(2)}) - I_{\mathrm{f}}(T(\boldsymbol{z}^{(1)}), \boldsymbol{z}^{(2)})$. For simplicity, we assume the joint distribution of $(\boldsymbol{x}, \boldsymbol{z}^{(1)}, \boldsymbol{z}^{(2)})$ is either discrete or has a continuous density w.r.t. some base measure on $\mathcal{X}^{\otimes 3}$. We abuse the notation $\mathbb{P}(\cdot)$ to refer to either discrete distributions or the density of continuous distributions, with the intended meaning clear from the context. Also, we occasionally omit the subscript kl when referring to KL-sufficiency.

SimCLR [5] learns a representation of the sample $\boldsymbol{x}$ (i.e., $f(\boldsymbol{x})$ or $f(g(\boldsymbol{x}))$) through performing contrastive learning on the augmented views $(\boldsymbol{z}^{(1)}, \boldsymbol{z}^{(2)})$. Specifically, given a batch of $K$ i.i.d. samples $\{\boldsymbol{x}_i\}_{i=1}^{K}$ from $\mathbb{P}_{\mathcal{X}}$, we generate $K$ pairs of augmented views $\{(\boldsymbol{z}_i^{(1)}, \boldsymbol{z}_i^{(2)})\}_{i=1}^{K}$ using $2K$ i.i.d. transformations $\{(g_i^{(1)}, g_i^{(2)})\}_{i=1}^{K}$ from $\mathbb{P}_{\mathcal{G}}$. Let $f : \mathcal{X} \mapsto \mathbb{R}^p$ be an encoder function, potentially parametrized by neural networks. The SimCLR risk function is defined as the expected InfoNCE loss [29]:

$$\overline{\mathsf{R}}_{\mathsf{simclr}, K}(\mathsf{S}) := \frac{1}{2}\mathbb{E}\Big[-\log \frac{\exp(\mathsf{S}(\boldsymbol{z}_1^{(1)}, \boldsymbol{z}_1^{(2)}))}{\sum_{j \in [K]} \exp(\mathsf{S}(\boldsymbol{z}_1^{(1)}, \boldsymbol{z}_j^{(2)}))}\Big] + \frac{1}{2}\mathbb{E}\Big[-\log \frac{\exp(\mathsf{S}(\boldsymbol{z}_1^{(1)}, \boldsymbol{z}_1^{(2)}))}{\sum_{j \in [K]} \exp(\mathsf{S}(\boldsymbol{z}_j^{(1)}, \boldsymbol{z}_1^{(2)}))}\Big], \text{ and}$$
$$(2)$$

$$\mathsf{R}_{\mathsf{simclr}, K}(f) := \overline{\mathsf{R}}_{\mathsf{simclr}, K}(\mathsf{S}_f), \text{ where } \mathsf{S}_f := \tau(\langle f(\boldsymbol{z}^{(1)}), f(\boldsymbol{z}^{(2)})\rangle), \ \tau : \mathbb{R} \mapsto \mathbb{R} \text{ is some simple link function.}$$

Given a set of encoders denoted by $\mathcal{F}$ and $n = n_1 K$ i.i.d. pairs of augmented views $\{(\boldsymbol{z}_i^{(1)}, \boldsymbol{z}_i^{(2)})\}_{i=1}^{n}$, SimCLR learns an encoder function $\widehat{f} \in \mathcal{F}$ through empirical risk minimization (ERM), namely,

$$\widehat{f} := \underset{f \in \mathcal{F}}{\operatorname{argmin}}\Big\{\widehat{\mathsf{R}}_{\mathsf{simclr}, K}(\mathsf{S}_f) := \frac{1}{2n} \sum_{i=1}^{n_1} \Big[\sum_{j=1}^{K} \Big[-\log \frac{\exp(\mathsf{S}_f(\boldsymbol{z}_{(i-1)K+j}^{(1)}, \boldsymbol{z}_{(i-1)K+j}^{(2)}))}{\sum_{l \in [K]} \exp(\mathsf{S}_f(\boldsymbol{z}_{(i-1)K+j}^{(1)}, \boldsymbol{z}_{(i-1)K+l}^{(2)}))}\Big]$$
$$+ \Big[-\log \frac{\exp(\mathsf{S}_f(\boldsymbol{z}_{(i-1)K+j}^{(1)}, \boldsymbol{z}_{(i-1)K+j}^{(2)}))}{\sum_{l \in [K]} \exp(\mathsf{S}_f(\boldsymbol{z}_{(i-1)K+l}^{(1)}, \boldsymbol{z}_{(i-1)K+j}^{(2)}))}\Big]\Big]\Big\}. \quad (3)$$

With the encoder $\widehat{f}(\cdot)$ at hand, $\widehat{f}(\boldsymbol{x})$ (or $\widehat{f}(g(\boldsymbol{x}))$) serves as a representation for each $\boldsymbol{x} \in \mathcal{X}$, which can be used for downstream tasks.

We now show that the sufficiency of the ERM estimator $\widehat{f}$ can be properly controlled. We will demonstrate in Section 3.2 that the downstream performance of $\widehat{f}$ is closely tied to its sufficiency. First, we note that a global minimizer of the SimCLR risk is $\mathsf{S}_{\star}(\boldsymbol{z}^{(1)}, \boldsymbol{z}^{(2)}) := \log \Big[\frac{\mathbb{P}(\boldsymbol{z}^{(1)}, \boldsymbol{z}^{(2)})}{\mathbb{P}(\boldsymbol{z}^{(1)}) \cdot \mathbb{P}(\boldsymbol{z}^{(2)})}\Big]$ (see Lemma 2 for the proof). To analyze the properties of the ERM estimator, we introduce the following boundedness assumption on the score function $\mathsf{S}$ and regularity assumption on $\tau$.

**Assumption 1** (Bounded score). *There exists a constant $B_{\mathsf{S}} > 0$ such that for all pairs $(\boldsymbol{z}^{(1)}, \boldsymbol{z}^{(2)})$, we have $\exp(\mathsf{S}_f(\boldsymbol{z}^{(1)}, \boldsymbol{z}^{(2)})) \in [1/B_{\mathsf{S}}, B_{\mathsf{S}}]$ for all $f \in \mathcal{F}$ and $\frac{\mathbb{P}(\boldsymbol{z}^{(1)}, \boldsymbol{z}^{(2)})}{\mathbb{P}(\boldsymbol{z}^{(1)})\mathbb{P}(\boldsymbol{z}^{(2)})} \in [1/B_{\mathsf{S}}, B_{\mathsf{S}}]$.*

**Assumption 2** (Simple link function). *The link function $\tau : \mathbb{R} \mapsto \mathbb{R}$ is invertible and there exists some constant $B_{\tau} > 0$ such that $|\tau(0)| \leqslant B_{\tau}$ and $\tau, \tau^{-1}$ are $B_{\tau}$-Lipschitz.*

Note that the first part of Assumption 1 is satisfied with $B_{\mathsf{S}} = \exp(B_f^2)$ when $\|f(\boldsymbol{x})\|_2 \leqslant B_f$ for all $f \in \mathcal{F}, \boldsymbol{x} \in \mathcal{X}$ and $\tau$ is the identity function. Based on these assumptions, we have

---

[2] More generally, we only need each transformation $g : \mathcal{X} \to \mathcal{Z}$ maps $\mathcal{X}$ to a space $\mathcal{Z}$, which entails a natural injective map back to $\mathcal{X}$.

**Theorem 1** (Sufficiency bound for the ERM estimator). *Suppose Assumption 1 and 2 hold for some $B_{\mathsf{S}} \geqslant 1, B_\tau > 0$. Let $\widehat{f}$ be the empirical risk minimizer defined in Eq. (3) and let $\mathsf{S}_\star$ be as defined in Section 3.1. Let $\mathsf{supp}(\boldsymbol{z}^{(1)})$ be the support of $\boldsymbol{z}^{(1)}$ and $\mathcal{N}(u, \|\cdot\|_{2,\infty}, \mathcal{F})$ be the $u$-covering number of $\mathcal{F}$ under the $(2,\infty)$-norm $\|f\|_{2,\infty} := \sup_{x \in \mathsf{supp}(\boldsymbol{z}^{(1)})} \|f(x)\|_2$. Then, with probability at least $1 - \delta$, we have*

$$\mathrm{Suff}_{\mathsf{kl}}(\widehat{f}) \leqslant \left(1 + \frac{C}{K}\right) \cdot [\text{generalization error} + \text{approximation error}], \tag{4}$$

*where*

$$\text{generalization error} := \frac{C}{\sqrt{n}}\left[\sqrt{\log(1/\delta)} + B_\tau^2 \int_0^{2(\log B_{\mathsf{S}} + B_\tau)} \sqrt{\log \mathcal{N}(u, \|\cdot\|_{2,\infty}, \mathcal{F})}du\right], \tag{5a}$$

$$\text{approximation error} := \inf_{f \in \mathcal{F}} \overline{\mathsf{R}}_{\mathsf{simclr},K}(\mathsf{S}_f) - \overline{\mathsf{R}}_{\mathsf{simclr},K}(\mathsf{S}_\star) \tag{5b}$$

*for some constant $C > 0$ depending polynomially on $B_{\mathsf{S}}$.*

See the proof in Appendix C.2. In the decomposition on the R.H.S. of (4), the approximation error term represents the error incurred when approximating the optimal score $\mathsf{S}_\star$ within the function class $\mathcal{F}$. It is a property of the function class $\mathcal{F}$, and a richer class tends to have a smaller approximation error. The generalization error bound is derived using concentration properties of functions with bounded differences. Notably, it depends only on the total sample size $n = n_1 K$ rather than the batch size $K$ or the number of batches $n_1$. This allows our results to account for large or full-batch training, as used in SimCLR [5] and CLIP [31]. When $n \to \infty$, the generalization error vanishes while the approximation error remains constant.

**Why does the SimCLR loss work?** Intuitively, $\overline{\mathsf{R}}_{\mathsf{simclr},K}(\mathsf{S})$ can be viewed as an approximation of the KL-contrastive loss $R_{\mathsf{kl}}(\mathsf{S})$ in Eq. (1) using a finite batch size $K$. Namely,

$$R_{\mathsf{kl}}(\mathsf{S}) = -\mathbb{E}[\mathsf{S}(\boldsymbol{z}^{(1)}, \boldsymbol{z}^{(2)})] + \mathbb{E}_{\boldsymbol{z}_1^{(1)}}\left[\log \mathbb{E}_{\boldsymbol{z}_2^{(2)}}[\exp(\mathsf{S}(\boldsymbol{z}_1^{(1)}, \boldsymbol{z}_2^{(2)}))]\right] = \lim_{K \to \infty} \overline{\mathsf{R}}_{\mathsf{simclr},K}(\mathsf{S}) - \log K. \tag{6}$$

See the proof in Appendix C.1. As a result, by the definition of VFS in Definition 1

$$\mathrm{Suff}_{\mathsf{kl}}(f) \leqslant R_{\mathsf{kl}}(\mathsf{S}_f) - \inf_{\mathsf{S}} R_{\mathsf{kl}}(\mathsf{S}) \approx \underbrace{\overline{\mathsf{R}}_{\mathsf{simclr},K}(\mathsf{S}_f) - \inf_{\mathsf{S}} \overline{\mathsf{R}}_{\mathsf{simclr},K}(\mathsf{S})}_{\text{Excess risk}},$$

and thus minimizing the SimCLR loss $\widehat{\mathsf{R}}_{\mathsf{simclr},K}(\mathsf{S}_f)$ effectively controls the sufficiency $\mathrm{Suff}_{\mathsf{kl}}(f)$.

### 3.2 Using the encoder for downstream tasks

Given an encoder function $f : \mathcal{X} \to \mathbb{R}^p$, we are interested in applying it to downstream tasks. Specifically, the goal is to leverage the learned representation $f(\boldsymbol{x})$ (or $f(g(\boldsymbol{x}))$) to facilitate learning in downstream tasks, such as regression or classification. By mapping the raw sample $\boldsymbol{x}$ to the feature space $\mathbb{R}^p$, the representation $f(\boldsymbol{x})$ (or $f(g(\boldsymbol{x}))$) is expected to capture the most salient information of $\boldsymbol{x}$, simplifying the downstream task while maintaining high performance. In this section, we demonstrate that the downstream performance of the encoder depends on its sufficiency $\mathrm{Suff}_{\mathsf{kl}}(f)$ and the robustness of the downstream task to the random transformation $g \sim \mathbb{P}_{\mathcal{G}}$.

**Adaptation to downstream regression tasks.** We first study regression tasks. Consider the task of learning an unknown target function $h_\star : \mathcal{X} \mapsto \mathbb{R}$. Given an encoder $f$, our objective is to find a function $\mathsf{h} : \mathbb{R}^p \mapsto \mathbb{R}$ such that $\mathsf{h}(f(\boldsymbol{x})) \approx h_\star(\boldsymbol{x})$ (or $\mathsf{h}(f(g(\boldsymbol{x}))) \approx h_\star(\boldsymbol{x})$). The estimation error of $\mathsf{h}$ is measured by the risk

$$\mathsf{R}_{\mathcal{G}}(\mathsf{h} \circ f) := \mathbb{E}_{\boldsymbol{x} \sim \mathbb{P}_{\mathcal{X}}, g \sim \mathbb{P}_{\mathcal{G}}}[(\mathsf{h}(f(g(\boldsymbol{x}))) - h_\star(\boldsymbol{x}))^2], \quad \text{or} \quad \mathsf{R}(\mathsf{h} \circ f) := \mathbb{E}_{\boldsymbol{x} \sim \mathbb{P}_{\mathcal{X}}}[(\mathsf{h}(f(\boldsymbol{x})) - h_\star(\boldsymbol{x}))^2].$$

For example, in regression tasks where the goal is to predict the outcome $\boldsymbol{y}$ based on the covariates $\boldsymbol{x}$, one can choose $h_\star(\boldsymbol{x}) = \mathbb{E}[\boldsymbol{y}|\boldsymbol{x}]$. The two risks $\mathsf{R}_{\mathcal{G}}(\cdot), \mathsf{R}(\cdot)$ correspond to the cases where a random transformation $g$ is (or is not) applied before passing the input to the encoder $f$, respectively. Theorem 2 illustrates how the downstream performance of the encoder $f$ depends on its sufficiency.

**Theorem 2** (Performance on downstream regression). *Suppose $h_\star$ satisfies $\left|\mathbb{E}[h_\star(\boldsymbol{x})|g(\boldsymbol{x})]\right| \leqslant B_{h_\star}$ almost surely. Given an encoder $f : \mathcal{X} \mapsto \mathbb{R}^p$, there exists a measurable function $\mathsf{h} : \mathbb{R}^p \mapsto \mathbb{R}$ such that*

$$\mathsf{R}_{\mathcal{G}}(\mathsf{h} \circ f) \leqslant c(B_{h_\star}^2 \sqrt{\mathrm{Suff}_{\mathsf{kl}}(f)} + \epsilon_{\mathcal{G}}), \tag{7a}$$

*where $c > 0$ is some absolute constant and $\epsilon_{\mathcal{G}} := \mathbb{E}_{\boldsymbol{x} \sim \mathbb{P}_{\mathcal{X}}, g \sim \mathbb{P}_{\mathcal{G}}}[(h_\star(g(\boldsymbol{x})) - h_\star(\boldsymbol{x}))^2]$. Moreover, if the augmented view has the same marginal distribution as the original sample, i.e., $\boldsymbol{z}^{(1)} \overset{d}{=} \boldsymbol{x}$, then*

$$\mathsf{R}(\mathsf{h} \circ f) \leqslant c(B_{h_\star}^2 \sqrt{\mathrm{Suff}_{\mathsf{kl}}(f)} + \epsilon_{\mathcal{G}}) \tag{7b}$$

*for some absolute constant $c > 0$.*

The proof of Theorem 2 is contained in Appendix C.3. The term $\epsilon_{\mathcal{G}}$ characterizes the impact of a random transformation $g$ on the value of the target function $h_\star$. In SimCLR, since the encoder $f$ is trained only on the augmented views $(\boldsymbol{z}^{(1)}, \boldsymbol{z}^{(2)})$, the random transformation $g$ need to preserve sufficient information on $h_\star$ (e.g., $\epsilon_{\mathcal{G}}$ is small) for $f$ to be effective. This is often the case in practice: for example, random cropping ($g$) typically does not alter the class label ($h_\star$) of an image; similarly, rotations and scaling ($g$) should not affect the true age ($h_\star$) of a person in facial images. In addition, Eq. (7a) still holds when $\epsilon_{\mathcal{G}}$ is replaced by the minimum error $\widetilde{\epsilon}_{\mathcal{G}} := \inf_h \mathbb{E}_{\boldsymbol{x} \sim \mathbb{P}_{\mathcal{X}}, g \sim \mathbb{P}_{\mathcal{G}}}[(h(g(\boldsymbol{x})) - h_\star(\boldsymbol{x}))^2] \leqslant \epsilon_{\mathcal{G}}$. We refer to the proof for more details.

**Adaptation to downstream classification tasks.** We next turn to classification tasks. Suppose in the downstream we are given samples $(\boldsymbol{x}, \boldsymbol{y})$ from some joint distribution $\mathbb{P}$ on $\mathcal{X} \times [\mathsf{K}]$, where $\boldsymbol{x} \sim \mathbb{P}_{\mathcal{X}}$ is the input and $\boldsymbol{y} \in [\mathsf{K}]$ is the corresponding label. Note that for any $\boldsymbol{x}$, the label $\boldsymbol{y}$ follows the conditional probability $\mathbb{P}(\boldsymbol{y}|\boldsymbol{x})$. Given an encoder $f$, for any function $\mathsf{h} : \mathbb{R}^p \mapsto \Delta([\mathsf{K}])$, we measure its classification error by

$$\mathsf{R}_{\mathcal{G}}^{\mathsf{cls}}(\mathsf{h} \circ f) := \mathbb{E}_{(\boldsymbol{x}, \boldsymbol{y}) \sim \mathbb{P}, g}[\mathsf{D}_{\mathrm{KL}}(\mathbb{P}(\boldsymbol{y}|\boldsymbol{x})||\mathsf{h}(f(g(\boldsymbol{x}))))].$$

**Theorem 3** (Performance on downstream classification). *Suppose $\inf_{y \in [\mathsf{K}]} \mathbb{P}(y|g(\boldsymbol{x})) \geqslant \exp(-B)$ for some $B > 0$ on the support of $g(\boldsymbol{x})$. Given an encoder $f : \mathcal{X} \mapsto \mathbb{R}^p$, there exists a measurable function $\mathsf{h} : \mathbb{R}^p \mapsto \Delta([\mathsf{K}])$ such that*

$$\mathsf{R}_{\mathcal{G}}^{\mathsf{cls}}(\mathsf{h} \circ f) \leqslant c\left(B\sqrt{\mathrm{Suff}_{\mathsf{kl}}(f)} + \epsilon_{\mathcal{G}}^{\mathsf{cls}}\right), \tag{8}$$

*where $\epsilon_{\mathcal{G}}^{\mathsf{cls}} := \mathbb{E}_{\boldsymbol{x} \sim \mathbb{P}_{\mathcal{X}}, g \sim \mathbb{P}_{\mathcal{G}}}[\mathsf{D}_2(\mathbb{P}(\boldsymbol{y}|\boldsymbol{x})||\mathbb{P}(\boldsymbol{y}|\boldsymbol{z})) + \mathsf{D}_2(\mathbb{P}(\boldsymbol{y}|\boldsymbol{z})||\mathbb{P}(\boldsymbol{y}|\boldsymbol{x}))]$ and $c > 0$ is some absolute constant. Here, $\mathsf{D}_2$ denotes the 2-Rényi divergence.*

The proof of Theorem 3 is contained in Appendix C.4. Similar to the regression case in Theorem 2, the downstream classification error is bounded by the sum of a sufficiency term and an error term that characterizes the change in label probabilities induced by the transformation $g$.

### 3.3 General f-contrastive learning

We generalize our theoretical framework to using general f-sufficiency as defined in Definition 1, which could be controlled by minimizing the f-contrastive learning risk. We discuss (1) how to find encoders $f$ with low f-sufficiency $\mathrm{Suff}_{\mathsf{f}}(f)$ via data augmentation-based contrastive learning and (2) the implications of low f-sufficiency on downstream performance. Note that $\mathsf{f}(x) = x \log x$ yields the standard SimCLR setup.

#### 3.3.1 Finding encoders with low f-sufficiency

Recall the variational form sufficiency (VFS) in Definition 1. We see that for any $\mathsf{f}$ and encoder $f$

$$\mathrm{Suff}_{\mathsf{f}}(f) \leqslant \inf_{\mathsf{S}: f(\mathcal{X}) \times \mathcal{X} \mapsto \mathbb{R}} R_{\mathsf{f}}(\mathsf{S} \circ f) - \inf_{\mathsf{S}: \mathcal{X} \times \mathcal{X} \mapsto \mathbb{R}} R_{\mathsf{f}}(\mathsf{S}) \leqslant \underbrace{R_{\mathsf{f}}(\mathsf{S}_f) - \inf_{\mathsf{S}: \mathcal{X} \times \mathcal{X} \mapsto \mathbb{R}} R_{\mathsf{f}}(\mathsf{S})}_{\text{Excess risk}}.$$

Thus, for any $\varepsilon > 0$, if there exists an encoder $\widehat{f} \in \mathcal{F}$ such that the excess risk of $\mathsf{S}_{\widehat{f}}$ is less than $\varepsilon$, then the sufficiency $\mathrm{Suff}_{\mathsf{f}}(\widehat{f}) \leqslant \varepsilon$. Consequently, given i.i.d. pairs of augmented views, we can

obtain an encoder $\widehat{f}$ with low f-sufficiency by choosing $\widehat{f}$ as the empirical risk minimizer (ERM) of a finite-sample estimate $\widehat{R}_{\mathrm{f}}(\mathsf{S}_f)$ of $R_{\mathrm{f}}(\mathsf{S}_f)$, provided that $\widehat{R}_{\mathrm{f}}(\mathsf{S}_f) \approx R_{\mathrm{f}}(\mathsf{S}_f)$, the function class $\mathcal{F}$ is sufficiently rich, and its $\|\cdot\|_{2,\infty}$-covering number is well-controlled.

We focus on $\chi^2$-sufficiency (i.e., $\mathrm{f}(x) = (x-1)^2/2$) in the following. For general f, the $\mathsf{S}_x(x)$ that attains the infimum in Eq. (1) may not have a closed-form solution, and estimating $\widehat{R}_{\mathrm{f}}(\mathsf{S}_f)$ requires solving estimating equations, adding complexity to the analysis. Thus, we leave a detailed investigation of the general f case for future work.

When $\mathrm{f}(x) = (x-1)^2/2$, basic algebra shows that the $\chi^2$-contrastive loss (1) takes the form

$$R_{\chi^2}(\mathsf{S}) = \mathbb{E}_{\mathbb{P}(x,y)}[-\mathsf{S}(x,y)] + \mathbb{E}_{\mathbb{P}(x)\mathbb{P}(y)}[(\mathsf{S}(x,y) - \mathbb{E}_{\mathbb{P}(y)}[\mathsf{S}(x,y)])^2/2 + \mathsf{S}(x,y)]. \qquad (9)$$

Given $n = n_1 K$ i.i.d. pairs of augmented views $\{(\boldsymbol{z}_i^{(1)}, \boldsymbol{z}_i^{(2)})\}_{i=1}^n$, an unbiased finite-sample estimate of $R_{\chi^2}(\mathsf{S})$ gives

$$\widehat{\mathsf{R}}_{\mathsf{chisq},K}(\mathsf{S}_f) := \frac{1}{n} \sum_{i=1}^{n_1} \sum_{j=1}^{K} \Bigg[ \frac{1}{4(K-1)(K-2)} \sum_{\substack{k,l \in [K] \\ j \neq k,\ k \neq l,\ l \neq j}} \left( \mathsf{S}_f(\boldsymbol{z}_{ij}^{(1)}, \boldsymbol{z}_{ik}^{(2)}) - \mathsf{S}_f(\boldsymbol{z}_{ij}^{(1)}, \boldsymbol{z}_{il}^{(2)}) \right)^2$$

$$+ \frac{1}{K-1} \sum_{k \neq j} \mathsf{S}_f(\boldsymbol{z}_{ij}^{(1)}, \boldsymbol{z}_{ik}^{(2)}) - \mathsf{S}_f(\boldsymbol{z}_{ij}^{(1)}, \boldsymbol{z}_{ij}^{(2)}) \Bigg], \quad \mathsf{S}_f := \tau(\langle f(\boldsymbol{z}^{(1)}), f(\boldsymbol{z}^{(2)})\rangle), \quad (10)$$

where we adopt the shorthand $\boldsymbol{z}_{ab}^{(i)} = \boldsymbol{z}_{(a-1)K+b}^{(i)}$ for $i \in [2]$. Let $\widehat{f} = \mathrm{argmin}_{f \in \mathcal{F}} \widehat{\mathsf{R}}_{\mathsf{chisq},K}(\mathsf{S}_f)$ be the ERM estimator. Similar to Theorem 1, we have

**Theorem 4** ($\chi^2$-sufficiency bound for the ERM estimator). *Suppose* $\mathsf{S}_f(\boldsymbol{z}^{(1)}, \boldsymbol{z}^{(2)}) \in [-\bar{B}_{\mathsf{S}}, \bar{B}_{\mathsf{S}}]$ *for all* $f \in \mathcal{F}$ *and pairs* $(\boldsymbol{z}^{(1)}, \boldsymbol{z}^{(2)})$, *and that Assumption 2 holds for some* $B_\tau > 0$. *Let* $\mathsf{S}_\star(\boldsymbol{z}^{(1)}, \boldsymbol{z}^{(2)}) := \frac{\mathbb{P}(\boldsymbol{z}^{(1)}, \boldsymbol{z}^{(2)})}{\mathbb{P}(\boldsymbol{z}^{(1)})\mathbb{P}(\boldsymbol{z}^{(2)})}$. *For any* $K \geqslant 3$, *with probability at least* $1 - \delta$, *we have*

$$\mathrm{Suff}_{\chi^2}(\widehat{f}) \leqslant \text{generalization error} + \text{approximation error}, \qquad (11)$$

*where*

$$\text{generalization error} := \frac{c\bar{B}_{\mathsf{S}}^2}{\sqrt{n}} \Big[ \sqrt{\log(1/\delta)} + B_\tau^2 \int_0^{2(\bar{B}_{\mathsf{S}}+B_\tau)} \sqrt{\log \mathcal{N}(u, \|\cdot\|_{2,\infty}, \mathcal{F})} du \Big],$$
$$\text{approximation error} := \inf_{f \in \mathcal{F}} R_{\chi^2}(\mathsf{S}_f) - R_{\chi^2}(\mathsf{S}_\star)$$

*for some absolute constant* $c > 0$.

The proof of Theorem 4 is provided in Appendix C.5. Note that we do not assume the boundedness of $\mathsf{S}_\star$ as in Theorem 1.

### 3.3.2 Implications of low f-Sufficiency

Similar to the KL case in Section 3.2, the downstream performance of $f$ can be controlled by its f-sufficiency for a broad class of f considered in Definition 1. Recall the CBS form in Definition 1.

**Proposition 5** (f-sufficiency bound on downstream performance). *The results in Theorem 2 and 3 hold with* $\mathrm{Suff}_{\mathsf{kl}}(f)$ *replaced by* $c_2^2 \cdot \mathrm{Suff}_{\mathrm{f}}(f)$ *for some value* $c_2 > 0$ *if*

$$\mathbb{E}_{\boldsymbol{z}^{(1)}, \boldsymbol{z}^{(2)}}[\mathsf{D}_{\mathrm{TV}}(\mathbb{P}(\cdot|\boldsymbol{z}^{(1)})\|\mathbb{P}_{\boldsymbol{z}^{(2)}|\boldsymbol{z}^{(1)}}(\cdot|f(\boldsymbol{z}^{(1)})))] \leqslant c_2 \cdot \sqrt{\mathrm{Suff}_{\mathrm{f}}(f)}. \qquad (13)$$

Proposition 5 follows immediately by noting that, in the proof of Theorem 2 and 3, $\mathrm{Suff}_{\mathsf{kl}}(f)$ is only used as an upper bound of the expected total variation distance (e.g., by Pinsker's inequality). It can be verified that KL-divergence and $\chi^2$-divergence satisfy Eq. (13) with $c_2 = 1/\sqrt{2}$. Let $r = \mathbb{P}(\boldsymbol{z}^{(1)}, \boldsymbol{z}^{(2)})/[\mathbb{P}(\boldsymbol{z}^{(1)})\mathbb{P}(\boldsymbol{z}^{(2)})]$ denote the density ratio. Moreover, for general f, we can choose $c_2 = (2\inf_{(\boldsymbol{z}^{(1)}, \boldsymbol{z}^{(2)})} \mathrm{f}''(r))^{-1/2}$, which is bounded when f is strongly convex on the range of the density ratio $r$. For example, we can choose $c_2 = \sqrt{2}B^{3/4}$ when $\mathrm{f}(x) = 1 - \sqrt{x}$ corresponds to squared Hellinger-sufficiency if the density ratio $r \leqslant B$ for all pairs $(\boldsymbol{z}^{(1)}, \boldsymbol{z}^{(2)})$. We refer the readers to Lemma 3 in Appendix B.2 for further details. Combining the results from Sections 3.3.1 and 3.3.2, we provide end-to-end theoretical guarantees for the downstream performance of encoders obtained by minimizing the $\chi^2$-contrastive losses.

# 4 Examples

In this section, we present concrete examples on linear regression and topic classification to illustrate the applicability of our general results in Section 3.

## 4.1 Linear regression

Let $\boldsymbol{x}$ follow some distribution $\mathbb{P}_{\mathcal{X}}$ on $\mathcal{X} \subseteq \mathbb{R}^d$. We assume the downstream task is linear regression, where we observe samples of the form $(\boldsymbol{x}, \boldsymbol{y}) \in \mathbb{R}^d \times \mathbb{R}$, with $\boldsymbol{y} = \langle \boldsymbol{x}, \boldsymbol{\theta}_\star \rangle + \varepsilon$ for some unknown parameter $\boldsymbol{\theta}_\star \in \mathbb{R}^d$ and zero-mean noise $\varepsilon$ independent of $\boldsymbol{x}$. The goal is to predict $\boldsymbol{y}$ given $\boldsymbol{x}$. While fitting a linear model using only the downstream samples yields a risk of order $\mathcal{O}(d/m)$, a smaller risk may be achieved by fitting a linear model on a low-dimensional representation $f(\boldsymbol{z}) \in \mathbb{R}^p$, where $p \ll d$, that captures sufficient information about $\boldsymbol{x}$ relevant to the downstream task. Theorem 6 gives a theoretical guarantee for learning the downstream task using a given linear encoder.

**Theorem 6** (Linear regression with encoder representation). *Let $p \leqslant d$. Suppose we are given a linear encoder $f(\boldsymbol{z}) = \boldsymbol{W}\boldsymbol{z}$ for some $\boldsymbol{W} \in \mathbb{R}^{p \times d}$ and $m$ i.i.d. samples $\{(\boldsymbol{x}_i, \boldsymbol{y}_i)\}_{i=1}^m$ from the downstream linear model $\boldsymbol{y} = \langle \boldsymbol{x}, \boldsymbol{\theta}_\star \rangle + \varepsilon$, where $\varepsilon \sim \mathcal{N}(0, \bar{\sigma}^2) \perp\!\!\!\perp \boldsymbol{x}$. Suppose $\sup_{\boldsymbol{x} \in \mathcal{X}} \|\boldsymbol{x}\|_2 \leqslant B_{\boldsymbol{x}}, \|\boldsymbol{\theta}_\star\|_2 \leqslant B_{\boldsymbol{\theta}}$ for some $B_{\boldsymbol{x}}, B_{\boldsymbol{\theta}} > 0$ and let $B = B_{\boldsymbol{x}}B_{\boldsymbol{\theta}}$. Also assume that $\mathbb{E}[(\mathrm{I}_d - \boldsymbol{W}^\dagger \boldsymbol{W})\boldsymbol{z}|\boldsymbol{W}\boldsymbol{z}] = 0$ almost surely. Consider fitting a (random) linear model $\mathsf{h}_{\widehat{\boldsymbol{\eta}}}(\boldsymbol{x}) = \langle f(\boldsymbol{z}), \widehat{\boldsymbol{\eta}} \rangle$ by ordinary least squares, i.e.,*

$$\widehat{\boldsymbol{\eta}} := \mathrm{argmin}_{\boldsymbol{\eta} \in \mathbb{R}^p} \left\{ \widehat{\mathsf{R}}_{\mathsf{lin}}(\mathsf{h}_{\boldsymbol{\eta}}) := \frac{1}{m} \sum_{i=1}^m (\langle f(\boldsymbol{z}_i), \boldsymbol{\eta} \rangle - \boldsymbol{y}_i)^2 \right\},$$

*where $\boldsymbol{z} = g(\boldsymbol{x})$, $\boldsymbol{z}_i = g_i(\boldsymbol{x}_i)$, and $g, \{g_i\}_{i=1}^m$ are i.i.d. transformations from $\mathbb{P}_{\mathcal{G}}$. Then the expected risk of the truncated linear model $\widetilde{\mathsf{h}}_{\widehat{\boldsymbol{\eta}}}(\boldsymbol{x}) := \mathrm{proj}_{[-B,B]}(\mathsf{h}_{\widehat{\boldsymbol{\eta}}}(\boldsymbol{x}))$ satisfies*

$$\mathbb{E}[\mathsf{R}_{\mathsf{lin}}(\widetilde{\mathsf{h}}_{\widehat{\boldsymbol{\eta}}})] := \mathbb{E}\left[\mathbb{E}_{\boldsymbol{x},\boldsymbol{y},g}[(\boldsymbol{y} - \widetilde{\mathsf{h}}_{\widehat{\boldsymbol{\eta}}}(\boldsymbol{x}))^2]\right] \leqslant \underbrace{\bar{\sigma}^2}_{\text{irreducible risk}} + c\left((B^2 c_2 \sqrt{\mathrm{Suff}_{\mathsf{cb,f}}(f)} + \epsilon_{\mathcal{G}}) + (\bar{\sigma}^2 + B^2)\frac{p \log m}{m}\right),$$

*where $\epsilon_{\mathcal{G}} = \mathbb{E}[\langle \boldsymbol{x} - \boldsymbol{z}, \boldsymbol{\theta}_\star \rangle^2]$ and the outer expectation is over $\{(\boldsymbol{x}_i, \boldsymbol{y}_i, g_i)\}_{i=1}^n$, and $c_2 > 0$ is any value that satisfies Eq. (13).*

See the proof of Theorem 6 and more discussion in Appendix D.1. Compared to fitting a linear model using $\boldsymbol{x} \in \mathbb{R}^d$, which yields an excess risk of $\mathcal{O}(d/m)$, Theorem 6 achieves a smaller excess risk of order $\widetilde{\mathcal{O}}(p/m)$ when $p \ll d$ and $f(g(\boldsymbol{x}))$ is a "good" representation of $\boldsymbol{x}$, in the sense that $\mathrm{Suff}_{\mathsf{f}}(f)$ and $\epsilon_{\mathcal{G}}$ are sufficiently small. In Appendix D.2, we present a scenario where a linear encoder $f$ with low KL-sufficiency $\mathrm{Suff}_{\mathsf{kl}}(f)$ can be efficiently learned by minimizing the SimCLR loss in Eq. (3). Specifically, we consider a case where two augmented views $(\boldsymbol{z}^{(1)}, \boldsymbol{z}^{(2)})$ follow a joint von Mises-Fisher (vMF) distribution [8] on a low-dimensional unit sphere, allowing $\mathsf{S}_\star$ to be realized by $\mathsf{S}_f$ for some linear encoder $f$. Combined with Theorem 6, this yields an end-to-end result on the downstream performance of the SimCLR-trained encoder.

## 4.2 Topic classification

We also demonstrate our results in a classification setting. Let $\mathcal{Y} = \{1, 2, \ldots, M\}$ represent a set of classes. A sample $\boldsymbol{x}$ is generated by first selecting a class $\boldsymbol{y} \in \mathcal{Y}$ from some distribution $\mathbb{P}_{\mathcal{Y}}$, and then drawing $\boldsymbol{x} = (\boldsymbol{x}^{c_1}, \boldsymbol{x}^{c_2}) \in [S] \times [S]$ conditioned on $\boldsymbol{y}$, with the joint distribution

$$\mathbb{P}(\boldsymbol{x}|\boldsymbol{y}) = \mathbb{P}_c(\boldsymbol{x}^{c_1}|\boldsymbol{y}) \times \mathbb{P}_c(\boldsymbol{x}^{c_2}|\boldsymbol{y}),$$

where $\mathbb{P}_c(\cdot|\boldsymbol{y})$ is some conditional distribution over $[S]$. For example, in a topic classification task, each sample consists of a two-part sentence (or a two-word phrase), with the class $\boldsymbol{y}$ representing the topic (e.g., sports, technology, or health). The first and second parts (or words), $\boldsymbol{x}^{c_1}$ and $\boldsymbol{x}^{c_2}$, are independently sampled from a vocabulary of size $S$, conditioned on the topic $\boldsymbol{y}$.

**Contrastive learning.** We use the random dropout transformation $g : [S] \times [S] \to [S]$, which selects one component $\boldsymbol{x}^{c_i}$ from the pair $(\boldsymbol{x}^{c_1}, \boldsymbol{x}^{c_2})$ with equal probability as the augmented view $\boldsymbol{z}$ and drops the other. Denote the augmented view $\boldsymbol{z}$ using one-hot encoding. We consider encoders $f$ that are linear functions of $\boldsymbol{z}$ augmented with the one-hot encoding, i.e., consider the encoder space

$$\mathcal{F} = \{f_{\mathsf{aug}} : \cup_{i=1}^S \{e_i\} \mapsto \mathbb{R}^{M+S} | f_{\mathsf{aug}}(\boldsymbol{z}) = ((\boldsymbol{W}\boldsymbol{z})^\top, w\boldsymbol{z}^\top)^\top, \boldsymbol{W} \in \mathbb{R}^{M \times S}, w \in \mathbb{R}, \|\boldsymbol{W}\|_{2,\infty} \vee |\frac{w}{\sqrt{S}}| \leqslant B_{\boldsymbol{W}}\}$$

with $B_{\boldsymbol{W}} = M$. To learn an encoder $\widehat{f}_{\mathsf{aug}}$, we minimize the $\chi^2$-contrastive loss computed using $n$ i.i.d. pairs of augmented views via Eq. (10). Importantly, class labels $\{\boldsymbol{y}_i\}_{i=1}^n$ remain *unobservable* during contrastive learning.

**Downstream classification.** Let $\widehat{f}_{\mathsf{aug}}(\boldsymbol{z}) = ((\widehat{\boldsymbol{W}}\boldsymbol{z})^\top, \widehat{w} \cdot \boldsymbol{z}^\top)^\top$ be the learned representation, and define the encoder as $\widehat{f}(\boldsymbol{z}) := \widehat{\boldsymbol{W}}\boldsymbol{z} \in \mathbb{R}^M$. We train a linear classifier on $\widehat{f}$ to predict the conditional topic distribution $\mathbb{P}(\boldsymbol{y} = y|\boldsymbol{x})_{y \in [M]} \in \mathbb{R}^M$. Define $\boldsymbol{E}_\star \in \mathbb{R}^{M \times S}$ such that $\boldsymbol{E}_{\star, \cdot j} = \left( \frac{\mathbb{P}_c(\boldsymbol{y}=1|\boldsymbol{x}^{c_1}=j)}{\sqrt{\mathbb{P}_{\mathcal{Y}}(\boldsymbol{y}=1)}}, \ldots, \frac{\mathbb{P}_c(\boldsymbol{y}=M|\boldsymbol{x}^{c_1}=j)}{\sqrt{\mathbb{P}_{\mathcal{Y}}(\boldsymbol{y}=M)}} \right)^\top$ for $j \in [S]$. Assume that (a) the marginal distributions of $\boldsymbol{y}$ and $\boldsymbol{x}^{c_1}$ are uniform over $[M]$ and $[S]$, respectively; (b) the minimum singular value $\sigma_{\min}(\boldsymbol{E}_\star \boldsymbol{E}_\star^\top) \geqslant \sigma_{\boldsymbol{E}_\star}^2$ for some $\sigma_{\boldsymbol{E}_\star} > 0$; (c) $S \geqslant 4M$ and $\inf_{y \in [M], s \in [S]} \mathbb{P}_c(y|s) \geqslant \exp(-B)$ for some $B > 0$.

**Theorem 7** (Classification using the $\chi^2$-trained encoder). *Under the setup and assumptions in Section 4.2 and let $\widehat{f}_{\mathsf{aug}}$ be the ERM in Eq. (10). Then, with probability at least $1 - \delta$,*

$$\mathrm{Suff}_{\chi^2}(\widehat{f}_{\mathsf{aug}}) \leqslant R_{\mathrm{f}}(\mathsf{S}_{\widehat{f}_{\mathsf{aug}}}) - R_{\mathrm{f}}(\mathsf{S}_\star) =: \mathrm{Suff}_{\chi^2}(\mathsf{S}_{\widehat{f}_{\mathsf{aug}}}) \leqslant \frac{cS^2 M^4}{\sqrt{n}} \Big[ \sqrt{\log(1/\delta)} + \sqrt{S} M^{1.5} \Big] \quad (14)$$

*for some absolute constant $c > 0$.*

*In downstream classification, given $m$ i.i.d. samples $\{(\boldsymbol{x}_i, \boldsymbol{y}_i)\}_{i=1}^m$, consider fitting a multi-class classifier $\mathsf{h}_{\widehat{\boldsymbol{\Gamma}}}(\boldsymbol{x}) = \bar{\mathsf{h}}_{\widehat{\boldsymbol{\Gamma}}}(\widehat{f}(\boldsymbol{z})) := \mathrm{softmax}(\log \mathsf{trun}(\widehat{\boldsymbol{\Gamma}}_w \widehat{f}(\boldsymbol{z}) + \widehat{\boldsymbol{\Gamma}}_b))$ with*

$$\widehat{\boldsymbol{\Gamma}} := \operatorname{argmin}_{\boldsymbol{\Gamma}_w \in \mathbb{R}^{M \times M}, \boldsymbol{\Gamma}_b \in \mathbb{R}^M, \; \|\boldsymbol{\Gamma}_w\|_{op} \vee \|\boldsymbol{\Gamma}_b\|_2 \leqslant B_\Gamma} \left\{ \widehat{\mathsf{R}}_{\mathrm{cls}}(\mathsf{h}_\Gamma) := -\frac{1}{m} \sum_{i=1}^m \log \bar{\mathsf{h}}_\Gamma(\widehat{f}(\boldsymbol{z}_i))_{\boldsymbol{y}_i} \right\}, \quad (15)$$

*where $\boldsymbol{z} = g(\boldsymbol{x})$, $\boldsymbol{z}_i = g_i(\boldsymbol{x}_i)$ and $g, \{g\}_{i=1}^m$ are i.i.d. dropout transformations, $B_\Gamma \geqslant 4\sqrt{S} M/\sigma_{\boldsymbol{E}_\star}$, and $\mathsf{trun}(x) := \mathrm{proj}_{[\exp(-B), 1]}(x)$. Then there exists some absolute constants $c, c' > 0$ such that, given the encoder $\widehat{f}$ and suppose $\mathrm{Suff}_{\chi^2}(\mathsf{S}_{\widehat{f}_{\mathsf{aug}}}) \leqslant c' \frac{\sigma_{\boldsymbol{E}_\star}^2}{S^2 M}$, with probability at least $1 - \delta_1$*

$$\mathsf{R}_{\mathrm{cls}}(\bar{\mathsf{h}}_{\widehat{\boldsymbol{\Gamma}}}) := \mathbb{E}_{\boldsymbol{x}, \boldsymbol{y}, g}[\mathsf{D}_{\mathrm{KL}}(\mathbb{P}(\boldsymbol{y}|\boldsymbol{x}) || \mathsf{h}_{\widehat{\boldsymbol{\Gamma}}}(\widehat{f}(g(\boldsymbol{x}))))]$$

$$\leqslant c \Big( \underbrace{\Big[ \epsilon_{\mathcal{G}}^{\mathsf{cls}} + \frac{S \exp(B)}{\sigma_{\boldsymbol{E}_\star}^2} \cdot \mathrm{Suff}_{\chi^2}(\mathsf{S}_{\widehat{f}_{\mathsf{aug}}}) \Big]}_{\text{approximation error}} + \underbrace{\frac{B}{\sqrt{m}} \Big[ \sqrt{\log(1/\delta_1)} + M(\sqrt{\log B_\Gamma} + \sqrt{B}) \Big]}_{\text{generalization error}} \Big).$$

See the proof in Appendix D.5. Note that the bound on downstream classification depends on the sufficiency of the score function $\mathrm{Suff}_{\chi^2}(\mathsf{S}_{\widehat{f}_{\mathsf{aug}}})$, introduced in Appendix B.3, rather than $\mathrm{Suff}_{\chi^2}(\widehat{f})$. This distinction arises because we restrict ourselves to linear classifiers, whereas Theorem 3 considers arbitrary measurable functions, leading to an additional approximation error term.

## 5 Experiments

We conduct synthetic experiments to learn data representations via contrastive learning using two-layer neural networks, and evaluate them on downstream linear regression.

In the contrastive learning stage, we generate $n$ i.i.d. samples $\boldsymbol{x}_i \sim \mathcal{N}(0, \mathrm{I}_d)$. The augmentation $g$ adds i.i.d. $\mathcal{N}(0, \sigma_1^2)$ noise to the first $s < d$ coordinates of $\boldsymbol{x}_i$, and replaces the remaining coordinates with i.i.d. $\mathcal{N}(0, 1)$ noise. We apply KL and $\chi^2$-contrastive learning (Eq. 3 and 10) with link function $\tau(x) = x$, and encoder $f(\cdot)$ being a two-layer ReLU neural network mapping $\mathbb{R}^d$ to $\mathbb{R}^s$. We set $s = 10, d = 100, n = 500$, hidden dimension 64, and batch size $K = 64$. The encoder is trained using Adam (learning rate 0.001) for 1000 epochs until convergence.

For downstream regression, we generate $m$ i.i.d. samples $(\boldsymbol{x}_i, \boldsymbol{y}_i)$, where $\boldsymbol{x}_i \sim \mathcal{N}(0, \mathrm{I}_d)$ and $\boldsymbol{y}_i = \langle \boldsymbol{x}_i, \boldsymbol{\theta}_\star \rangle + \varepsilon_i$, with $\varepsilon_i \sim \mathcal{N}(0, \sigma^2)$ independent of $\boldsymbol{x}_i$. We choose $\boldsymbol{\theta}_\star = (\mathbf{1}_s^\top/\sqrt{s}, \mathbf{0}_{d-s}^\top)^\top$ and $\sigma = 1$. Using the learned representation $\widehat{f}(\boldsymbol{x}_i) \in \mathbb{R}^s$ from KL (or $\chi^2$)-contrastive learning, we

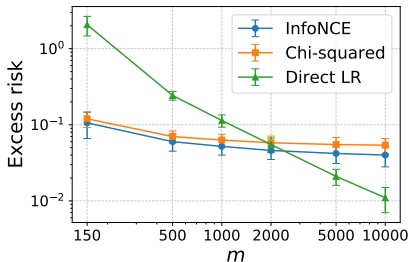

Figure 1: Excess risk for various downstream sample sizes $m$. The errorbars represent the standard deviation over 10 runs.

fit a downstream linear model to predict $\boldsymbol{y}_i$. We define the excess risk of any predictor $h$ as $\mathbb{E}[(\boldsymbol{y}_i - h(\boldsymbol{x}_i))^2] - \sigma^2$, and evaluate the excess risk of the linear model trained on $\widehat{f}(\boldsymbol{x}_i)$. For comparison, we also report the excess risk of a linear model trained directly on the original features $\boldsymbol{x}_i$ (denoted as Direct LR). Results for various downstream sample size $m$ and the standard deviation over 10 runs are shown in Figure 1.

From the figure, we observe that linear regression based on KL (i.e., InfoNCE) or $\chi^2$-pretrained representations achieve comparable excess risks, both much lower than that of direct linear regression when the sample size $m$ is relatively small (e.g., $m = 150, 500$). This suggests that both KL and $\chi^2$-contrastive learning can learn a "good" low-dimensional representation for the downstream task. As the sample size increases, the excess risk of direct linear regression converges to zero, while those of KL and $\chi^2$-pretrained representations converge to non-zero constants. This is consistent with our theoretical results, which attribute the excess risk to the non-zero sufficiency of $\widehat{f}$ and the augmentation error $\epsilon_{\mathcal{G}}$. More results comparing KL (i.e., InfoNCE) and $\chi^2$-contrastive learning in the CLIP setting are provided in Appendix E.

## 6    Conclusion

In this work, we present a new theoretical framework for data augmentation-based contrastive learning, with SimCLR as a representative example. Based on the extended concept of approximate sufficient statistics, we establish a connection between minimizing the f-contrastive losses and minimizing the conditional Bregman sufficiency (CBS) of the encoder. Moreover, we show that the learned encoders can be effectively applied to downstream tasks with performance depending on their sufficiency and the error on the downstream task induced by data augmentation.

Our work opens up many directions for future research. First, as seen in Definition 1, the concept of approximate sufficient statistics is not limited to contrastive learning; exploring its applicability to other self-supervised and supervised learning paradigms is a promising direction. Second, while approximate sufficiency quantifies the information preserved by the encoder, it does not reflect the redundancy in its representation. Thus, it would be interesting to generalize the concept of minimal sufficient statistics and develop practical algorithms for finding representations that are both approximately sufficient and minimal. Lastly, our work mainly focuses on the empirical risk minimizers in contrastive learning. Understanding what representations are learned and how training algorithms influence the learned representation remains another exciting avenue for future research.

## Acknowledgement

We would like to thank the anonymous reviewers for their valuable comments and suggestions. This project was supported by NSF grants DMS-2210827, CCF-2315725, CAREER DMS-2339904, ONR grant N00014-24-S-B001, DARPA AIQ grant HR001124S0029-AIQ-FP-003, an Amazon Research Award, a Google Research Scholar Award, an Okawa Foundation Research Grant, and a Sloan Research Fellowship.

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

# Appendix

## Table of Contents

## A   Related work

**Self-supervised learning and contrastive learning.**   Self-supervised learning (SSL) dates back to the early work of De Sa [7], which leverages cross-modality information as a self-supervised substitute for labels to improve classification performance. In the past decade, SSL has been explored in image classification through various data augmentations, including rotation [9], colorization [50], and Jigsaw puzzles [26]. More recently, contrastive learning based on paired and non-paired samples has emerged as a prominent approach in SSL [14, 5, 10, 18, 31]. Notably, SimCLR [5] learns image representations by minimizing the InfoNCE loss [29] on randomly augmented views of images, while CLIP [31] does so on paired and non-paired image-text samples.

**Choices of the loss function.**   Various loss functions have been used in contrastive learning, including NCE [11], InfoNCE [29], Multi-class N-pair loss [37], SigLIP [49], f-MICL [23]. These losses utilize cross-entropy and its variants to distinguish paired from non-paired samples. Most relevant to our work is the InfoNCE loss [29], derived based on the InfoMax principle [22, 15].

**Theoretical understanding of contrastive learning.**   Thus far, there is a rich body of literature on the theoretical understanding of self-supervised learning [32, 30, 38, 45, 42, 27, 51, 1, 39, 40, 13, 17, 47, 20, 46, 6, 34, 35, 25, 36, 41, 23, 28]. Notably, early works [32, 45, 1] derived generalization error bounds for downstream classification tasks, using linear classifiers trained on representations learned by minimizing the InfoNCE loss. Wang and Isola [45] explained contrastive learning through alignment (pulling paired samples together) and uniformity (separating non-paired samples). Zimmermann et al. [51] showed that InfoNCE minimization can implicitly learn the inverse of the data-generating function. Tosh et al. [39] demonstrated that contrastive learning recovers document representations

that reveal topic posterior information in a document classification problem. More recently, Van Elst and Ghoshdastidar [41] derived new PAC-Bayes bounds on the generalization error of SimCLR using bounded difference concentration and applied them to downstream linear classification. Compared with their results, our generalization error bound in Theorem 1 is independent of the batch size $K$ and thus allows for large or full-batch learning. The most related work to ours is Oko et al. [28], which introduced the concept of approximate sufficiency to assess the quality of representations. They also demonstrated that the learned representation from CLIP [31] can be effectively adapted to several multimodal downstream tasks in a joint hierarchical graphical model.

Our work differs from existing theories of contrastive learning in several aspects: (1) Similar to Oko et al. [28], we derive more refined "excess risk bounds" instead of the "absolute risk bounds" established under structural conditions for downstream tasks in many prior works. (2) We derive novel unified risk bounds for downstream tasks that depend solely on the sufficiency of the encoder and the error induced by data augmentation. (3) We extend the concept of approximate sufficient statistics and theoretically analyze a broader class of contrastive losses.

# B Properties of approximate sufficient statistics

In this section, we discuss some properties of approximate sufficient statistics introduced in Definition 1 and provide some concrete examples.

## B.1 Equivalence in Definition 1

**Lemma 1** (Equivalence of three forms of sufficiency). *The ILS, VFS, CBS definitions in Definition 1 are equivalent, i.e., for any statistic $T$*

$$\mathrm{Suff}_{\mathrm{il,f}}(T) = \mathrm{Suff}_{\mathrm{vf,f}}(T) = \mathrm{Suff}_{\mathrm{cb,f}}(T) =: \mathrm{Suff}_{\mathrm{f}}(T).$$

*Proof of Lemma 1.* (**ILS**) $\Leftrightarrow$ (**VFS**). Note that by the variational form of f-divergence, we have

$$
\begin{aligned}
&- I_{\mathrm{f}}(X, Y) \\
&= \inf_{\mathsf{S}:\mathcal{X}\times\mathcal{Y}\to\mathbb{R}} \mathbb{E}_{\mathbb{P}(x,y)}[-\mathsf{S}(x,y)] + \mathbb{E}_{\mathbb{P}(x)\mathbb{P}(y)}[\mathrm{f}^*(\mathsf{S}(x,y))] \\
&= \inf_{\mathsf{S}_x:\mathcal{X}\to\mathbb{R},\mathsf{S}:\mathcal{X}\times\mathcal{Y}\to\mathbb{R}} \mathbb{E}_{\mathbb{P}(x,y)}[\mathsf{S}_x(x) - \mathsf{S}(x,y)] + \mathbb{E}_{\mathbb{P}(x)\mathbb{P}(y)}[\mathrm{f}^*(\mathsf{S}(x,y) - \mathsf{S}_x)] \\
&= \inf_{\mathsf{S}:\mathcal{X}\times\mathcal{Y}\to\mathbb{R}} \mathbb{E}_{\mathbb{P}(x,y)}[-\mathsf{S}(x,y)] + \inf_{\mathsf{S}_x:\mathcal{X}\mapsto\mathbb{R}} \mathbb{E}_{\mathbb{P}(x)\mathbb{P}(y)}[\mathrm{f}^*(\mathsf{S}(x,y) - \mathsf{S}_x(x)) + \mathsf{S}_x(x)] = \inf_{\mathsf{S}:\mathcal{X}\times\mathcal{Y}\to\mathbb{R}} R_{\mathrm{f}}(\mathsf{S}).
\end{aligned}
$$

Similarly,

$$
\begin{aligned}
&- I_{\mathrm{f}}(T(X), Y) \\
&= \inf_{\mathsf{S}:T(\mathcal{X})\times\mathcal{Y}\to\mathbb{R}} \mathbb{E}_{\mathbb{P}(T(x),y)}[-\mathsf{S}(T(x),y)] + \mathbb{E}_{\mathbb{P}(T(x))\mathbb{P}(y)}[\mathrm{f}^*(\mathsf{S}(T(x),y))] \\
&= \inf_{\mathsf{S}:T(\mathcal{X})\times\mathcal{Y}\to\mathbb{R}} \mathbb{E}_{\mathbb{P}(T(x),y)}[-\mathsf{S}(T(x),y)] + \inf_{\mathsf{S}_x:T(\mathcal{X})\mapsto\mathbb{R}} \mathbb{E}_{\mathbb{P}(T(x))\mathbb{P}(y)}[\mathrm{f}^*(\mathsf{S}(T(x),y) - \mathsf{S}_x(T(x))) + \mathsf{S}_x(T(x))] \\
&= \inf_{\mathsf{S}:T(\mathcal{X})\times\mathcal{Y}\to\mathbb{R}} \mathbb{E}_{\mathbb{P}(x,y)}[-\mathsf{S}(T(x),y)] + \inf_{\mathsf{S}_x:T(\mathcal{X})\mapsto\mathbb{R}} \mathbb{E}_{\mathbb{P}(x)\mathbb{P}(y)}[\mathrm{f}^*(\mathsf{S}(T(x),y) - \mathsf{S}_x(T(x))) + \mathsf{S}_x(T(x))] \\
&= \inf_{\mathsf{S}:T(\mathcal{X})\times\mathcal{Y}\to\mathbb{R}} R_{\mathrm{f}}(\mathsf{S} \circ T).
\end{aligned}
$$

Combining the two results yields the equivalence between (**ILS**) and (**VFS**).

(**ILS**) $\Leftrightarrow$ (**CBS**). By definition of the (**ILS**)

$$
\begin{aligned}
\mathrm{Suff}_{\mathrm{il,f}}(T) &= I_{\mathrm{f}}(X, Y) - I_{\mathrm{f}}(T(X), Y) \\
&= \int \mathrm{f}\Big(\frac{\mathbb{P}(x,y)}{\mathbb{P}(x)\mathbb{P}(y)}\Big)\mathbb{P}(x)\mathbb{P}(y)d\boldsymbol{\mu} - \int \mathrm{f}\Big(\frac{\mathbb{P}(T(x),y)}{\mathbb{P}(T(x))\mathbb{P}(y)}\Big)\mathbb{P}(T(x))\mathbb{P}(y)d\boldsymbol{\mu} \\
&= \int \mathrm{f}\Big(\frac{\mathbb{P}(y|x)}{\mathbb{P}(y)}\Big)\mathbb{P}(x)\mathbb{P}(y)d\boldsymbol{\mu} - \int \mathrm{f}\Big(\frac{\mathbb{P}(y|T(x))}{\mathbb{P}(y)}\Big)\mathbb{P}(x)\mathbb{P}(y)d\boldsymbol{\mu} \\
&= \mathbb{E}_{\mathbb{P}(x)\mathbb{P}(y)}\Big[\mathrm{f}\Big(\frac{\mathbb{P}(y|x)}{\mathbb{P}(y)}\Big) - \mathrm{f}\Big(\frac{\mathbb{P}(y|T(x))}{\mathbb{P}(y)}\Big)\Big] = \mathrm{Suff}_{\mathrm{cb,f}}(T),
\end{aligned}
$$

where the last equality follows since

$$\mathbb{E}_{\mathbb{P}(x)\mathbb{P}(y)}\Big[\mathrm{f}'\Big(\frac{\mathbb{P}(y|T(x))}{\mathbb{P}(y)}\Big)\Big(\frac{\mathbb{P}(y|x)}{\mathbb{P}(y)} - \frac{\mathbb{P}(y|T(x))}{\mathbb{P}(y)}\Big)\Big]$$

$$= \mathbb{E}\Big[\mathbb{E}\Big[\mathrm{f}'\Big(\frac{\mathbb{P}(y|T(x))}{\mathbb{P}(y)}\Big)\Big(\frac{\mathbb{P}(y|x)}{\mathbb{P}(y)} - \frac{\mathbb{P}(y|T(x))}{\mathbb{P}(y)}\Big)\Big|T(x)\Big]\Big]$$

$$= \mathbb{E}\Big[\frac{1}{\mathbb{P}(y)}\mathrm{f}'\Big(\frac{\mathbb{P}(y|T(x))}{\mathbb{P}(y)}\Big)\Big[\mathbb{E}[\mathbb{P}(y|x)|T(x)] - \mathbb{P}(y|T(x))\Big]\Big] = 0. \tag{16}$$

**An equivalent expression of** (**CBS**). We now show that

$$\mathbb{E}_{\mathbb{P}(x)\times\mathbb{P}(y)}\Big[B_{\mathrm{f}}\Big(\frac{\mathbb{P}(y|x)}{\mathbb{P}(y)}, \frac{\mathbb{P}(y|T(x))}{\mathbb{P}(y)}\Big)\Big] = \inf_{\mathbb{Q}:T(\mathcal{X})\mapsto\Delta(\mathcal{Y})} \mathbb{E}_{\mathbb{P}(x)\times\mathbb{P}(y)}\Big[B_{\mathrm{f}}\Big(\frac{\mathbb{P}(y|x)}{\mathbb{P}(y)}, \frac{\mathbb{Q}(y|T(x))}{\mathbb{P}(y)}\Big)\Big].$$

This follows immediately as for any $\mathbb{Q} : T(\mathcal{X}) \mapsto \Delta(\mathcal{Y})$

$$\mathbb{E}_{\mathbb{P}(x)\times\mathbb{P}(y)}\Big[B_{\mathrm{f}}\Big(\frac{\mathbb{P}(y|x)}{\mathbb{P}(y)}, \frac{\mathbb{Q}(y|T(x))}{\mathbb{P}(y)}\Big)\Big] - \mathbb{E}_{\mathbb{P}(x)\times\mathbb{P}(y)}\Big[B_{\mathrm{f}}\Big(\frac{\mathbb{P}(y|x)}{\mathbb{P}(y)}, \frac{\mathbb{P}(y|T(x))}{\mathbb{P}(y)}\Big)\Big]$$

$$= \mathbb{E}_{\mathbb{P}(x)\times\mathbb{P}(y)}\Big[\mathrm{f}\Big(\frac{\mathbb{P}(y|T(x))}{\mathbb{P}(y)}\Big) - \mathrm{f}\Big(\frac{\mathbb{Q}(y|T(x))}{\mathbb{P}(y)}\Big) - \mathrm{f}'\Big(\frac{\mathbb{Q}(y|T(x))}{\mathbb{P}(y)}\Big)\Big(\frac{\mathbb{P}(y|x)}{\mathbb{P}(y)} - \frac{\mathbb{Q}(y|T(x))}{\mathbb{P}(y)}\Big)\Big]$$

$$\geqslant \mathbb{E}_{\mathbb{P}(x)\times\mathbb{P}(y)}\Big[\mathrm{f}'\Big(\frac{\mathbb{Q}(y|T(x))}{\mathbb{P}(y)}\Big)\Big(\frac{\mathbb{P}(y|T(x))}{\mathbb{P}(y)} - \frac{\mathbb{P}(y|x)}{\mathbb{P}(y)}\Big)\Big] = 0,$$

where the first equality uses Eq. (16).

$\square$

## B.2 Properties and examples

**Lemma 2** (Global minimizers of $R_{\mathrm{f}}(\mathsf{S})$). *Recall*

$$R_{\mathrm{f}}(\mathsf{S}) = \mathbb{E}_{\mathbb{P}(x,y)}[-\mathsf{S}(x,y)] + \inf_{\mathsf{S}_x:\mathcal{X}\mapsto\mathbb{R}} \mathbb{E}_{\mathbb{P}(x)\mathbb{P}(y)}[\mathrm{f}^*(\mathsf{S}(x,y) - \mathsf{S}_x(x)) + \mathsf{S}_x(x)].$$

*For* $\mathrm{f}$ *that is strictly convex and differentiable, the following results hold for* $R_{\mathrm{f}}(\cdot)$.

*(1). The infimum in the definition of* $R_{\mathrm{f}}(\cdot)$ *is obtained by* $\mathsf{S}_x(x)$ *such that* $\mathbb{E}_{\mathbb{P}(y)}[(\mathrm{f}')^{-1}(\mathsf{S}(x,y) - \mathsf{S}_x(x))] = 1$ *for all* $x$.

*(2). Let* $\mathsf{S}_\star(x,y) := \mathrm{f}'\big(\frac{\mathbb{P}(x,y)}{\mathbb{P}(x)\mathbb{P}(y)}\big)$. *The global minimizers of* $R_{\mathrm{f}}(\cdot)$ *form the set*

$$\mathcal{M}_{\mathrm{f}} := \Big\{\mathsf{S} : \mathcal{X}\times\mathcal{Y}\mapsto\mathbb{R}, \mathsf{S}(x,y) = \mathsf{S}_\star(x,y) + \mathsf{S}_x(x) \text{ for some } \mathsf{S}_x : \mathcal{X}\mapsto\mathbb{R}\Big\}.$$

*Proof of Lemma 2.* For any fixed $x$, we have

$$\nabla_c \mathbb{E}_{\mathbb{P}(y)}[\mathrm{f}^*(\mathsf{S}(x,y) - c) + c] = \mathbb{E}_{\mathbb{P}(y)}[-\nabla\mathrm{f}^*(\mathsf{S}(x,y) - c) + 1]$$

Claim (1) follows immediately from setting the derivative equal to zero and noting that $\nabla\mathrm{f}^* = (\mathrm{f}')^{-1}$.

To prove claim (2), we first note that adding any function $\mathsf{S}_x(x)$ to $\mathsf{S}(x,y)$ does not change the value of $R_{\mathrm{f}}(\mathsf{S})$ due to the infimum inside the definition of $R_{\mathrm{f}}(\mathsf{S})$. Therefore, it suffices to show that the unique minimizer of

$$\bar{R}_{\mathrm{f}}(\mathsf{S}) := \mathbb{E}_{\mathbb{P}(x,y)}[-\mathsf{S}(x,y)] + \mathbb{E}_{\mathbb{P}(x)\mathbb{P}(y)}[\mathrm{f}^*(\mathsf{S}(x,y))]$$

is $\mathsf{S}_\star = \mathrm{f}'\big(\frac{\mathbb{P}(x,y)}{\mathbb{P}(x)\mathbb{P}(y)}\big)$. Write $\mathsf{S} = \mathsf{S}_\star + ch$. It can be verified that $\bar{R}_{\mathrm{f}}(\mathsf{S}_\star + ch)$ is strictly convex in $c$. Thus $\mathsf{S}_\star$ is the unique minimizer of $\bar{R}_{\mathrm{f}}$ if $\nabla_c \bar{R}_{\mathrm{f}}(\mathsf{S}_\star + ch)|_{c=0} = 0$ for all $h$. This is true since

$$\nabla_c \bar{R}_{\mathrm{f}}(\mathsf{S}_\star + ch)|_{c=0} = \mathbb{E}_{\mathbb{P}(x,y)}[-h(x,y)] + \mathbb{E}_{\mathbb{P}(x)\mathbb{P}(y)}[\nabla\mathrm{f}^*(\mathsf{S}(x,y))h(x,y)]$$

$$= \mathbb{E}_{\mathbb{P}(x,y)}[-h(x,y)] + \mathbb{E}_{\mathbb{P}(x)\mathbb{P}(y)}\Big[\frac{\mathbb{P}(x,y)}{\mathbb{P}(x)\mathbb{P}(y)}h(x,y)\Big] = 0,$$

where the second inequality uses the property of convex conjugates that $\nabla\mathrm{f}^*(\mathrm{f}'(x)) = x$.

$\square$

**Lemma 3** (A general bound on $D_{\mathrm{TV}}(\mathbb{P}(y|x)||\mathbb{P}(y|T(x)))$ based on sufficiency.). *For* f *in Definition 1 that is twice continuously differentiable, and for any statistic $T$, we have*

$$\mathbb{E}_{\mathbb{P}(x)}[D_{\mathrm{TV}}(\mathbb{P}(y|x)||\mathbb{P}(y|T(x)))] \leqslant c_2 \cdot \sqrt{\mathrm{Suff}_{\mathrm{cb,f}}(T)}, \tag{17}$$

*where $c_2 := \left( 2 \inf_{(x,y)\in\mathsf{supp}(x,y)} \mathrm{f}''\left( \frac{\mathbb{P}(y|x)}{\mathbb{P}(y)} \right) \right)^{-1/2}$, and $\mathsf{supp}(x,y)$ denotes the support of $\mathbb{P}(x) \times \mathbb{P}(y)$. Notably, when $\mathrm{f}(x) = (x-1)^2/2$ ($\chi^2$-divergence), we have $c_2 = 1/\sqrt{2}$.*

*Proof of Lemma 3.* Using the CBS form of sufficiency, we find that

$$
\begin{aligned}
\mathrm{Suff}(T) &= \mathbb{E}_{\mathbb{P}(x)\times\mathbb{P}(y)}\left[ B_{\mathrm{f}}\left( \frac{\mathbb{P}(y|x)}{\mathbb{P}(y)}, \frac{\mathbb{P}(y|T(x))}{\mathbb{P}(y)} \right) \right] \\
&\geqslant \frac{1}{2}\mathbb{E}_{\mathbb{P}(x)\times\mathbb{P}(y)}\left[ \mathrm{f}''\left( \frac{\mathbb{P}(y|x)}{\mathbb{P}(y)} \right) \cdot \left[ \frac{\mathbb{P}(y|x)}{\mathbb{P}(y)} - \frac{\mathbb{P}(y|T(x))}{\mathbb{P}(y)} \right]^2 \right] \\
&\geqslant \frac{1}{2} \inf_{(x,y)\in\mathsf{supp}(x,y)} \mathrm{f}''\left( \frac{\mathbb{P}(y|x)}{\mathbb{P}(y)} \right) \cdot \mathbb{E}_{\mathbb{P}(x)\times\mathbb{P}(y)}\left[ \left[ \frac{\mathbb{P}(y|x)}{\mathbb{P}(y)} - \frac{\mathbb{P}(y|T(x))}{\mathbb{P}(y)} \right]^2 \right],
\end{aligned}
$$

where the first inequality follows from the definition of Bregman divergence and the fact that the range of $\mathbb{P}(y|T(x))$ belongs to the range of $\mathbb{P}(y|x)$. Moreover, by Jensen's inequality, we have

$$
\begin{aligned}
\left( \mathbb{E}_{\mathbb{P}(x)\times\mathbb{P}(y)}\left[ \left[ \frac{\mathbb{P}(y|x)}{\mathbb{P}(y)} - \frac{\mathbb{P}(y|T(x))}{\mathbb{P}(y)} \right]^2 \right] \right)^{1/2} &\geqslant \mathbb{E}_{\mathbb{P}(x)\times\mathbb{P}(y)}\left[ \left| \frac{\mathbb{P}(y|x)}{\mathbb{P}(y)} - \frac{\mathbb{P}(y|T(x))}{\mathbb{P}(y)} \right| \right] \\
&= 2\mathbb{E}_{\mathbb{P}(x)}[D_{\mathrm{TV}}(\mathbb{P}(y|x)||\mathbb{P}(y|T(x)))].
\end{aligned}
$$

Putting pieces together yields Lemma 3. $\qquad\square$

**Example 1** (KL-sufficiency). *Take $\mathrm{f}(x) = x \log x$ (KL-divergence), then we have*

$$
\begin{aligned}
\mathrm{Suff}_{\mathrm{cb,f}}(T) &= \mathbb{E}_{\mathbb{P}(x)}\left[ D_{\mathrm{KL}}\left( \mathbb{P}(y|x)||\mathbb{P}(y|T(x)) \right) \right], \quad and \\
R_{\mathrm{f}}(\mathsf{S}) &= \mathbb{E}_{\mathbb{P}(x,y)}[-\mathsf{S}(x,y)] + \mathbb{E}_{\mathbb{P}(x)}[\log \mathbb{E}_{\mathbb{P}(y)}[\exp(\mathsf{S}(x,y))]].
\end{aligned}
$$

*It can be verified that the InfoNCE loss in Eq. (2) is an asymptotically unbiased estimate of $R_{\mathrm{f}}(\mathsf{S})$ as the batch size $K \to \infty$ (see Eq. 6). Moreover, by Pinsker's inequality*

$$\mathbb{E}_{\mathbb{P}(x)}[D_{\mathrm{TV}}(\mathbb{P}(y|x)||\mathbb{P}(y|T(x)))] \leqslant \frac{1}{\sqrt{2}} \cdot \sqrt{\mathrm{Suff}_{\mathrm{cb,kl}}(T)}.$$

**Example 2** (Chi-sufficiency). *Take $\mathrm{f}(x) = (x-1)^2/2$ ($\chi^2$-divergence), then we have*

$$
\begin{aligned}
\mathrm{Suff}_{\mathrm{cb,f}}(T) &= \mathbb{E}_{\mathbb{P}(x)\times\mathbb{P}(y)}\left[ \frac{1}{2}\left( \frac{\mathbb{P}(y|x)}{\mathbb{P}(y)} - \frac{\mathbb{P}(y|T(x))}{\mathbb{P}(y)} \right)^2 \right], \\
R_{\mathrm{f}}(\mathsf{S}) &= \mathbb{E}_{\mathbb{P}(x,y)}[-\mathsf{S}(x,y)] + \mathbb{E}_{\mathbb{P}(x)\mathbb{P}(y)}[(\mathsf{S}(x,y) - \mathbb{E}_{\mathbb{P}(y)}[\mathsf{S}(x,y)])^2/2 + \mathsf{S}(x,y)].
\end{aligned}
$$

*Lemma 3 gives*

$$\mathbb{E}_{\mathbb{P}(x)}[D_{\mathrm{TV}}(\mathbb{P}(y|x)||\mathbb{P}(y|T(x)))] \leqslant \frac{1}{\sqrt{2}}\sqrt{\mathrm{Suff}_{\mathrm{cb},\chi^2}(T)}.$$

*Also, we can bound the $\chi^2$-divergence by the sufficiency:*

$$\mathbb{E}_{\mathbb{P}(x)}\chi^2(\mathbb{P}(y|x)||\mathbb{P}(y|T(x))) \leqslant \mathrm{Suff}_{\mathrm{cb},f}(T) \cdot \left[ 2 \sup_{(x,y)\in\mathsf{supp}(x,y)} \frac{\mathbb{P}(T(x))\mathbb{P}(y)}{\mathbb{P}(T(x),y)} \right].$$

**Example 3** (Squared Hellinger-sufficiency). *Take $\mathrm{f}(x) = 1 - \sqrt{x}$, then we have $\mathrm{f}^*(x) = -1 - \frac{1}{4x}$ for $x < 0$, and*

$$\mathrm{Suff}_{\mathrm{cb,f}}(T) = \mathbb{E}_{\mathbb{P}(x)}\left[ H^2(\mathbb{P}(y)||\mathbb{P}(y|x)) - H^2(\mathbb{P}(y)||\mathbb{P}(y|T(x))) \right],$$

*where $H^2(p||q) := \int (\sqrt{p(x)} - \sqrt{q(x)})^2 \, dx/2$ is the squared Hellinger distance. Similarly, the squared Hellinger distance between $\mathbb{P}(y|x), \mathbb{P}(y|T(x))$ can be bounded by the sufficiency of $T$:*

$$\mathbb{E}_{\mathbb{P}(x)}\left[ H^2(\mathbb{P}(y|x)||\mathbb{P}(y|T(x))) \right] = \frac{1}{2}\mathbb{E}_{\mathbb{P}(x)}\left[ \sum_y \left( \sqrt{\mathbb{P}(y|x)} - \sqrt{\mathbb{P}(y|T(x))} \right)^2 \right]$$

$$\leqslant \left[\sup_{(x,y)\in\text{supp}(x,y)} \sqrt{\frac{\mathbb{P}(T(x),y)}{\mathbb{P}(T(x))\mathbb{P}(y)}}\right] \cdot \mathbb{E}_{\mathbb{P}(x)}\left[\sum_y \sqrt{\mathbb{P}(y)}\frac{\left(\sqrt{\mathbb{P}(y|T(x))}-\sqrt{\mathbb{P}(y|x)}\right)^2}{2\sqrt{\mathbb{P}(y|T(x))}}\right]$$

$$= \left[\sup_{(x,y)\in\text{supp}(x,y)} \sqrt{\frac{\mathbb{P}(T(x),y)}{\mathbb{P}(T(x))\mathbb{P}(y)}}\right] \cdot \text{Suff}_{\text{cb,f}}(T),$$

*where the last equality follows from*

$$\mathbb{E}\left[\sqrt{\mathbb{P}(y|T(x))}-\sqrt{\mathbb{P}(y|x)}\,\middle|\,y,T(x)\right]$$

$$= \mathbb{E}\left[\left(\sqrt{\mathbb{P}(y|T(x))}-\sqrt{\mathbb{P}(y|x)}\right)\cdot\frac{\sqrt{\mathbb{P}(y|T(x))}-\sqrt{\mathbb{P}(y|x)}}{2\sqrt{\mathbb{P}(y|T(x))}}\,\middle|\,y,T(x)\right] + \mathbb{E}\left[\frac{\mathbb{P}(y|T(x))-\mathbb{P}(y|x)}{2\sqrt{\mathbb{P}(y|T(x))}}\,\middle|\,y,T(x)\right]$$

$$= \mathbb{E}\left[\frac{\left(\sqrt{\mathbb{P}(y|T(x))}-\sqrt{\mathbb{P}(y|x)}\right)^2}{2\sqrt{\mathbb{P}(y|T(x))}}\,\middle|\,y,T(x)\right].$$

### B.3 Sufficiency of similarity scores

The definition of approximate sufficiency can be extended to score functions $\mathsf{S} : \mathcal{X} \times \mathcal{Y} \mapsto \mathbb{R}$ that measure the similarity between $(X, Y)$.

**Definition 2** (Approximate sufficient score functions). *Let $\mathsf{S} : \mathcal{X} \times \mathcal{Y} \mapsto \mathbb{R}$ be a similarity score function. It induces a conditional density $\mathbb{P}_\mathsf{S}$ on $\mathcal{X} \times \mathcal{Y}$ w.r.t. the base measure $\mu$ via*

$$\mathbb{P}_\mathsf{S}(y|x) = \mathbb{P}(y)(\text{f}')^{-1}(\bar{\mathsf{S}}(x,y)),$$

*where $\bar{\mathsf{S}}(x,y) = \mathsf{S}(x,y) - \mathsf{S}_x(x)$ such that $\mathbb{E}_{\mathbb{P}(y)}[(\text{f}')^{-1}\bar{\mathsf{S}}(x,y)] = 1$ for all $x$. We define the sufficiency of $\mathsf{S}$ in two equivalent forms:*

- **Variational Form Sufficiency (VFS):** *The variational form sufficiency of $T$ is given by*

$$\text{Suff}_{\text{vf,f}}(\mathsf{S}) = R_\text{f}(\mathsf{S}) - \inf_{\widetilde{\mathsf{S}}:\mathcal{X}\times\mathcal{Y}\mapsto\mathbb{R}} R_\text{f}(\widetilde{\mathsf{S}}),$$

  *and the $\text{f}$-contrastive loss*

$$R_\text{f}(\mathsf{S}) := \mathbb{E}_{\mathbb{P}(x,y)}[-\mathsf{S}(x,y)] + \inf_{\mathsf{S}_x:\mathcal{X}\mapsto\mathbb{R}} \mathbb{E}_{\mathbb{P}(x)\mathbb{P}(y)}[\text{f}^*(\mathsf{S}(x,y)-\mathsf{S}_x(x))+\mathsf{S}_x(x)], \quad (18)$$

  *where $\text{f}^*$ is the Fenchel-dual of $\text{f}$.*

- **Conditional Bregman Sufficiency (CBS):** *The conditional Bregman sufficiency of $T$ is defined as*

$$\text{Suff}_{\text{cb,f}}(\mathsf{S}) = \mathbb{E}_{\mathbb{P}(x)\times\mathbb{P}(y)}\left[B_\text{f}\left(\frac{\mathbb{P}(y|x)}{\mathbb{P}(y)},\frac{\mathbb{P}_\mathsf{S}(y|x)}{\mathbb{P}(y)}\right)\right],$$

  *where $B_\text{f}(a,b) := \text{f}(a) - \text{f}(b) - (a-b)\text{f}'(b)$ is the Bregman divergence of $\text{f}$.*

Note that the excess risk of the contrastive loss equals the sufficiency of $\mathsf{S}$ under our definition. Similar to Definition 1, we have

**Lemma 4** (Equivalence of two forms of score sufficiency). *For any similarity score $\mathsf{S} : \mathcal{X} \times \mathcal{Y} \mapsto \mathbb{R}$, the three forms of sufficiency in Definition 2 are equivalent, i.e.,*

$$\text{Suff}_{\text{vf,f}}(\mathsf{S}) = \text{Suff}_{\text{cb,f}}(\mathsf{S}) =: \text{Suff}_\text{f}(\mathsf{S}).$$

*Proof of Lemma 4.* **(VFS)** $\Leftrightarrow$ **(CBS).** Let $\mathsf{S}_\star(x,y) = \text{f}'(\frac{\mathbb{P}(x,y)}{\mathbb{P}(x)\mathbb{P}(y)})$. We have by Lemma 2 that $\mathsf{S}_\star \in \arg\min_{\widetilde{\mathsf{S}}} R_\text{f}(\widetilde{\mathsf{S}})$. By the definition of the **(VFS)**, we have

$$\text{Suff}_{\text{vf,f}}(\mathsf{S}) = R_\text{f}(\mathsf{S}) - R_\text{f}(\mathsf{S}_\star)$$

$$= \mathbb{E}_{\mathbb{P}(x,y)}[\mathsf{S}_\star - \bar{\mathsf{S}}(x,y)] + \mathbb{E}_{\mathbb{P}(x)\mathbb{P}(y)}[\text{f}^*(\bar{\mathsf{S}}(x,y))-\text{f}^*(\mathsf{S}_\star(x,y))]$$

$$\overset{(i)}{=} \mathbb{E}_{\mathbb{P}(x,y)}[\mathsf{S}_\star - \bar{\mathsf{S}}(x,y)] + \mathbb{E}_{\mathbb{P}(x)\mathbb{P}(y)}\left[\text{f}\left(\frac{\mathbb{P}(x,y)}{\mathbb{P}(x)\mathbb{P}(y)}\right) - \frac{\mathbb{P}(x,y)}{\mathbb{P}(x)\mathbb{P}(y)}\mathsf{S}_\star(x,y)\right]$$

$$= -\mathbb{E}_{\mathbb{P}(x,y)}[\bar{\mathsf{S}}(x,y)] + \mathbb{E}_{\mathbb{P}(x)\mathbb{P}(y)}[\mathsf{f}^*(\bar{\mathsf{S}}(x,y))] + \mathbb{E}_{\mathbb{P}(x)\mathbb{P}(y)}\left[\mathsf{f}\left(\frac{\mathbb{P}(x,y)}{\mathbb{P}(x)\mathbb{P}(y)}\right)\right]$$

$$\overset{(ii)}{=} -\mathbb{E}_{\mathbb{P}(x,y)}[\bar{\mathsf{S}}(x,y)] + \mathbb{E}_{\mathbb{P}(x)\mathbb{P}(y)}\left[\mathsf{f}\left(\frac{\mathbb{P}(x,y)}{\mathbb{P}(x)\mathbb{P}(y)}\right) + \frac{\mathbb{P}_{\mathsf{S}}(y|x)}{\mathbb{P}(y)}\bar{\mathsf{S}}(x,y) - \mathsf{f}\left(\frac{\mathbb{P}_{\mathsf{S}}(y|x)}{\mathbb{P}(y)}\right)\right]$$

$$= \mathbb{E}_{\mathbb{P}(x)\mathbb{P}(y)}\left[\mathsf{f}\left(\frac{\mathbb{P}(y|x)}{\mathbb{P}(y)}\right) - \mathsf{f}\left(\frac{\mathbb{P}_{\mathsf{S}}(y|x)}{\mathbb{P}(y)}\right)\right] - \mathbb{E}_{\mathbb{P}(x)\times\mathbb{P}(y)}\left[\bar{\mathsf{S}}(x,y)\left(\frac{\mathbb{P}(y|x)}{\mathbb{P}(y)} - \frac{\mathbb{P}_{\mathsf{S}}(y|x)}{\mathbb{P}(y)}\right)\right],$$

where step (i) and (ii) use $f((f')^{-1}(z)) + f^*(z) = z(f')^{-1}(z)$ with $z = \mathsf{S}_\star(x,y)$ and $\bar{\mathsf{S}}(x,y)$, respectively. Since $\bar{\mathsf{S}}(x,y) = \mathsf{f}'(\frac{\mathbb{P}_{\mathsf{S}}(y|x)}{\mathbb{P}(y)})$, it follows immediately that $\mathrm{Suff}_{\mathrm{vf},\mathsf{f}}(\mathsf{S}) = \mathrm{Suff}_{\mathrm{cb},\mathsf{f}}(\mathsf{S})$.

$\square$

**Example 4.** *Take* $f(x) = x\log x$ *(KL-divergence). Then* $\mathsf{S}_\star(x,y) = \log\left(\mathbb{P}(x,y)/[\mathbb{P}(x)\mathbb{P}(y)]\right)$, $B_{\mathrm{f}}(a,b) = a\log(a/b) - (a-b)$, *and* $\mathbb{P}_{\mathsf{S}}(y|x) = \mathbb{P}(y)\exp(\mathsf{S}(x,y))/\mathbb{E}_{\mathbb{P}(y)}[\exp(\mathsf{S}(x,y))]$. *Also, we have*

$$\mathrm{Suff}_{\mathrm{kl}}(\mathsf{S}) = R_{\mathrm{f}}(\mathsf{S}) - R_{\mathrm{f}}(\mathsf{S}_\star) = \int \mathbb{P}(y|x)\log\left(\frac{\mathbb{P}(y|x)}{\mathbb{P}_{\mathsf{S}}(y|x)}\right) - \left(\mathbb{P}(y|x) - \mathbb{P}_{\mathsf{S}}(y|x)\right)\mathbb{P}(x)dy\,dx$$

$$= \mathbb{E}_{x\sim\mathbb{P}(x)}\left[\mathsf{D}_{\mathrm{KL}}(\mathbb{P}(y|x)\,\|\,\mathbb{P}_{\mathsf{S}}(y|x))\right].$$

**Example 5.** *Take* $f(x) = (x-1)^2/2$ *($\chi^2$-divergence). Then* $\mathsf{S}_\star(x,y) = \mathbb{P}(x,y)/[\mathbb{P}(x)\mathbb{P}(y)] - 1$, $B_{\mathrm{f}}(a,b) = (a-b)^2/2$, *and* $\mathbb{P}_{\mathsf{S}}(y|x) = \mathbb{P}(y)\left(\mathsf{S}(x,y) - \mathbb{E}_y[\mathsf{S}(x,y)] + 1\right)$. *Moreover,*

$$\mathrm{Suff}_{\chi^2}(\mathsf{S}) = R_{\mathrm{f}}(\mathsf{S}) - R_{\mathrm{f}}(\mathsf{S}_\star) = \frac{1}{2}\mathbb{E}_{\mathbb{P}(x)\times\mathbb{P}(y)}\left[\frac{\left(\mathbb{P}(y|x) - \mathbb{P}_{\mathsf{S}}(y|x)\right)^2}{\mathbb{P}(y)^2}\right]$$

$$= \frac{1}{2}\mathbb{E}_{\mathbb{P}(x)}\sum_y\left[\frac{\left(\mathbb{P}(y|x) - \mathbb{P}_{\mathsf{S}}(y|x)\right)^2}{\mathbb{P}(y|x)} \cdot \frac{\mathbb{P}(y|x)}{\mathbb{P}(y)}\right]$$

$$\geqslant \inf_{(x,y)\in\mathsf{supp}(x,y)}\frac{\mathbb{P}(x,y)}{\mathbb{P}(x)\mathbb{P}(y)} \cdot \mathbb{E}_{\mathbb{P}(x)}\left[\chi^2(\mathbb{P}(y|x)||\mathbb{P}_{\mathsf{S}}(y|x))\right].$$

## C  Proofs in Section 3

### C.1  Proof of Eq. (6)

As given in Example 1 (which can be established using Lemma 2), the KL-contrastive loss has the form

$$R_{\mathrm{kl}}(\mathsf{S}) = \mathbb{E}_{(\boldsymbol{z}^{(1)},\boldsymbol{z}^{(2)})}[-\mathsf{S}(\boldsymbol{z}^{(1)},\boldsymbol{z}^{(2)})] + \mathbb{E}_{\boldsymbol{z}^{(1)}\sim\mathbb{P}_{\boldsymbol{z}}}[\log\mathbb{E}_{\boldsymbol{z}^{(2)}\sim\mathbb{P}_{\boldsymbol{z}}}[\exp(\mathsf{S}(\boldsymbol{z}^{(1)},\boldsymbol{z}^{(2)}))]].$$

Recall the SimCLR loss $\bar{\mathsf{R}}_{\mathrm{simclr},K}(\mathsf{S})$ in Eq. (2). We then have

$$\lim_{K\to\infty}\bar{\mathsf{R}}_{\mathrm{simclr},K}(\mathsf{S}) - \log K$$

$$= \frac{1}{2}\lim_{K\to\infty}\mathbb{E}\left[-\log\frac{\exp(\mathsf{S}(\boldsymbol{z}_1^{(1)},\boldsymbol{z}_1^{(2)}))}{\sum_{j\in[K]}\exp(\mathsf{S}(\boldsymbol{z}_1^{(1)},\boldsymbol{z}_j^{(2)}))/K}\right] + \frac{1}{2}\mathbb{E}\left[-\log\frac{\exp(\mathsf{S}(\boldsymbol{z}_1^{(1)},\boldsymbol{z}_1^{(2)}))}{\sum_{j\in[K]}\exp(\mathsf{S}(\boldsymbol{z}_j^{(1)},\boldsymbol{z}_1^{(2)}))/K}\right]$$

$$= \lim_{K\to\infty}\mathbb{E}\left[\log\sum_{j\in[K]}\exp(\mathsf{S}(\boldsymbol{z}_1^{(1)},\boldsymbol{z}_j^{(2)}))/K\right] - \mathbb{E}[\exp(\mathsf{S}(\boldsymbol{z}_1^{(1)},\boldsymbol{z}_1^{(2)}))] = R_{\mathrm{kl}}(\mathsf{S}),$$

where the second equality follows from the symmetry of $\mathsf{S}$ in its arguments and the last equality uses the law of large numbers (note that $\boldsymbol{z}_1^{(1)}$ is independent of $\boldsymbol{z}_j^{(2)}$ for $j \neq 1$) and the bounded convergence theorem.

### C.2  Proof of Theorem 1

We begin the proof by stating the following proposition that connects the excess risk with sufficiency.

**Proposition 8** (Near-minimizers of SimCLR as near-sufficient statistics; Proposition 1 in Oko et al. [28])**.** *Suppose Assumption 1 holds and $S_\star$ is a global minimizer of $\overline{R}_{\mathsf{simclr},K}(S)$ as defined in Section 3.1. Then, there exists a constant $C > 0$, which depends polynomially on $B_S$, such that for any function $f \in \mathcal{F}$, its sufficiency can be bounded by its SimCLR excess risk. Namely, for any $K \geqslant 2$, we have*

$$\mathrm{Suff}(f) \leqslant \lim_{K' \to \infty} \left[ \overline{R}_{\mathsf{simclr},K'}(S_f) - \overline{R}_{\mathsf{simclr},K'}(S_\star) \right] \leqslant \underbrace{\left[ \overline{R}_{\mathsf{simclr},K}(S_f) - \overline{R}_{\mathsf{simclr},K}(S_\star) \right]}_{\text{SimCLR excess risk}} \cdot \left( 1 + \frac{C}{K} \right).$$

A similar version of this result has been established for contrastive language-image pretraining (CLIP) in Proposition 1 in Oko et al. [28]. The proof of Proposition 8 follows immediately from the proof of Proposition 1 in Oko et al. [28] as the SimCLR setup can be viewed as a special case of CLIP in which the text and image follows a symmetric distribution conditioned on their shared information.

Adopt the shorthand notation $\overline{R}_K$ for $\overline{R}_{\mathsf{simclr},K}$. With Proposition 8 at hand, we obtain the following decomposition for some $C > 0$ polynomially dependent on $B_S$

$$
\begin{aligned}
\mathrm{Suff}(\widehat{f}) &\leqslant \left[ \overline{R}_K(S_{\widehat{f}}) - \overline{R}_K(S_\star) \right] \cdot \left( 1 + \frac{C}{K} \right) \\
&= \left[ [\overline{R}_K(S_{\widehat{f}}) - \inf_{f \in \mathcal{F}} \overline{R}_K(S_f)] + [\inf_{f \in \mathcal{F}} \overline{R}_K(S_f) - \overline{R}_K(S_\star)] \right] \cdot \left( 1 + \frac{C}{K} \right) \\
&\leqslant \underbrace{\left[ \overline{R}_K(S_{\widehat{f}}) - \inf_{f \in \mathcal{F}} \overline{R}_K(S_f) \right] \cdot \left( 1 + \frac{C}{K} \right)}_{\text{generalization error}} + \underbrace{\left[ \inf_{f \in \mathcal{F}} \overline{R}_K(S_f) - \overline{R}_K(S_\star) \right] \cdot \left( 1 + \frac{C}{K} \right)}_{\text{approximation error}}.
\end{aligned}
$$

Therefore, it remains to prove the following bound.

(1). With probability at least $1 - \delta$, the excess risk

$$\overline{R}_K(S_{\widehat{f}}) - \inf_{f \in \mathcal{F}} \overline{R}_K(S_f) \leqslant \frac{C}{\sqrt{n}} \left[ \sqrt{\log(1/\delta)} + B_\tau^2 \int_0^{2(\log B_S + B_\tau)} \sqrt{\log \mathcal{N}(u, \|\cdot\|_{2,\infty}, \mathcal{F})} ] du \right] \tag{19}$$

for some constant $C > 0$ that is polynomially dependent on $B_S$.

Proof of Eq. (19). Recall the definition of $\widehat{R}_{\mathsf{simclr},K}$ in Eq. (3) and adopt the shorthand $\widehat{R}_K$ for $\widehat{R}_{\mathsf{simclr},K}$. Let $B_f := \sqrt{B_\tau(\log B_S + B_\tau)}$, $B := c(B_S^6 + 1)B_f B_\tau$ for some absolute constant $c > 0$. It can be verified by Assumption 2 that $\mathcal{F}$ must satisfy $\|f\|_{2,\infty} \leqslant B_f$ for all $f \in \mathcal{F}$ to ensure Assumption 1 holds. Define the zero-mean random process $X_f := \widehat{R}_K(S_f) - \mathbb{E}[\widehat{R}_K(S_f)]$, $f \in \mathcal{F}$. We will show that

$$\mathbb{P}\left( \left| \sup_{f \in \mathcal{F}} |X_f| - \mathbb{E}[\sup_{f \in \mathcal{F}} |X_f|] \right| \geqslant t \right) \leqslant 2 \exp\left( -\frac{2nt^2}{9B_S^4} \right), \quad \text{for all } t \geqslant 0, \quad \text{and} \tag{20a}$$

$$\mathbb{E}[\sup_{f \in \mathcal{F}} |X_f|] \leqslant \mathbb{E}[|X_{f_0}|] + \mathbb{E}[\sup_{f, \widetilde{f} \in \mathcal{F}} |X_f - X_{\widetilde{f}}|]$$

$$\leqslant c \frac{B_S^2}{\sqrt{n}} + 32 \frac{B}{\sqrt{n}} \cdot \int_0^{2B_f} \sqrt{\log \mathcal{N}(u, \|\cdot\|_{2,\infty}, \mathcal{F})} du \tag{20b}$$

for any $f_0 \in \mathcal{F}$ and some absolute constant $c > 0$. Combining the two bounds and noting

$$\overline{R}_K(S_{\widehat{f}}) - \inf_{f \in \mathcal{F}} \overline{R}_K(S_f) \leqslant 2 \sup_{f \in \mathcal{F}} |\widehat{R}_K(S_f) - \overline{R}_K(S_f)| = 2 \sup_{f \in \mathcal{F}} |\widehat{R}_K(S_f) - \mathbb{E}[\widehat{R}_K(S_f)]| = 2 \sup_{f \in \mathcal{F}} |X_f| \tag{21}$$

yields claim (1).

**Proof of Eq.** (20a). Let $\bar{z}_i = (z_i^{(1)}, z_i^{(2)})$. Then $\{\bar{z}_i\}_{i=1}^n$ are i.i.d. pairs of augmented views. For any $i \in [n_1], j \in [K]$, suppose $\bar{z}_{(i-1)K+j}$ is replaced by some alternative sample $\tilde{z}_{(i-1)K+j} = (\tilde{z}_{(i-1)K+j}^{(1)}, \tilde{z}_{(i-1)K+j}^{(2)})$ in the calculation of $\hat{\mathsf{R}}_K(\mathsf{S}_f)$. Then we have

$$|X_f(\bar{z}_1, \ldots, \bar{z}_{(i-1)K+j}, \ldots, \bar{z}_n) - X_f(\bar{z}_1, \ldots, \tilde{z}_{(i-1)K+j}, \ldots, \bar{z}_n)|$$
$$= |\hat{\mathsf{R}}_K(\mathsf{S}_f)(\bar{z}_1, \ldots, \bar{z}_{(i-1)K+j}, \ldots, \bar{z}_n) - \hat{\mathsf{R}}_K(\mathsf{S}_f)(\bar{z}_1, \ldots, \tilde{z}_{(i-1)K+j}, \ldots, \bar{z}_n)| \leqslant U_1 + U_2,$$

$$(22)$$

where (assuming $\tilde{z}_s = \bar{z}_s$ for $j \in [n]\backslash\{(i-1)K + j\}$)

$$U_1 := \frac{1}{n}\left|\mathsf{S}_f(z_{(i-1)K+j}^{(1)}, z_{(i-1)K+j}^{(2)}) - \mathsf{S}_f(\tilde{z}_{(i-1)K+j}^{(1)}, \tilde{z}_{(i-1)K+j}^{(2)})\right| \leqslant \frac{2\log B_\mathsf{S}}{n},$$

and

$$U_2 := \frac{1}{2n}\sum_{k=1}^K \left|\left[\log\left(\frac{1}{K}\sum_{l\in[K]}\exp(\mathsf{S}_f(z_{(i-1)K+k}^{(1)}, z_{(i-1)K+l}^{(2)}))\right) + \log\left(\frac{1}{K}\sum_{l\in[K]}\exp(\mathsf{S}_f(z_{(i-1)K+l}^{(1)}, z_{(i-1)K+k}^{(2)}))\right)\right]\right.$$

$$\left. - \left[\log\left(\frac{1}{K}\sum_{l\in[K]}\exp(\mathsf{S}_f(\tilde{z}_{(i-1)K+k}^{(1)}, \tilde{z}_{(i-1)K+l}^{(2)}))\right) + \log\left(\frac{1}{K}\sum_{l\in[K]}\exp(\mathsf{S}_f(\tilde{z}_{(i-1)K+l}^{(1)}, \tilde{z}_{(i-1)K+k}^{(2)}))\right)\right]\right|$$

$$\overset{(i)}{\leqslant} \frac{B_\mathsf{S}}{2n}\sum_{k=1}^K \left|\frac{1}{K}\left|\sum_{l\in[K]}\exp(\mathsf{S}_f(z_{(i-1)K+k}^{(1)}, z_{(i-1)K+l}^{(2)})) - \sum_{l\in[K]}\exp(\mathsf{S}_f(\tilde{z}_{(i-1)K+k}^{(1)}, \tilde{z}_{(i-1)K+l}^{(2)}))\right|\right.$$

$$\left. + \frac{1}{K}\left|\sum_{l\in[K]}\exp(\mathsf{S}_f(z_{(i-1)K+l}^{(1)}, z_{(i-1)K+k}^{(2)})) - \sum_{l\in[K]}\exp(\mathsf{S}_f(\tilde{z}_{(i-1)K+l}^{(1)}, \tilde{z}_{(i-1)K+k}^{(2)}))\right|\right|$$

$$\leqslant \frac{B_\mathsf{S}}{nK}\sum_{k=1}^K\sum_{l=1}^K\left|\exp(\mathsf{S}_f(z_{(i-1)K+k}^{(1)}, z_{(i-1)K+l}^{(2)})) - \exp(\mathsf{S}_f(\tilde{z}_{(i-1)K+k}^{(1)}, \tilde{z}_{(i-1)K+l}^{(2)}))\right|$$

$$\overset{(ii)}{\leqslant} \frac{2(B_\mathsf{S}^2 - 1)}{n},$$

Here, step (i) follows from the triangle inequality, a Taylor expansion of $\log(x)$, and Assumption 1; step (ii) follows from Assumption 1 and noting that $\left|\exp(\mathsf{S}_f(z_{(i-1)K+k}^{(1)}, z_{(i-1)K+l}^{(2)})) - \exp(\mathsf{S}_f(\tilde{z}_{(i-1)K+k}^{(1)}, \tilde{z}_{(i-1)K+l}^{(2)}))\right| \neq 0$ for at most $2K$ terms with indices $k, l \in [K]$.

Putting pieces together, we find

$$|\hat{\mathsf{R}}_K(\mathsf{S}_f)(\bar{z}_1, \ldots, \bar{z}_{(i-1)K+j}, \ldots, \bar{z}_n) - \hat{\mathsf{R}}_K(\mathsf{S}_f)(\bar{z}_1, \ldots, \tilde{z}_{(i-1)K+j}, \ldots, \bar{z}_n)|$$
$$\leqslant \frac{2\log B_\mathsf{S} + 2B_\mathsf{S}^2 - 2}{n} \leqslant \frac{3B_\mathsf{S}^2}{n}$$

for any $\tilde{z}_{(i-1)K+j}$ and any $i \in [n_1], j \in [K]$ and all $f \in \mathcal{F}$. Therefore, Eq. (20a) follows from Corollary 2.21 in [44] for functions with bounded differences.

**Proof of Eq.** (20b). First, we have $\mathbb{E}[|X_{f_0}|] \leqslant cB_\mathsf{S}^2/\sqrt{n}$ by properties of sub-Gaussian variables and the fact that, for any $f_0 \in \mathcal{F}$, $X_{f_0}$ is zero-mean with bounded differences $cB_\mathsf{S}^2/n$, as implied by the proof of Eq. (20a). By Dudley's entropy integral bound (see Theorem 5.22 in [44]), it suffices to show $\{X_f, f \in \mathcal{F}\}$ is a zero-mean sub-Gaussian process with respect to the metric $\rho_X(f, \tilde{f}) := B\|f - \tilde{f}\|_{2,\infty}/\sqrt{n}$.

Let $\|x\|_\psi := \inf\{t > 0 : \mathbb{E}[\psi(x/t)] \leqslant 1\}$ denote the Orlicz norm for random variables and let $\psi_2(u) = \exp(u^2) - 1$. We have

$$\|X_f - X_{\tilde{f}}\|_{\psi_2} = \|\hat{\mathsf{R}}_K(\mathsf{S}_f) - \hat{\mathsf{R}}_K(\mathsf{S}_{\tilde{f}}) - \mathbb{E}[\hat{\mathsf{R}}_K(\mathsf{S}_f) - \hat{\mathsf{R}}_K(\mathsf{S}_{\tilde{f}})]\|_{\psi_2} \leqslant c(\|U_3 - \mathbb{E}[U_3]\|_{\psi_2} + \|U_4 - \mathbb{E}[U_4]\|_{\psi_2})$$

$$(23)$$

for some absolute constant $c > 0$ (we allow the value of $c$ to vary from place to place), where

$$U_3 := \frac{1}{n} \sum_{i=1}^{n_1} \sum_{j=1}^{K} \left[ \mathsf{S}_f(z_{(i-1)K+j}^{(1)}, z_{(i-1)K+j}^{(2)}) - \mathsf{S}_{\widetilde{f}}(z_{(i-1)K+j}^{(1)}, z_{(i-1)K+j}^{(2)}) \right],$$

$$U_4 := \frac{1}{2n} \sum_{i=1}^{n_1} \sum_{j=1}^{K} \left[ \left[ \log \left( \frac{1}{K} \sum_{l \in [K]} \exp(\mathsf{S}_f(z_{(i-1)K+j}^{(1)}, z_{(i-1)K+l}^{(2)})) \right) + \log \left( \frac{1}{K} \sum_{l \in [K]} \exp(\mathsf{S}_f(z_{(i-1)K+l}^{(1)}, z_{(i-1)K+j}^{(2)})) \right) \right] \right.$$
$$\left. - \left[ \log \left( \frac{1}{K} \sum_{l \in [K]} \exp(\mathsf{S}_{\widetilde{f}}(z_{(i-1)K+j}^{(1)}, z_{(i-1)K+l}^{(2)})) \right) + \log \left( \frac{1}{K} \sum_{l \in [K]} \exp(\mathsf{S}_{\widetilde{f}}(z_{(i-1)K+l}^{(1)}, z_{(i-1)K+j}^{(2)})) \right) \right] \right].$$

It remains to show both $U_3 - \mathbb{E}[U_3]$ and $U_4 - \mathbb{E}[U_4]$ are $\rho_X(f, \widetilde{f})$ sub-Gaussian.

Notice that for any $z^{(1)}, z^{(2)} \in \mathcal{X}, f, \widetilde{f} \in \mathcal{F}$, by Assumption 2, we have

$$|\mathsf{S}_f(z^{(1)}, z^{(2)}) - \mathsf{S}_{\widetilde{f}}(z^{(1)}, z^{(2)})| \leqslant B_\tau \cdot |\langle f(z^{(1)}), f(z^{(2)}) \rangle - \langle \widetilde{f}(z^{(1)}), \widetilde{f}(z^{(2)}) \rangle|$$
$$\leqslant B_\tau(\|f(z^{(2)})\|_2 \cdot \|f - \widetilde{f}\|_{2,\infty} + \|\widetilde{f}(z^{(1)})\|_2 \cdot \|f - \widetilde{f}\|_{2,\infty})$$
$$\overset{(i)}{\leqslant} 2 B_f B_\tau \|f - \widetilde{f}\|_{2,\infty}, \tag{24}$$

where step (i) uses $\mathsf{S}_f(z, z) = \|f(z)\|_2^2 \leqslant B_f^2$ for $z \in \mathcal{X}$. Since $\bar{z}_i = (z_i^{(1)}, z_i^{(2)}), i \in [n]$ are i.i.d., it follows immediately that $U_3 - \mathbb{E}[U_3]$ is $2 B_f B_\tau \|f - \widetilde{f}\|_{2,\infty}/\sqrt{n}$-sub-Gaussian, i.e.,

$$\|U_3 - \mathbb{E}[U_3]\|_{\psi_2} \leqslant \frac{c B_f B_\tau}{\sqrt{n}} \|f - \widetilde{f}\|_{2,\infty}. \tag{25}$$

Recall the definition of $\{\bar{z}_s, \widetilde{z}_s\}_{s=1}^{n}$ in the proof of Eq. (20a). To bound $\|U_4\|_{\psi_2}$, we start with introducing the shorthands for any fixed indices $i \in [n_1], j \in [K]$

$$\mathcal{U}_k(\bar{z}) := \frac{1}{K} \sum_{l \in [K]} \exp(\mathsf{S}_f(z_{(i-1)K+k}^{(1)}, z_{(i-1)K+l}^{(2)})), \quad \mathcal{V}_k(\bar{z}) := \frac{1}{K} \sum_{l \in [K]} \exp(\mathsf{S}_f(z_{(i-1)K+l}^{(1)}, z_{(i-1)K+k}^{(2)})),$$

$$\widetilde{\mathcal{U}}_k(\bar{z}) := \frac{1}{K} \sum_{l \in [K]} \exp(\mathsf{S}_{\widetilde{f}}(z_{(i-1)K+k}^{(1)}, z_{(i-1)K+l}^{(2)})), \quad \widetilde{\mathcal{V}}_k(\bar{z}) := \frac{1}{K} \sum_{l \in [K]} \exp(\mathsf{S}_{\widetilde{f}}(z_{(i-1)K+l}^{(1)}, z_{(i-1)K+k}^{(2)}))$$

for all $k \in [K]$. Similar to the proof of Eq. (20a), for any given index $(i-1)K + j$, we have

$$|U_4(\bar{z}_1, \ldots, \bar{z}_{(i-1)K+j}, \ldots, \bar{z}_n) - U_4(\bar{z}_1, \ldots, \widetilde{z}_{(i-1)K+j}, \ldots, \bar{z}_n)|$$

$$= \left| \frac{1}{2n} \sum_{k=1}^{K} \left[ \log \left( \frac{\mathcal{U}_k(\bar{z})}{\widetilde{\mathcal{U}}_k(\bar{z})} \right) + \log \left( \frac{\mathcal{V}_k(\bar{z})}{\widetilde{\mathcal{V}}_k(\bar{z})} \right) - \log \left( \frac{\mathcal{U}_k(\widetilde{z})}{\widetilde{\mathcal{U}}_k(\widetilde{z})} \right) - \log \left( \frac{\mathcal{V}_k(\widetilde{z})}{\widetilde{\mathcal{V}}_k(\widetilde{z})} \right) \right] \right|$$

$$\leqslant \frac{B_\mathsf{S}^2}{2n} \sum_{k=1}^{K} \left[ \left| \frac{\mathcal{U}_k(\bar{z})}{\widetilde{\mathcal{U}}_k(\bar{z})} - \frac{\mathcal{U}_k(\widetilde{z})}{\widetilde{\mathcal{U}}_k(\widetilde{z})} \right| + \left| \frac{\mathcal{V}_k(\bar{z})}{\widetilde{\mathcal{V}}_k(\bar{z})} - \frac{\mathcal{V}_k(\widetilde{z})}{\widetilde{\mathcal{V}}_k(\widetilde{z})} \right| \right],$$

where the last line follows from Assumption 1 and a Taylor expansion of $\log(x)$. Moreover,

$$\sum_{k=1}^{K} \left| \frac{\mathcal{U}_k(\bar{z})}{\widetilde{\mathcal{U}}_k(\bar{z})} - \frac{\mathcal{U}_k(\widetilde{z})}{\widetilde{\mathcal{U}}_k(\widetilde{z})} \right| = \sum_{k=1}^{K} \left| \frac{\mathcal{U}_k(\bar{z}) - \widetilde{\mathcal{U}}_k(\bar{z})}{\widetilde{\mathcal{U}}_k(\bar{z})} - \frac{\mathcal{U}_k(\widetilde{z}) - \widetilde{\mathcal{U}}_k(\widetilde{z})}{\widetilde{\mathcal{U}}_k(\widetilde{z})} \right|$$

$$\overset{(ii)}{\leqslant} B_\mathsf{S}^2 \sum_{k=1}^{K} |(\mathcal{U}_k(\bar{z}) - \widetilde{\mathcal{U}}_k(\bar{z}))\widetilde{\mathcal{U}}_k(\widetilde{z}) - (\mathcal{U}_k(\widetilde{z}) - \widetilde{\mathcal{U}}_k(\widetilde{z}))\widetilde{\mathcal{U}}_k(\bar{z})|$$

$$\leqslant B_\mathsf{S}^2 \sum_{k=1}^{K} \left[ |((\mathcal{U}_k - \widetilde{\mathcal{U}}_k)(\bar{z}) - (\mathcal{U}_k - \widetilde{\mathcal{U}}_k)(\widetilde{z}))\widetilde{\mathcal{U}}_k(\widetilde{z})| + |(\mathcal{U}_k - \widetilde{\mathcal{U}}_k)(\widetilde{z})(\widetilde{\mathcal{U}}_k(\widetilde{z}) - \widetilde{\mathcal{U}}_k(\bar{z}))| \right]$$

$$\overset{(iii)}{\leqslant} B_\mathsf{S}^3 \sum_{k=1}^{K} \left[ |(\mathcal{U}_k - \widetilde{\mathcal{U}}_k)(\bar{z}) - (\mathcal{U}_k - \widetilde{\mathcal{U}}_k)(\widetilde{z})| + 2 B_f B_\tau \|f - \widetilde{f}\|_{2,\infty} |\widetilde{\mathcal{U}}_k(\widetilde{z}) - \widetilde{\mathcal{U}}_k(\bar{z})| \right],$$

where step (ii) uses Assumption 1, step (iii) uses Assumption 1, Eq. (24) and a Taylor expansion of $\exp(x)$. Similar to the proof of Eq. (20a), by counting the number of terms in the summations that are different and using Assumption 1, we find

$$\sum_{k=1}^{K} |\widetilde{\mathcal{U}}_k(\widetilde{\boldsymbol{z}}) - \widetilde{\mathcal{U}}_k(\bar{\boldsymbol{z}})| \leqslant 2B_{\mathsf{S}}, \text{ and}$$

$$\sum_{k=1}^{K} |(\mathcal{U}_k - \widetilde{\mathcal{U}}_k)(\bar{\boldsymbol{z}}) - (\mathcal{U}_k - \widetilde{\mathcal{U}}_k)(\widetilde{\boldsymbol{z}})| \leqslant 4B_{\mathsf{S}}B_f B_\tau \|f - \widetilde{f}\|_{2,\infty}.$$

Similar results hold for $\mathcal{V}$ by symmetry. Putting pieces together, we obtain

$$|U_4(\bar{\boldsymbol{z}}_1, \ldots, \bar{\boldsymbol{z}}_{(i-1)K+j}, \ldots, \bar{\boldsymbol{z}}_n) - U_4(\bar{\boldsymbol{z}}_1, \ldots, \widetilde{\boldsymbol{z}}_{(i-1)K+j}, \ldots, \bar{\boldsymbol{z}}_n)| \leqslant \frac{4B_{\mathsf{S}}^6 B_f B_\tau}{n}.$$

Therefore, it follows from Corollary 2.21 in [44] for functions with bounded differences that

$$\|U_4 - \mathbb{E}[U_4]\|_{\psi_2} \leqslant \frac{cB_{\mathsf{S}}^6 B_f B_\tau}{\sqrt{n}}. \tag{26}$$

Substituting Eq. (25) and (26) into Eq. (23), we obtain that $\{X_f, f \in \mathcal{F}\}$ is a zero-mean sub-Gaussian process with respect to the metric $\rho_X(f, \widetilde{f}) := B\|f - \widetilde{f}\|_{2,\infty}/\sqrt{n}$. This concludes the proof of Eq. (20b).

### C.3 Proof of Theorem 2

Write $\boldsymbol{z} = g(\boldsymbol{x})$ with $g \sim \mathbb{P}_\mathcal{G} \perp\!\!\!\perp \boldsymbol{x} \sim \mathbb{P}_\mathcal{X}$. Define $h_{\min} := \operatorname{argmin}_h \mathbb{E}_{\boldsymbol{x} \sim \mathbb{P}_\mathcal{X}, g \sim \mathbb{P}_\mathcal{G}}[(h(g(\boldsymbol{x})) - h_\star(\boldsymbol{x}))^2]$ and $\mathsf{h}(\boldsymbol{u}) := \mathbb{E}[h_{\min}(\boldsymbol{z}^{(1)}) | f(\boldsymbol{z}^{(1)}) = \boldsymbol{u}]$. Note that $|h_{\min}(\boldsymbol{z}^{(1)})| = |\mathbb{E}[h_\star(\boldsymbol{x})|\boldsymbol{z}^{(1)}]|$ is bounded by $B_{h_\star}$ almost surely by the assumption in Theorem 2. We first show that $\mathsf{R}_\mathcal{G}(\mathsf{h} \circ f)$ satisfies bound (7a) with $\epsilon_\mathcal{G}$ replaced by $\widetilde{\epsilon}_\mathcal{G} = \inf_h \mathbb{E}_{\boldsymbol{x} \sim \mathbb{P}_\mathcal{X}, g \sim \mathbb{P}_\mathcal{G}}[(h(g(\boldsymbol{x})) - h_\star(\boldsymbol{x}))^2]$. The original bound (7a) follows immediately since $\widetilde{\epsilon}_\mathcal{G} \leqslant \epsilon_\mathcal{G}$.

Since $(a + b)^2 \leqslant 2a^2 + 2b^2$, we have

$$\mathsf{R}_\mathcal{G}(\mathsf{h} \circ f) = \mathbb{E}_{\boldsymbol{x}, \boldsymbol{z}^{(1)}, \boldsymbol{z}^{(2)}}[(\mathsf{h}(f(\boldsymbol{z}^{(1)})) - h_\star(\boldsymbol{x}))^2] \leqslant 2\mathbb{E}_{\boldsymbol{x}, \boldsymbol{z}^{(1)}, \boldsymbol{z}^{(2)}}[(\mathsf{h}(f(\boldsymbol{z}^{(1)})) - h_{\min}(\boldsymbol{z}^{(2)}))^2] + 2\widetilde{\epsilon}_\mathcal{G}. \tag{27a}$$

Introduce a random variable which follows the distribution of $\boldsymbol{z}^{(1)}$ conditioned on $f(\boldsymbol{z}^{(1)})$ and is independent of $(\boldsymbol{z}^{(1)}, \boldsymbol{z}^{(2)})$ when conditioned on $f(\boldsymbol{z}^{(1)})$, i.e., $[\widetilde{\boldsymbol{z}}^{(1)} \sim \mathbb{P}_{\boldsymbol{z}}(\boldsymbol{z}^{(1)}|f(\boldsymbol{z}^{(1)})) \perp\!\!\!\perp (\boldsymbol{z}^{(1)}, \boldsymbol{z}^{(2)})]|f(\boldsymbol{z}^{(1)})$. Consider the joint distribution of the tuple $(\widetilde{\boldsymbol{z}}^{(1)}, \boldsymbol{z}^{(1)}, \boldsymbol{z}^{(2)})$. By Bayes' formula, we have $\widetilde{\boldsymbol{z}}^{(1)} \stackrel{d}{=} \boldsymbol{z}^{(1)} \sim \mathbb{P}_{\boldsymbol{z}}$ and $\boldsymbol{z}^{(2)}|\widetilde{\boldsymbol{z}}^{(1)} \sim \mathbb{P}(\boldsymbol{z}^{(2)}|f(\boldsymbol{z}^{(1)}) = f(\widetilde{\boldsymbol{z}}^{(1)}))$ and therefore

$$\mathbb{E}[(\mathsf{h}(f(\boldsymbol{z}^{(1)})) - h_{\min}(\boldsymbol{z}^{(2)}))^2] \stackrel{(i)}{\leqslant} \mathbb{E}[(h_{\min}(\widetilde{\boldsymbol{z}}^{(1)}) - h_{\min}(\boldsymbol{z}^{(2)}))^2]$$

$$= \mathbb{E}_{\widetilde{\boldsymbol{z}}^{(1)} \sim \mathbb{P}_{\boldsymbol{z}}, \boldsymbol{z}^{(2)} \sim \mathbb{P}(\boldsymbol{z}^{(2)}|f(\boldsymbol{z}^{(1)}) = f(\widetilde{\boldsymbol{z}}^{(1)}))}[(h_{\min}(\widetilde{\boldsymbol{z}}^{(1)}) - h_{\min}(\boldsymbol{z}^{(2)}))^2], \tag{27b}$$

where step (i) follows from

$$\mathbb{E}[(\mathsf{h}(f(\boldsymbol{z}^{(1)})) - h_{\min}(\boldsymbol{z}^{(2)}))^2|f(\boldsymbol{z}^{(1)})] \leqslant \mathbb{E}[(h_{\min}(\widetilde{\boldsymbol{z}}^{(1)}) - h_{\min}(\boldsymbol{z}^{(2)}))^2|f(\boldsymbol{z}^{(1)})],$$

which uses Jensen's inequality, the independence of $\widetilde{\boldsymbol{z}}^{(1)}$ and $\boldsymbol{z}^{(2)}$ conditioned on $f(\boldsymbol{z}^{(1)})$, and the fact that $\mathbb{E}[h_{\min}(\widetilde{\boldsymbol{z}}^{(1)})|f(\boldsymbol{z}^{(1)})] = \mathsf{h}(f(\boldsymbol{z}^{(1)}))$. Moreover,

$$\mathbb{E}_{\widetilde{\boldsymbol{z}}^{(1)} \sim \mathbb{P}_{\boldsymbol{z}}, \boldsymbol{z}^{(2)} \sim \mathbb{P}(\boldsymbol{z}^{(2)}|f(\boldsymbol{z}^{(1)}) = f(\widetilde{\boldsymbol{z}}^{(1)}))}[(h_{\min}(\widetilde{\boldsymbol{z}}^{(1)}) - h_{\min}(\boldsymbol{z}^{(2)}))^2]$$

$$\stackrel{(ii)}{\leqslant} \mathbb{E}_{\widetilde{\boldsymbol{z}}^{(1)} \sim \mathbb{P}_{\boldsymbol{z}}, \boldsymbol{z}^{(2)} \sim \mathbb{P}(\boldsymbol{z}^{(2)}|\boldsymbol{z}^{(1)} = \widetilde{\boldsymbol{z}}^{(1)})}[(h_{\min}(\widetilde{\boldsymbol{z}}^{(1)}) - h_{\min}(\boldsymbol{z}^{(2)}))^2]$$

$$+ \sqrt{2}B_{h_\star}^2 \cdot \mathbb{E}_{\widetilde{\boldsymbol{z}}^{(1)} \sim \mathbb{P}_{\boldsymbol{z}}}\left[\sqrt{\mathsf{D}_{\mathrm{KL}}\left(\mathbb{P}_{\boldsymbol{z}^{(2)}|\boldsymbol{z}^{(1)}}(\cdot|\widetilde{\boldsymbol{z}}^{(1)}) \middle\| \mathbb{P}_{\boldsymbol{z}^{(2)}|\boldsymbol{z}^{(1)}}(\cdot|f(\widetilde{\boldsymbol{z}}^{(1)}))\right)}\right]$$

$$\stackrel{(iii)}{\leqslant} \mathbb{E}_{\widetilde{\boldsymbol{z}}^{(1)} \sim \mathbb{P}_{\boldsymbol{z}}, \boldsymbol{z}^{(2)} \sim \mathbb{P}(\boldsymbol{z}^{(2)}|\boldsymbol{z}^{(1)} = \widetilde{\boldsymbol{z}}^{(1)})}[(h_{\min}(\widetilde{\boldsymbol{z}}^{(1)}) - h_{\min}(\boldsymbol{z}^{(2)}))^2] + \sqrt{2}B_{h_\star}^2 \cdot \sqrt{\mathrm{Suff}_{\mathsf{cb},\mathsf{kl}}(f)}$$

$$= \mathbb{E}_{\boldsymbol{z}^{(1)}, \boldsymbol{z}^{(2)}}[(h_{\min}(\boldsymbol{z}^{(1)}) - h_{\min}(\boldsymbol{z}^{(2)}))^2] + \sqrt{2} B_{h_\star}^2 \cdot \sqrt{\mathrm{Suff}_{\mathrm{cb,kl}}(f)}, \tag{27c}$$

where step (ii) follows from the variational form of total variation distance and Pinsker's inequality, while step (iii) uses the (CBS) definition of $\mathrm{Suff}_{\mathrm{kl}}(f)$ in Definition 1 and Jensen's inequality. Lastly, we have from a triangle inequality that

$$\mathbb{E}_{\boldsymbol{z}^{(1)}, \boldsymbol{z}^{(2)}}[(h_{\min}(\boldsymbol{z}^{(1)}) - h_{\min}(\boldsymbol{z}^{(2)}))^2]$$
$$\leqslant 2(\mathbb{E}_{\boldsymbol{x}, \boldsymbol{z}^{(1)}}[(h_{\min}(\boldsymbol{z}^{(1)}) - h_\star(\boldsymbol{x}))^2] + \mathbb{E}_{\boldsymbol{x}, \boldsymbol{z}^{(2)}}[(h_{\min}(\boldsymbol{z}^{(2)}) - h_\star(\boldsymbol{x}))^2]) = 4\widetilde{\epsilon}_{\mathcal{G}}. \tag{27d}$$

Combining Eq. (27a)—(27d) yields Eq. (7a) in Theorem 2. Eq. (7b) in Theorem 2 follows immediately by noting

$$\mathsf{R}(\mathsf{h} \circ f) = \mathbb{E}[(\mathsf{h}(f(\boldsymbol{x})) - h_\star(\boldsymbol{x}))^2] = \mathbb{E}_{\boldsymbol{z}^{(1)}}[(\mathsf{h}(f(\boldsymbol{z}^{(1)})) - h_\star(\boldsymbol{z}^{(1)}))^2]$$
$$\leqslant 2\mathbb{E}_{\boldsymbol{z}^{(1)}, \boldsymbol{z}^{(2)}}[(\mathsf{h}(f(\boldsymbol{z}^{(1)})) - h_\star(\boldsymbol{x}))^2] + 2\mathbb{E}_{\boldsymbol{z}^{(1)}, \boldsymbol{z}^{(2)}}[(h_\star(\boldsymbol{z}^{(1)}) - h_\star(\boldsymbol{x}))^2]$$
$$= 2\mathbb{E}_{\boldsymbol{z}^{(1)}, \boldsymbol{z}^{(2)}}[(\mathsf{h}(f(\boldsymbol{z}^{(1)})) - h_\star(\boldsymbol{x}))^2] + 2\epsilon_{\mathcal{G}}$$

and using Eq. (7a).

**Comments on Theorem 2.** Following the same proof strategy, it can be verified that Eq. (7a) and (7b) also hold when choosing $\mathsf{h}(\boldsymbol{u}) := \mathbb{E}[h_\star(\boldsymbol{z}^{(1)}) | f(\boldsymbol{z}^{(1)}) = \boldsymbol{u}]$. The main difference in the proof is to replace $h_{\min}$ by $h_\star$ in Eq. (27a)— (27d).

Moreover, although we consider the expected squared loss (i.e., $\ell(x, y) = (x - y)^2$) for simplicity, it can be seen from the proof that a similar version of Eq. (7a) and (7b) hold for any semimetric $\ell(x, y)$ that is convex in $x$ for all $y$. This includes the absolute loss, Huber loss, losses induced by norms, etc.

### C.4 Proof of Theorem 3

For any densities $\mathbb{P}, \mathbb{Q}$, define $\alpha$-Rényi divergence

$$\mathsf{D}_\alpha(\mathbb{P} || \mathbb{Q}) := \frac{1}{\alpha - 1} \log \left( \mathbb{E}_{x \sim \mathbb{P}} \left[ \left( \frac{\mathbb{P}(x)}{\mathbb{Q}(x)} \right)^{\alpha - 1} \right] \right)$$

for any $\alpha > 0$. Note that the 1-Rényi divergence corresponds to the KL divergence. For any densities $\mathbb{P}, \mathbb{Q}, \mathbb{T}$, we have the following triangle-like inequality which we will repeatedly use in the proof.

**Lemma 5** (Triangle-like inequality for Rényi divergence (Lemma 26 in Bun and Steinke [4])). *Let $\mathbb{P}$, $\mathbb{Q}$, and $\mathbb{T}$ be probability densities w.r.t. the same measure. Then*

$$\mathsf{D}_\alpha(\mathbb{P} || \mathbb{Q}) \leqslant \frac{k\alpha}{k\alpha - 1} \mathsf{D}_{\frac{k\alpha - 1}{k - 1}}(\mathbb{P} || \mathbb{T}) + \mathsf{D}_{k\alpha}(\mathbb{T} || \mathbb{Q})$$

*for all $k, \alpha \in (1, \infty)$.*

Write $\boldsymbol{z} = g(\boldsymbol{x})$ with $g \sim \mathbb{P}_{\mathcal{G}} \perp\!\!\!\perp \boldsymbol{x} \sim \mathbb{P}_{\mathcal{X}}$ and define $\mathsf{h}(f(\boldsymbol{z})) := \mathbb{P}(\boldsymbol{y} | f(\boldsymbol{z})) \in \Delta([\mathsf{K}])$ as the conditional distribution of $\boldsymbol{y}$ given $f(\boldsymbol{z})$, where $\boldsymbol{z} = g(\boldsymbol{x})$ for some random transformation $g \sim \mathbb{P}_{\mathcal{G}}$. It can be verified that $\mathsf{h} = \operatorname{argmin}_{\mathbb{Q}: \mathbb{R}^p \mapsto \Delta([\mathsf{K}])} \mathsf{D}_{\mathrm{KL}}(\mathbb{P}(\boldsymbol{y} | \boldsymbol{x}) || \mathbb{Q}(\boldsymbol{y} | f(\boldsymbol{z})))$. Therefore, using Lemma 5 with $k = 4/3, \alpha = 1$ (by taking the limit $\alpha \to 1$), we obtain

$$\mathsf{R}_{\mathcal{G}}^{\mathsf{cls}}(\mathsf{h} \circ f) = \mathbb{E}_{\boldsymbol{x}, \boldsymbol{y}, \boldsymbol{z}^{(1)}}[\mathsf{D}_{\mathrm{KL}}(\mathbb{P}(\boldsymbol{y} | \boldsymbol{x}) || \mathbb{P}(\boldsymbol{y} | f(\boldsymbol{z}^{(1)})))]$$
$$\leqslant 4\mathbb{E}_{\boldsymbol{x}, \boldsymbol{y}, \boldsymbol{z}^{(2)}}[\mathsf{D}_{\mathrm{KL}}(\mathbb{P}(\boldsymbol{y} | \boldsymbol{x}) || \mathbb{P}(\boldsymbol{y} | \boldsymbol{z}^{(2)}))] + \mathbb{E}_{\boldsymbol{x}, \boldsymbol{y}, \boldsymbol{z}^{(1)}, \boldsymbol{z}^{(2)}}[\mathsf{D}_{4/3}(\mathbb{P}(\boldsymbol{y} | \boldsymbol{z}^{(2)}) || \mathbb{P}(\boldsymbol{y} | f(\boldsymbol{z}^{(1)})))]$$
$$\leqslant 4\epsilon_{\mathcal{G}}^{\mathsf{cls}} + \mathbb{E}_{\boldsymbol{x}, \boldsymbol{y}, \boldsymbol{z}^{(1)}, \boldsymbol{z}^{(2)}}[\mathsf{D}_{4/3}(\mathbb{P}(\boldsymbol{y} | \boldsymbol{z}^{(2)}) || \mathbb{P}(\boldsymbol{y} | f(\boldsymbol{z}^{(1)})))], \tag{28a}$$

where the last inequality uses the monotonicity of $\alpha$-Rényi divergence w.r.t. $\alpha$. Similar to the proof of Theorem 2, introduce a random variable which follows the distribution of $\boldsymbol{z}^{(1)}$ conditioned on $f(\boldsymbol{z}^{(1)})$ and is independent of $(\boldsymbol{z}^{(1)}, \boldsymbol{z}^{(2)})$ when conditioned on $f(\boldsymbol{z}^{(1)})$, i.e., $[\widetilde{\boldsymbol{z}}^{(1)} \sim \mathbb{P}_{\boldsymbol{z}}(\boldsymbol{z}^{(1)} | f(\boldsymbol{z}^{(1)})) \perp\!\!\!\perp (\boldsymbol{z}^{(1)}, \boldsymbol{z}^{(2)})] | f(\boldsymbol{z}^{(1)})$. Consider the joint distribution of the tuple $(\widetilde{\boldsymbol{z}}^{(1)}, \boldsymbol{z}^{(1)}, \boldsymbol{z}^{(2)})$. By Bayes' formula, we have $\widetilde{\boldsymbol{z}}^{(1)} \stackrel{d}{=} \boldsymbol{z}^{(1)} \sim \mathbb{P}_{\boldsymbol{z}}$ and $\boldsymbol{z}^{(2)} | \widetilde{\boldsymbol{z}}^{(1)} \sim \mathbb{P}(\boldsymbol{z}^{(2)} | f(\boldsymbol{z}^{(1)}) = f(\widetilde{\boldsymbol{z}}^{(1)}))$ and thus

$$\mathbb{E}_{\boldsymbol{x}, \boldsymbol{y}, \boldsymbol{z}^{(1)}, \boldsymbol{z}^{(2)}}[\mathsf{D}_{4/3}(\mathbb{P}(\boldsymbol{y} | \boldsymbol{z}^{(2)}) || \mathbb{P}(\boldsymbol{y} | f(\boldsymbol{z}^{(1)})))] \stackrel{(i)}{\leqslant} \mathbb{E}_{\boldsymbol{x}, \boldsymbol{y}, \boldsymbol{z}^{(1)}, \boldsymbol{z}^{(2)}}[\mathsf{D}_{4/3}(\mathbb{P}(\boldsymbol{y} | \boldsymbol{z}^{(2)}) || \mathbb{P}(\boldsymbol{y} | \widetilde{\boldsymbol{z}}^{(1)}))]$$

$$= \mathbb{E}_{\widetilde{\boldsymbol{z}}^{(1)} \sim \mathbb{P}_{\boldsymbol{z}}, \boldsymbol{z}^{(2)} \sim \mathbb{P}(\boldsymbol{z}^{(2)} | f(\boldsymbol{z}^{(1)}) = f(\widetilde{\boldsymbol{z}}^{(1)}))} [\mathsf{D}_{4/3}(\mathbb{P}(\boldsymbol{y} | \boldsymbol{z}^{(2)}) || \mathbb{P}(\boldsymbol{y} | \widetilde{\boldsymbol{z}}^{(1)}))], \tag{28b}$$

where step (i) uses Jensen's inequality, the convexity of Rényi divergence w.r.t. its second argument and the fact that $\mathbb{E}[\mathbb{P}(\boldsymbol{y} | \widetilde{\boldsymbol{z}}^{(1)}) | f(\widetilde{\boldsymbol{z}}^{(1)})] = \mathbb{P}(\boldsymbol{y} | f(\boldsymbol{z}^{(1)}) = f(\widetilde{\boldsymbol{z}}^{(1)}))$. Moreover,

$$\mathbb{E}_{\widetilde{\boldsymbol{z}}^{(1)} \sim \mathbb{P}_{\boldsymbol{z}}, \boldsymbol{z}^{(2)} \sim \mathbb{P}(\boldsymbol{z}^{(2)} | f(\boldsymbol{z}^{(1)}) = f(\widetilde{\boldsymbol{z}}^{(1)}))} [\mathsf{D}_{4/3}(\mathbb{P}(\boldsymbol{y} | \boldsymbol{z}^{(2)}) || \mathbb{P}(\boldsymbol{y} | \widetilde{\boldsymbol{z}}^{(1)}))]$$

$$\overset{(ii)}{\leqslant} \mathbb{E}_{\widetilde{\boldsymbol{z}}^{(1)} \sim \mathbb{P}_{\boldsymbol{z}}, \boldsymbol{z}^{(2)} \sim \mathbb{P}(\boldsymbol{z}^{(2)} | \boldsymbol{z}^{(1)} = \widetilde{\boldsymbol{z}}^{(1)})} [\mathsf{D}_{4/3}(\mathbb{P}(\boldsymbol{y} | \boldsymbol{z}^{(2)}) || \mathbb{P}(\boldsymbol{y} | \widetilde{\boldsymbol{z}}^{(1)}))]$$

$$+ \sqrt{2} B \cdot \mathbb{E}_{\widetilde{\boldsymbol{z}}^{(1)} \sim \mathbb{P}_{\boldsymbol{z}}} \left[ \sqrt{\mathsf{D}_{\mathrm{KL}} \left( \mathbb{P}_{\boldsymbol{z}^{(2)} | \boldsymbol{z}^{(1)}} (\cdot | \widetilde{\boldsymbol{z}}^{(1)}) \Big\| \mathbb{P}_{\boldsymbol{z}^{(2)} | \boldsymbol{z}^{(1)}} (\cdot | f(\widetilde{\boldsymbol{z}}^{(1)})) \right)} \right]$$

$$\overset{(iii)}{\leqslant} \mathbb{E}_{\widetilde{\boldsymbol{z}}^{(1)} \sim \mathbb{P}_{\boldsymbol{z}}, \boldsymbol{z}^{(2)} \sim \mathbb{P}(\boldsymbol{z}^{(2)} | \boldsymbol{z}^{(1)} = \widetilde{\boldsymbol{z}}^{(1)})} [\mathsf{D}_{4/3}(\mathbb{P}(\boldsymbol{y} | \boldsymbol{z}^{(2)}) || \mathbb{P}(\boldsymbol{y} | \widetilde{\boldsymbol{z}}^{(1)}))] + \sqrt{2} B \cdot \sqrt{\mathrm{Suff}_{\mathrm{cb,kl}}(f)}$$

$$= \mathbb{E}_{\boldsymbol{z}^{(1)}, \boldsymbol{z}^{(2)}} [\mathsf{D}_{4/3}(\mathbb{P}(\boldsymbol{y} | \boldsymbol{z}^{(2)}) || \mathbb{P}(\boldsymbol{y} | \boldsymbol{z}^{(1)}))] + \sqrt{2} B \cdot \sqrt{\mathrm{Suff}_{\mathrm{cb,kl}}(f)}, \tag{28c}$$

where step (ii) follows from the variational form of total variation distance, Pinsker's inequality and the fact that

$$\mathsf{D}_{4/3}(\mathbb{P}(\boldsymbol{y} | \boldsymbol{z}^{(2)}) || \mathbb{P}(\boldsymbol{y} | \widetilde{\boldsymbol{z}}^{(1)})) \leqslant \mathsf{D}_2(\mathbb{P}(\boldsymbol{y} | \boldsymbol{z}^{(2)}) || \mathbb{P}(\boldsymbol{y} | \widetilde{\boldsymbol{z}}^{(1)})) = \log \mathbb{E}_{\boldsymbol{y} \sim \mathbb{P}(\cdot | \boldsymbol{z}^{(2)})} \left[ \frac{\mathbb{P}(\boldsymbol{y} | \boldsymbol{z}^{(2)})}{\mathbb{P}(\boldsymbol{y} | \widetilde{\boldsymbol{z}}^{(1)})} \right] \leqslant B,$$

and step (iii) uses the CBS definition of $\mathrm{Suff}_{\mathrm{kl}}(f)$ and Jensen's inequality. Finally, applying Lemma 5 another time using $\alpha = 4/3$ and $k = 1.5$ yields

$$\mathbb{E}_{\boldsymbol{z}^{(1)}, \boldsymbol{z}^{(2)}} [\mathsf{D}_{4/3}(\mathbb{P}(\boldsymbol{y} | \boldsymbol{z}^{(2)}) || \mathbb{P}(\boldsymbol{y} | \boldsymbol{z}^{(1)}))]$$

$$\leqslant \mathbb{E}_{\boldsymbol{x}, \boldsymbol{z}^{(1)}} [\mathsf{D}_2(\mathbb{P}(\boldsymbol{y} | \boldsymbol{z}^{(2)}) || \mathbb{P}(\boldsymbol{y} | \boldsymbol{x}))] + \mathbb{E}_{\boldsymbol{x}, \boldsymbol{z}^{(2)}} [\mathsf{D}_2(\mathbb{P}(\boldsymbol{y} | \boldsymbol{x}) || \mathbb{P}(\boldsymbol{y} | \boldsymbol{z}^{(1)}))]) \leqslant \epsilon_{\mathcal{G}}^{\mathrm{cls}}. \tag{28d}$$

Combining Eq. (28a)—(28d) yields Theorem 3.

## C.5  Proof of Theorem 4

Let $\mathsf{f}(x) = (x-1)^2/2$. The proof largely follows the same arguments as the proof of Theorem 1. Thus we only provide a sketch of the proof here. First, it can be readily verified that the set of minimizers of $R_{\mathrm{f}}(\mathsf{S})$ is

$$\mathcal{M}_{\mathsf{S}} := \left\{ \mathsf{S} : \mathsf{S} = \mathsf{S}_\star + \text{const for some const} \in \mathbb{R}, \quad \mathsf{S}_\star(\boldsymbol{z}^{(1)}, \boldsymbol{z}^{(2)}) := \frac{\mathbb{P}(\boldsymbol{z}^{(1)}, \boldsymbol{z}^{(2)})}{\mathbb{P}(\boldsymbol{z}^{(1)}) \cdot \mathbb{P}(\boldsymbol{z}^{(2)})} \right\}.$$

Moreover, basic algebra shows that $\widehat{\mathsf{R}}_{\mathrm{chisq}, K}(\mathsf{S}_f)$ is an unbiased estimate of $R_{\mathrm{f}}(\mathsf{S}_f)$. Thus, by the VFS in Definition 1, we have the decomposition

$$\mathrm{Suff}_{\chi^2}(\widehat{f}) \leqslant R_{\mathrm{f}}(\mathsf{S}_f) - R_{\mathrm{f}}(\mathsf{S}_\star) \leqslant \underbrace{\left[ R_{\mathrm{f}}(\mathsf{S}_{\widehat{f}}) - \inf_{f \in \mathcal{F}} R_{\mathrm{f}}(\mathsf{S}_f) \right]}_{\text{generalization error}} + \underbrace{\left[ \inf_{f \in \mathcal{F}} R_{\mathrm{f}}(\mathsf{S}_f) - R_{\mathrm{f}}(\mathsf{S}_\star) \right]}_{\text{approximation error}}.$$

Therefore, it remains to show

(1). With probability at least $1 - \delta$, the excess risk

$$R_{\mathrm{f}}(\mathsf{S}_{\widehat{f}}) - \inf_{f \in \mathcal{F}} R_{\mathrm{f}} \leqslant \frac{c \bar{B}_{\mathsf{S}}^2}{\sqrt{n}} \left[ \sqrt{\log(1/\delta)} + B_\tau^2 \int_0^{2(\bar{B}_{\mathsf{S}} + B_\tau)} \sqrt{\log \mathcal{N}(u, \| \cdot \|_{2, \infty}, \mathcal{F})} \, du \right] \tag{29}$$

for some absolute constant $c > 0$.

Proof of Eq. (29). Recall the definition of $\widehat{\mathsf{R}}_{\mathrm{chisq}, K}$ in Eq. (10) and adopt the shorthand $\widehat{\mathsf{R}}_K$ for $\widehat{\mathsf{R}}_{\mathrm{chisq}, K}$. Let $B_f := \sqrt{B_\tau (\bar{B}_{\mathsf{S}} + B_\tau)}$, $B := c(\bar{B}_{\mathsf{S}} + 1) B_f B_\tau$ for some absolute constant $c > 0$. It

can be verified using Assumption 2 that $\mathcal{F}$ must satisfy $\|f\|_{2,\infty} \leqslant B_f$ for all $f \in \mathcal{F}$ for Assumption 1 to hold. Define the zero-mean random process $X_f := \widehat{\mathsf{R}}_K(\mathsf{S}_f) - \mathbb{E}[\widehat{\mathsf{R}}_K(\mathsf{S}_f)]$, $f \in \mathcal{F}$. We will prove that for some absolute constant $c > 0$

$$\mathbb{P}\Big(\big|\sup_{f\in\mathcal{F}}|X_f| - \mathbb{E}[\sup_{f\in\mathcal{F}}|X_f|]\big| \geqslant t\Big) \leqslant 2\exp\Big(-\frac{cnt^2}{\bar{B}_\mathsf{S}^4}\Big), \quad \text{for all } t \geqslant 0. \tag{30a}$$

$$\mathbb{E}[\sup_{f\in\mathcal{F}}|X_f|] \leqslant \mathbb{E}[|X_{f_0}|] + \mathbb{E}[\sup_{f,\widetilde{f}\in\mathcal{F}}|X_f - X_{\widetilde{f}}|] \leqslant c\frac{\bar{B}_\mathsf{S}^2}{\sqrt{n}} + 32\frac{B}{\sqrt{n}}\cdot\int_0^{2B_f}\sqrt{\log\mathcal{N}(u, \|\cdot\|_{2,\infty}, \mathcal{F})}du. \tag{30b}$$

Combining the two bounds and noting

$$\overline{\mathsf{R}}_K(\mathsf{S}_{\widehat{f}}) - \inf_{f\in\mathcal{F}}\overline{\mathsf{R}}_K(\mathsf{S}_f) \leqslant 2\sup_{f\in\mathcal{F}}|\widehat{\mathsf{R}}_K(\mathsf{S}_f) - \overline{\mathsf{R}}_K(\mathsf{S}_f)| = 2\sup_{f\in\mathcal{F}}|\widehat{\mathsf{R}}_K(\mathsf{S}_f) - \mathbb{E}[\widehat{\mathsf{R}}_K(\mathsf{S}_f)]| = 2\sup_{f\in\mathcal{F}}X_f,$$

yields claim (1).

**Proof of Eq. (30a).** Similar to the proof of Eq. (20a), we establish the bound using concentration properties for functions with bounded differences. Following the notations in the proof of Theorem 1, we let $\bar{z}_i = (z_i^{(1)}, z_i^{(2)})$. For any $i \in [n_1], j \in [K]$, suppose $\bar{z}_{(i-1)K+j}$ is replaced by $\widetilde{z}_{(i-1)K+j} = (\widetilde{z}_{(i-1)K+j}^{(1)}, \widetilde{z}_{(i-1)K+j}^{(2)})$ in the calculation of $\widehat{\mathsf{R}}_K(\mathsf{S}_f)$. It can be verified using Assumption 1 that

$$|X_f(\bar{z}_1, \ldots, \bar{z}_{(i-1)K+j}, \ldots, \bar{z}_n) - X_f(\bar{z}_1, \ldots, \widetilde{z}_{(i-1)K+j}, \ldots, \bar{z}_n)|$$

$$= |\widehat{\mathsf{R}}_K(\mathsf{S}_f)(\bar{z}_1, \ldots, \bar{z}_{(i-1)K+j}, \ldots, \bar{z}_n) - \widehat{\mathsf{R}}_K(\mathsf{S}_f)(\bar{z}_1, \ldots, \widetilde{z}_{(i-1)K+j}, \ldots, \bar{z}_n)| \leqslant \frac{c\bar{B}_\mathsf{S}^2}{n} \tag{31}$$

for some absolute constant $c > 0$. As a result, Eq. (20a) follows immediately from Corollary 2.21 in [44] for functions with bounded differences.

**Proof of Eq. (30b).** Similar to the proof of Eq. (20b), $\mathbb{E}[|X_{f_0}|] \leqslant c\bar{B}_\mathsf{S}^2/\sqrt{n}$ by the properties of zero-mean sub-Gaussian variable $X_{f_0}$, and therefore, to establish Eq. (30b), it remains to show $\{X_f, f \in \mathcal{F}\}$ is a zero-mean sub-Gaussian process with respect to the metric $\rho_X(f, \widetilde{f}) := B\|f - \widetilde{f}\|_{2,\infty}/\sqrt{n}$. Let $\|x\|_\psi := \inf\{t > 0 : \mathbb{E}[\psi(x/t)] \leqslant 1\}$ denote the Orlicz norm for random variables and let $\psi_2(u) = \exp(u^2) - 1$. Note that for any $z^{(1)}, z^{(2)}, z^{(2)\prime} \in \mathcal{X}, f, \widetilde{f} \in \mathcal{F}$, we have from Eq. (24) that

$$|\mathsf{S}_f(z^{(1)}, z^{(2)}) - \mathsf{S}_{\widetilde{f}}(z^{(1)}, z^{(2)})| \leqslant 2B_f B_\tau\|f - \widetilde{f}\|_{2,\infty}, \tag{32a}$$

and

$$|(\mathsf{S}_f(z^{(1)}, z^{(2)}) - \mathsf{S}_f(z^{(1)}, z^{(2)\prime}))^2 - (\mathsf{S}_{\widetilde{f}}(z^{(1)}, z^{(2)}) - \mathsf{S}_{\widetilde{f}}(z^{(1)}, z^{(2)\prime}))^2|$$

$$\overset{(i)}{\leqslant} 4\bar{B}_\mathsf{S}(|\mathsf{S}_f(z^{(1)}, z^{(2)}) - \mathsf{S}_{\widetilde{f}}(z^{(1)}, z^{(2)})| + |\mathsf{S}_f(z^{(1)}, z^{(2)\prime}) - \mathsf{S}_{\widetilde{f}}(z^{(1)}, z^{(2)\prime})|)$$

$$\leqslant 16\bar{B}_\mathsf{S}B_f B_\tau\|f - \widetilde{f}\|_{2,\infty}, \tag{32b}$$

where step (i) uses Assumption 1. Then, following the proof of Eq. (20b), it can be verified that

$$\|X_f - X_{\widetilde{f}}\|_{\psi_2} = \|\widehat{\mathsf{R}}_K(\mathsf{S}_f) - \widehat{\mathsf{R}}_K(\mathsf{S}_{\widetilde{f}}) - \mathbb{E}[\widehat{\mathsf{R}}_K(\mathsf{S}_f) - \widehat{\mathsf{R}}_K(\mathsf{S}_{\widetilde{f}})]\|_{\psi_2} \leqslant \frac{c(\bar{B}_\mathsf{S}+1)B_f B_\tau}{\sqrt{n}}\|f - \widetilde{f}\|_{2,\infty}.$$

# D Proofs in Section 4

## D.1 Proof of Theorem 6

Recall that $B = B_x B_\theta$. For linear regression with misspecified model, by Theorem 11.3 in Györfi et al. [12] (see also e.g., Theorem 1.1 in Audibert and Catoni [2]), we have

$$\mathbb{E}[\mathsf{R}_{\text{lin}}(\widetilde{\mathsf{h}}_{\widehat{\eta}})] - \bar{\sigma}^2 \leqslant 8(\inf_{\eta\in\mathbb{R}^p}\mathsf{R}_{\text{lin}}(\mathsf{h}_\eta) - \bar{\sigma}^2) + c(B^2 + \bar{\sigma}^2)\frac{p\log m}{m}$$

for some absolute constant $c > 0$.

Thus it suffices to show

$$\inf_{\boldsymbol{\eta} \in \mathbb{R}^p} \mathsf{R}_{\mathsf{lin}}(\mathsf{h}_{\boldsymbol{\eta}}) - \bar{\sigma}^2 \leqslant c(B^2 c_2 \sqrt{\mathrm{Suff}_{\mathrm{f}}(f)} + \epsilon_{\mathcal{G}}) \tag{33}$$

for some absolute constant $c > 0$. Equivalently, we only need to find some $\boldsymbol{\eta} \in \mathbb{R}^p$ such that $\mathsf{R}_{\mathsf{lin}}(\mathsf{h}_{\boldsymbol{\eta}})$ satisfies the bound in Eq. (33). On the other hand, from the proof of Theorem 2, we see that if we choose $h_\star(\boldsymbol{x}) = \langle \boldsymbol{x}, \boldsymbol{\theta}_\star \rangle$ and $h(\boldsymbol{u}) := \mathbb{E}[h_\star(\boldsymbol{z})|f(\boldsymbol{z}) = \boldsymbol{u}] = \langle \boldsymbol{\theta}_\star, \mathbb{E}[\boldsymbol{z}|f(\boldsymbol{z}) = \boldsymbol{u}] \rangle$, then the excess risk

$$\mathsf{R}_{\mathsf{lin}}(h) - \bar{\sigma}^2 \leqslant c(B^2 c_2 \sqrt{\mathrm{Suff}_{\mathrm{f}}(f)} + \epsilon_{\mathcal{G}})$$

for some absolute constant $c > 0$ by Theorem 2 and Proposition 5. Therefore, it remains to show $h$ is linear in $f(\boldsymbol{z})$. Note that $f(\boldsymbol{z}) = \boldsymbol{W}\boldsymbol{z}$. Let $\boldsymbol{W}^\dagger = \boldsymbol{W}^\top (\boldsymbol{W}\boldsymbol{W}^\top)^{-1} \in \mathbb{R}^{d \times p}$ be the generalized inverse of $\boldsymbol{W}$ and $\tilde{\boldsymbol{\eta}} = \boldsymbol{W}^{\dagger\top} \boldsymbol{\theta}_\star \in \mathbb{R}^p$. In fact, choosing $\tilde{\boldsymbol{\eta}} = \boldsymbol{W}^{\dagger\top} \boldsymbol{\theta}_\star \in \mathbb{R}^p$, we have

$$h(\boldsymbol{u}) = \langle \boldsymbol{\theta}_\star, \mathbb{E}[\boldsymbol{z}|f(\boldsymbol{z}) = \boldsymbol{u}] \rangle = \langle \boldsymbol{\theta}_\star, \mathbb{E}[\boldsymbol{W}^\dagger \boldsymbol{u} + (\mathrm{I}_d - \boldsymbol{W}^\dagger \boldsymbol{W})\boldsymbol{z}|f(\boldsymbol{z}) = \boldsymbol{u}] \rangle = \langle \boldsymbol{\theta}_\star, \boldsymbol{W}^\dagger \boldsymbol{u} \rangle = \langle \tilde{\boldsymbol{\eta}}, \boldsymbol{u} \rangle,$$

where the third equality uses the assumption that $\mathbb{E}[(\mathrm{I}_d - \boldsymbol{W}^\dagger \boldsymbol{W})\boldsymbol{z}|\boldsymbol{W}\boldsymbol{z}] = 0$ almost surely.

**Comments on Theorem 6.** A similar bound can be established for the risk $\mathsf{R}_{\mathsf{lin}}(\tilde{\mathsf{h}}_{\hat{\boldsymbol{\eta}}})$ with high probability under additional sub-Gaussian assumptions on the representation $f(\boldsymbol{z}) = \boldsymbol{W}g(\boldsymbol{x})$ [16]. The assumption $\mathbb{E}[(\mathrm{I}_d - \boldsymbol{W}^\dagger \boldsymbol{W})\boldsymbol{z}|\boldsymbol{W}\boldsymbol{z}] = 0$ essentially states that the information of the augmented view $\boldsymbol{z}$ discarded by the encoder $f$ does not contain any signal with a non-zero mean. Without this assumption, there may not exist a linear function of $f(\boldsymbol{z})$ that achieves a small risk, even though Theorem 2 guarantees the existence of a general function of $f(\boldsymbol{z})$ with a small risk.

## D.2 Further results in Section 4.1

Following the setup in Section 4.1, we present a scenario where a linear encoder $f$ with low KL-sufficiency $\mathrm{Suff}_{\mathsf{kl}}(f)$ can be found through SimCLR loss minimization in Eq. (3).

Let $\mathsf{U} = (\mathsf{U}_1, \mathsf{U}_2) \in \mathbb{R}^{d \times d}$, where $\mathsf{U}_1 \in \mathbb{R}^{d \times p}$, be some fixed unitary matrix, and define $\boldsymbol{A} = \mathsf{U}_1 \mathsf{U}_1^\top$. For $i \in [2]$, we let $\mathbb{S}(\mathsf{U}_i) := \{\boldsymbol{v} \in \mathbb{R}^d : \|\boldsymbol{v}\|_2 = 1, (\mathrm{I}_d - \mathsf{U}_i \mathsf{U}_i^\top)\boldsymbol{v} = \boldsymbol{0}\}$ denote the unit sphere in the column space of $\mathsf{U}_i$. Assume $\boldsymbol{x} \in \mathbb{R}^d \sim \mathcal{N}(\boldsymbol{0}, \mathrm{I}_d/p)$ and consider the random transformation $g$ such that $\boldsymbol{z}^{(1)}|\boldsymbol{x} \stackrel{d}{=} \boldsymbol{A}\boldsymbol{x} + \eta$ conditioned on $\boldsymbol{z}^{(1)} \in \mathbb{S}(\mathsf{U}_1) \oplus \mathbb{S}(\mathsf{U}_2)$ [3], where the noise $\eta \sim \mathcal{N}(\boldsymbol{0}, \sigma^2 \mathrm{I}_d/p)$. A concrete example of this transformation involves zeroing out the second half of the coordinates of the sample $\boldsymbol{x}$, adding some Gaussian noise to all coordinates, and then normalizing both halves of the modified sample to have unit norm. Under this setup, it is readily verified that[4]

$$\mathbb{P}(\boldsymbol{z}^{(1)}, \boldsymbol{z}^{(2)}) \propto \exp\left(-\frac{p}{2}\left\langle \begin{pmatrix} \boldsymbol{z}^{(1)} \\ \boldsymbol{z}^{(2)} \end{pmatrix}, \begin{bmatrix} \mathsf{U}_1 \mathsf{U}_1^\top + \sigma^2 \mathrm{I}_d & \mathsf{U}_1 \mathsf{U}_1^\top \\ \mathsf{U}_1 \mathsf{U}_1^\top & \mathsf{U}_1 \mathsf{U}_1^\top + \sigma^2 \mathrm{I}_d \end{bmatrix}^{-1} \begin{pmatrix} \boldsymbol{z}^{(1)} \\ \boldsymbol{z}^{(2)} \end{pmatrix} \right\rangle\right) \cdot \mathbb{1}_{\{\boldsymbol{z}^{(1)}, \boldsymbol{z}^{(2)} \in \mathbb{S}(\mathsf{U}_1) \oplus \mathbb{S}(\mathsf{U}_2)\}},$$

$$\mathbb{P}(\boldsymbol{z}^{(1)}) \propto 1 \cdot \mathbb{1}_{\{\boldsymbol{z}^{(1)} \in \mathbb{S}(\mathsf{U}_1) \oplus \mathbb{S}(\mathsf{U}_2)\}},$$

$$\frac{\mathbb{P}(\boldsymbol{z}^{(1)}, \boldsymbol{z}^{(2)})}{\mathbb{P}(\boldsymbol{z}^{(1)})\mathbb{P}(\boldsymbol{z}^{(2)})} \propto \exp\left(\kappa \langle \boldsymbol{z}^{(1)}, \mathsf{U}_1 \mathsf{U}_1^\top \boldsymbol{z}^{(2)} \rangle\right) \cdot \mathbb{1}_{\{\boldsymbol{z}^{(1)}, \boldsymbol{z}^{(2)} \in \mathbb{S}(\mathsf{U}_1) \oplus \mathbb{S}(\mathsf{U}_2)\}}, \qquad \kappa := \frac{p}{\sigma^2(\sigma^2 + 2)} \leqslant \frac{p}{\sigma^4}.$$

Note that $(\boldsymbol{z}^{(1)}, \boldsymbol{z}^{(2)})$ restricting on $\mathbb{S}(\mathsf{U}_1)$ follows the joint von Mises-Fisher distribution (vMF) [8]. In this case, the optimal score $\mathsf{S}_\star(\boldsymbol{z}^{(1)}, \boldsymbol{z}^{(2)}) = \tau(\langle f_\star(\boldsymbol{z}^{(1)}), f_\star(\boldsymbol{z}^{(2)}) \rangle) + \mathrm{const}$ for $\tau(x) = \kappa x$ and $f_\star(\boldsymbol{z}) = \mathsf{U}_1 \boldsymbol{z}$. We present a sample complexity bound on learning $f_\star$ in Corollary 1.

**Corollary 1** (An upper bound on $\mathrm{Suff}_{\mathsf{cb,kl}}(\hat{f})$). *Under the setup in Section D.2, let $\mathcal{F} := \{f : f(\boldsymbol{z}) = \boldsymbol{W}\boldsymbol{z}, \ \boldsymbol{W} \in \mathbb{R}^{p \times d} \text{ and } \|\boldsymbol{W}\|_{op} \leqslant B_{\boldsymbol{W}}\}$ for some $B_{\boldsymbol{W}} \geqslant 1$, $\tau(x) = \kappa x$, and define $\hat{f}$ as the SimCLR*

---

[3] $\mathbb{S}(\mathsf{U}_1) \oplus \mathbb{S}(\mathsf{U}_2) := \{\boldsymbol{v} \in \mathbb{R}^d : \boldsymbol{v} = \boldsymbol{v}_1 + \boldsymbol{v}_2 \text{ for some } \boldsymbol{v}_1 \in \mathbb{S}(\mathsf{U}_1), \boldsymbol{v}_2 \in \mathbb{S}(\mathsf{U}_2)\}$.

[4] All densities are with respect to the Lebesgue measure.

*empirical risk minimizer obtained in Eq.* (3) *with batch size* $K$ *and* $n$ *samples. Then with probability at least* $1 - \delta$, *we have*

$$\text{Suff}_{\text{cb,kl}}(\widehat{f}) \leqslant \left(1 + \frac{C}{K}\right) \cdot \frac{\sqrt{dp \log B_{\boldsymbol{W}}} + \sqrt{\log(1/\delta)}}{\sqrt{n}},$$

*for some constant* $C > 0$ *depending polynomially on* $\exp(\kappa)$.

See the proof in Appendix D.3. Note that the constant $\exp(\kappa)$ depends on the noise level $\sigma$. When $\sigma \gtrsim p^{1/4}$, finding a near-sufficient encoder is relatively easy. By combining Theorem 6 and Corollary 1, we conclude that the encoder learned from SimCLR can achieve a small risk in the downstream linear regression task, provided there are sufficient pretraining and downstream samples, and data augmentation does not significantly alter the outcome of the true linear model. See Appendix D.4 for an end-to-end statement and its proof.

### D.3 Proof of Corollary 1

It suffices to apply Theorem 1 to the setup in Corollary 1.

By the boundedness of $\boldsymbol{z}^{(1)}, \boldsymbol{z}^{(2)}$ and the property that $\mathbb{E}_{\boldsymbol{z}^{(1)}, \boldsymbol{z}^{(2)} \sim \mathbb{P}_{\boldsymbol{z}} \times \mathbb{P}_{\boldsymbol{z}}} \left[ \frac{\mathbb{P}(\boldsymbol{z}^{(1)}, \boldsymbol{z}^{(2)})}{\mathbb{P}(\boldsymbol{z}^{(1)})\mathbb{P}(\boldsymbol{z}^{(2)})} \right] = 1$, we have

$$\sup_{\boldsymbol{z}^{(1)}, \boldsymbol{z}^{(2)}} \frac{\mathbb{P}(\boldsymbol{z}^{(1)}, \boldsymbol{z}^{(2)})}{\mathbb{P}(\boldsymbol{z}^{(1)})\mathbb{P}(\boldsymbol{z}^{(2)})} \leqslant \frac{\sup_{\boldsymbol{z}^{(1)}, \boldsymbol{z}^{(2)}} \frac{\mathbb{P}(\boldsymbol{z}^{(1)}, \boldsymbol{z}^{(2)})}{\mathbb{P}(\boldsymbol{z}^{(1)})\mathbb{P}(\boldsymbol{z}^{(2)})}}{\inf_{\boldsymbol{z}^{(1)}, \boldsymbol{z}^{(2)}} \frac{\mathbb{P}(\boldsymbol{z}^{(1)}, \boldsymbol{z}^{(2)})}{\mathbb{P}(\boldsymbol{z}^{(1)})\mathbb{P}(\boldsymbol{z}^{(2)})}} \leqslant \exp(2\kappa).$$

Similarly we have $\inf_{\boldsymbol{z}^{(1)}, \boldsymbol{z}^{(2)}} \frac{\mathbb{P}(\boldsymbol{z}^{(1)}, \boldsymbol{z}^{(2)})}{\mathbb{P}(\boldsymbol{z}^{(1)})\mathbb{P}(\boldsymbol{z}^{(2)})} \geqslant \exp(-2\kappa)$.

By properties of the von Mises-Fisher distribution (see e.g., [24]), it can be verified that

$$\frac{\mathbb{P}(\boldsymbol{z}^{(1)}, \boldsymbol{z}^{(2)})}{\mathbb{P}(\boldsymbol{z}^{(1)})\mathbb{P}(\boldsymbol{z}^{(2)})} = \mathcal{E}_p(\kappa) \cdot \exp\left(\kappa \langle \boldsymbol{z}^{(1)}, \mathsf{U}_1 \mathsf{U}_1^\top \boldsymbol{z}^{(2)} \rangle\right) \cdot \mathbb{1}_{\{\boldsymbol{z}^{(1)}, \boldsymbol{z}^{(2)} \in \mathbb{S}(\mathsf{U}_1) \oplus \mathbb{S}(\mathsf{U}_2)\}}, \qquad \kappa := \frac{p}{(1 + \sigma^2)^2 - 1},$$

where

$$\mathcal{E}_p(\kappa) := \frac{\Gamma(p/2) I_{p/2-1}(\kappa)}{(\frac{\kappa}{2})^{p/2-1}} = \Gamma(p/2) \cdot \sum_{m=0}^{\infty} \frac{1}{m! \Gamma(m + p/2)} \left(\frac{\kappa}{2}\right)^{2m} = \sum_{m=0}^{\infty} \frac{(p-2)!!}{(2m)!!(2m + p - 2)!!} \kappa^{2m}$$

$$< \sum_{m=0}^{\infty} \frac{1}{(2m)!} \kappa^{2m} < e^{\kappa}, \quad \text{and } \mathcal{E}_p(\kappa) > \frac{\Gamma(p/2)}{0! \Gamma(p/2)} \cdot \left(\frac{\kappa}{2}\right)^0 = 1. \tag{34}$$

Thus, when $\tau(x) = \kappa x$, Assumption 1 and 2 are satisfied with $B_{\mathsf{S}} = \exp(2\kappa), B_\tau = 2\kappa$ (note that the condition $\kappa^{-1} \leqslant B_\tau$ is unnecessary, as from the proof of Theorem 1, we only need $|\tau(\langle f(\boldsymbol{z}^{(1)}), \boldsymbol{z}^{(2)} \rangle)| \leqslant \log B_{\mathsf{S}}$, which follows from the boundedness of $\mathcal{F}$).

**Approximation error.** The approximation error $\inf_{f \in \mathcal{F}} \overline{\mathsf{R}}_{\text{simclr}, K}(\mathsf{S}_f) - \overline{\mathsf{R}}_{\text{simclr}, K}(\mathsf{S}_\star) = 0$ since $\mathsf{S}_\star + c_1$ is realized by $f_\star$ and the link function $\tau(x) = \kappa x$ for some normalizing constant $c_1$ and $\overline{\mathsf{R}}_{\text{simclr}, K}(\mathsf{S}_\star) = \overline{\mathsf{R}}_{\text{simclr}, K}(\mathsf{S}_\star + c_1)$.

**Generalization error.** Let $\mathcal{W} := \{\boldsymbol{W} \in \mathbb{R}^{p \times d}, \|\|\boldsymbol{W}\|\|_{\text{op}} \leqslant B_{\boldsymbol{W}}\}$. First, for $f_i(\boldsymbol{z}) = \boldsymbol{W}_i \boldsymbol{z}$ $(i = 1, 2)$, since $\|f_1 - f_2\|_{2,\infty} \leqslant \|\|\boldsymbol{W}_1 - \boldsymbol{W}_2\|\|_{\text{op}} \cdot \|\boldsymbol{z}\|_2 \leqslant 2\|\|\boldsymbol{W}_1 - \boldsymbol{W}_2\|\|_{\text{op}}$, it follows that

$$\log \mathcal{N}(u, \|\cdot\|_{2,\infty}, \mathcal{F}) \leqslant \log \mathcal{N}\left(\frac{u}{2}, \|\cdot\|_{\text{op}}, \mathcal{W}\right) \leqslant cdp \cdot \log\left(1 + \frac{4B_{\boldsymbol{W}}}{u}\right),$$

where the last inequality follows from the upper bound of the covering number of a unit ball (see e.g., exercise 5.8 in Wainwright [44]) and the assumption that $p \leqslant d$. Therefore,

$$B_\tau \int_0^{2(\log B_{\mathsf{S}} + B_\tau)} \sqrt{\log \mathcal{N}(u, \|\cdot\|_{2,\infty}, \mathcal{F})} du \leqslant c\kappa \int_0^{c\kappa} \sqrt{\log \mathcal{N}(u, \|\cdot\|_{2,\infty}, \mathcal{F})} du \leqslant c\sqrt{dp} \kappa^2 \sqrt{\log B_{\boldsymbol{W}}}.$$

Combining the result on the approximation error and the generalization error and applying Theorem 1 yields the desired result.

## D.4 An end-to-end result on downstream linear regression

Combining Theorem 6 and Corollary 1, we reach at the following result on the downstream performance of encoder learned by SimCLR.

**Theorem 9** (Linear regression using the SimCLR-trained encoder). *Under the setup described in Section 4.1, let $\widehat{f}$ be the empirical risk minimizer obtained from Eq. (3) in Corollary 1 on a restricted function space $\mathcal{F}^\circ := \{f(\boldsymbol{z}) = \boldsymbol{W}\boldsymbol{z} \in \mathcal{F}, \ \mathsf{span}(\boldsymbol{W}^\top) = (\mathsf{span}(\boldsymbol{W}^\top) \cap \mathsf{span}(\mathsf{U}_1)) \oplus (\mathsf{span}(\boldsymbol{W}^\top) \cap \mathsf{span}(\mathsf{U}_2))\} \subseteq \mathcal{F}$. In the downstream task, given $m$ i.i.d. samples $\{(\boldsymbol{x}_i, \boldsymbol{y}_i)\}_{i=1}^m$ from $\boldsymbol{y} = \mathrm{proj}_{[-B,B]}(\langle \boldsymbol{x}, \boldsymbol{\theta}_\star \rangle) + \varepsilon$, where $\boldsymbol{x} \sim \mathcal{N}(\boldsymbol{0}, \mathrm{I}_d/p)$ follows the same distribution as in contrastive learning, and $\varepsilon \sim \mathcal{N}(0, \bar{\sigma}^2) \perp\!\!\!\perp \boldsymbol{x}$.*

*(a). Consider fitting a (random) linear model $\mathsf{h}_{\boldsymbol{\eta}}(\boldsymbol{x}) = \langle \widehat{f}(\boldsymbol{z}), \boldsymbol{\eta} \rangle$ by ordinary least squares*

$$\widehat{\boldsymbol{\eta}} := \mathrm{argmin}_{\boldsymbol{\eta} \in \mathbb{R}^p} \Big\{ \widehat{\mathsf{R}}_{\mathsf{lin}}(\mathsf{h}_{\boldsymbol{\eta}}) := \frac{1}{m} \sum_{i=1}^m (\langle \widehat{f}(\boldsymbol{z}_i), \boldsymbol{\eta} \rangle - \boldsymbol{y}_i)^2 \Big\},$$

*where $\boldsymbol{z} = g(\boldsymbol{x})$, $\boldsymbol{z}_i = g_i(\boldsymbol{x}_i)$, and $g, \{g\}_{i=1}^m$ are i.i.d. transformations from $\mathbb{P}_\mathcal{G}$ as specified in Section 4.1. Then with probability at least $1 - \delta$ over the SimCLR training, the expected risk of the truncated linear model $\widetilde{\mathsf{h}}_{\widehat{\boldsymbol{\eta}}}(\boldsymbol{x}) := \mathrm{proj}_{[-B,B]}(\mathsf{h}_{\widehat{\boldsymbol{\eta}}}(\boldsymbol{x}))$ satisfies*

$$\mathbb{E}[\mathsf{R}_{\mathsf{lin}}(\widetilde{\mathsf{h}}_{\widehat{\boldsymbol{\eta}}})] := \mathbb{E}\big[\mathbb{E}_{\boldsymbol{x},\boldsymbol{y},g}[(\boldsymbol{y} - \widetilde{\mathsf{h}}_{\widehat{\boldsymbol{\eta}}}(\boldsymbol{x}))^2]\big]$$

$$\leqslant \underbrace{\bar{\sigma}^2}_{\text{irreducible risk}} + \underbrace{c\Big(B^2\Big(1 + \frac{C}{K}\Big) \cdot \frac{d^{1/4} p^{1/4} \log^{1/4} B_{\boldsymbol{W}} + \log^{1/4}(1/\delta)}{n^{1/4}} + \epsilon_\mathcal{G}\Big)}_{\text{Error from SimCLR training}} + \underbrace{c(\bar{\sigma}^2 + B^2)\frac{p \log m}{m}}_{\text{Error from downstream task}},$$

*where the outer expectation is over $\{(\boldsymbol{x}_i, \boldsymbol{y}_i, g_i)\}_{i=1}^n$, $c > 0$ is some absolute constant, $C > 0$ is some constant depending polynomially on $\exp(\kappa)$, and $\epsilon_\mathcal{G} \leqslant \mathbb{E}[\langle \boldsymbol{x} - \boldsymbol{z}, \boldsymbol{\theta}_\star \rangle^2]$.*

*(b). In contrast, suppose in addition $\bar{\sigma}^2 \geqslant 1$, $\|\boldsymbol{\theta}_\star\|_2 \leqslant B_{\boldsymbol{\theta}}$ and $m \geqslant cd$, $B \geqslant c(\bar{\sigma}^2 + B_{\boldsymbol{\theta}}^2)\log m/p$ for some absolute constant $c > 0$, then the truncated ordinary least squares estimator $\widetilde{\mathsf{h}}_{\mathsf{ols}}(\boldsymbol{x}) = \mathrm{proj}_{[-B,B]}(\langle \boldsymbol{x}, \widehat{\boldsymbol{\theta}}_{\mathsf{ols}} \rangle)$ obtained from $\{(\boldsymbol{x}_i, \boldsymbol{y}_i)\}_{i=1}^m$ satisfies*

$$\mathbb{E}[\mathsf{R}_{\mathsf{lin}}(\widetilde{\mathsf{h}}_{\mathsf{ols}})] - \bar{\sigma}^2 := \mathbb{E}\big[\mathbb{E}_{\boldsymbol{x},\boldsymbol{y}}[(\boldsymbol{y} - \widetilde{\mathsf{h}}_{\mathsf{ols}}(\boldsymbol{x}))^2]\big] - \bar{\sigma}^2 \asymp \bar{\sigma}^2 \frac{d}{m},$$

*where $\asymp$ denotes matching upper and lower bounds up to absolute constant factors, and the outer expectation is over $\{(\boldsymbol{x}_i, \boldsymbol{y}_i)\}_{i=1}^n$.*

We remark that the truncation in the data generation (i.e., $\boldsymbol{y} = \mathrm{proj}_{[-B,B]}(\langle \boldsymbol{x}, \boldsymbol{\theta}_\star \rangle) + \varepsilon$) is due to technical difficulties, however, we can choose the threshold $B$ sufficiently large, for example, $B = \mathcal{O}(\log m)$, so that the truncation rarely happens in the generated data. The restriction of the empirical risk minimization to $\mathcal{F}^\circ$ ensures that the condition $\mathbb{E}[(\mathrm{I}_d - \boldsymbol{W}^\dagger \boldsymbol{W})\boldsymbol{z}|\boldsymbol{W}\boldsymbol{z}] = 0$ in Theorem 6 holds for any $f(\boldsymbol{z}) = \boldsymbol{W}\boldsymbol{z} \in \mathcal{F}^\circ$. Without this restriction, when $\mathrm{Suff}(\widehat{f})$ is sufficiently small, the ERM $\widehat{f}(\boldsymbol{z}) = \widehat{\boldsymbol{W}}\boldsymbol{z}$ only satisfies $\mathbb{E}[(\mathrm{I}_d - \widehat{\boldsymbol{W}}^\dagger \widehat{\boldsymbol{W}})\boldsymbol{z}|\widehat{\boldsymbol{W}}\boldsymbol{z}] \approx 0$, and the downstream error bound would contain an additional term depending on the $\mathrm{Suff}(\widehat{f})$.

For the two-step estimator in (a), the first term in the SimCLR training error converges to zero as the pretraining sample size $n$ increases, and the second term $\epsilon_\mathcal{G}$ is negligible when either the ground truth $\mathbb{E}[\boldsymbol{y}|\boldsymbol{x}]$ does not vary significantly (i.e., $\|\boldsymbol{\theta}_\star\|_2$ is small) or the data augmentation introduces negligible error (i.e., $\|\boldsymbol{x} - \boldsymbol{z}\|_2$ is small). Thus, compared with the OLS estimator which has a risk of order $\mathcal{O}(d/m)$, the two-step estimator achieves a small risk of order $\mathcal{O}(p/m)$ when the error from SimCLR training is of higher order.

*Proof of Theorem 9.* First, we have from Corollary 1 that, with probability at least $1 - \delta$, the learned encoder satisfies

$$\mathrm{Suff}(\widehat{f}) \leqslant \Big(1 + \frac{C}{K}\Big) \cdot \frac{\sqrt{dp \log B_{\boldsymbol{W}}} + \sqrt{\log(1/\delta)}}{\sqrt{n}},$$

for some constant $C > 0$ depending polynomially on $\exp(\kappa)$. Note that the bound can be directly applied even though we consider the ERM on $\mathcal{F}^\circ \in \mathcal{F}$ since $f_\star \in \mathcal{F}^\circ$ and the proof of Corollary 1 follows from an upper bound on the supremum of an empirical process, which remains valid when restricting to a smaller function space $\mathcal{F}^\circ \subseteq \mathcal{F}$.

Consider the problem of fitting a linear regression using data $\{(\widehat{f}(\boldsymbol{z}_i), \boldsymbol{y}_i)\}_{i=1}^m$. We have

$$|\mathbb{E}[\boldsymbol{y}|\widehat{f}(\boldsymbol{z})]| \leqslant \mathbb{E}[|\mathbb{E}[\boldsymbol{y}|\boldsymbol{z}]||f(\boldsymbol{z})] = \mathbb{E}[|\mathbb{E}[\mathrm{proj}_{[-B,B]}(\langle \boldsymbol{x}, \boldsymbol{\theta}_\star \rangle)|\boldsymbol{z}]||f(\boldsymbol{z})] \leqslant B.$$

Thus the conditions required by Theorem 1.1 in [2] are satisfied and we have

$$\mathbb{E}[\mathsf{R}_{\mathsf{lin}}(\widetilde{\mathsf{h}}_{\widehat{\boldsymbol{\eta}}})] - \bar{\sigma}^2 \leqslant 8(\inf_{\boldsymbol{\eta} \in \mathbb{R}^p} \mathsf{R}_{\mathsf{lin}}(\mathsf{h}_{\boldsymbol{\eta}}) - \bar{\sigma}^2) + c(B^2 + \bar{\sigma}^2)\frac{p \log m}{m}.$$

Following the proof of Theorem 6, it remains to verify the condition $\mathbb{E}[(\mathrm{I}_d - \boldsymbol{W}^\dagger \widehat{\boldsymbol{W}})\boldsymbol{z}|\widehat{\boldsymbol{W}}\boldsymbol{z}] = 0$, where $\widehat{\boldsymbol{W}}$ is the linear map in $\widehat{f}$ (i.e., $\widehat{f}(\boldsymbol{z}) = \widehat{\boldsymbol{W}}\boldsymbol{z}$). This follows immediately as $\boldsymbol{z}$ follows the uniform distribution on $\mathbb{S}(\mathsf{U}_1) \oplus \mathbb{S}(\mathsf{U}_2)$.

Ordinary least squares estimator. Adopt the shorthand $\mathsf{p}$ for $\mathrm{proj}_{[-B,B]}$. When applying $\mathsf{p}$ to a vector, we apply it coordinate-wise. Let $\Sigma = \mathbb{E}[\boldsymbol{x}\boldsymbol{x}^\top] = \mathrm{I}_d/p$ be the covariance matrix. For the ordinary least squares (OLS) estimator, let $\boldsymbol{X} = (\boldsymbol{x}_1 \quad \ldots \quad \boldsymbol{x}_m)^\top \in \mathbb{R}^{m \times d}$ denote the sample matrix, $\boldsymbol{Y} = (\boldsymbol{y}_1 \quad \ldots \quad \boldsymbol{y}_m)^\top \in \mathbb{R}^m$ denote the response vector, and $\mathcal{E} = (\varepsilon_1 \quad \ldots \quad \varepsilon_m)^\top \in \mathbb{R}^m$ denote the noise vector. By the definition of OLS, we have $\widehat{\boldsymbol{\theta}} = (\boldsymbol{X}^\top \boldsymbol{X})^{-1} \boldsymbol{X}^\top \boldsymbol{Y}$ and

$$\mathbb{E}[\mathsf{R}_{\mathsf{lin}}(\widetilde{\mathsf{h}}_{\mathsf{ols}})] - \bar{\sigma}^2 = \mathbb{E}[(\mathsf{p}(\langle \boldsymbol{x}, (\boldsymbol{X}^\top \boldsymbol{X})^{-1} \boldsymbol{X}^\top \boldsymbol{Y} \rangle) - \mathsf{p}(\langle \boldsymbol{x}, \boldsymbol{\theta}_\star \rangle))^2].$$

We claim two results used later. The proof of them can be found at the end of this section.

$$\mathbb{E}[\mathrm{trace}((\boldsymbol{X}^\top \boldsymbol{X})^{-1}\Sigma)] = \frac{d}{m-d-1}, \quad \mathbb{E}[\mathrm{trace}((\boldsymbol{X}^\top \boldsymbol{X})^{-1}\Sigma)^2] = \frac{(m-1)d}{(m-d)(m-d-1)(m-d-3)}, \tag{35}$$

$$\mathbb{E}[\|[\mathsf{p}(\boldsymbol{X}\boldsymbol{\theta}_\star) - \boldsymbol{X}\boldsymbol{\theta}_\star]\|_2^4] \leqslant c\frac{m^2 B_{\boldsymbol{\theta}}^4}{p^2} \cdot \exp(-\frac{B^2}{cB_{\boldsymbol{\theta}}^2/p}) \tag{36}$$

for some absolute constant $c > 0$.

Choose $B \geqslant c(\bar{\sigma}^2 + B_{\boldsymbol{\theta}}^2)\log m/p$ for some sufficiently large absolute constant $c > 0$. We then have $\mathbb{E}[\|[\mathsf{p}(\boldsymbol{X}\boldsymbol{\theta}_\star) - \boldsymbol{X}\boldsymbol{\theta}_\star]\|_2^4] \leqslant m^{-4}$. On one hand, to establish the upper bound, we have

$$\mathbb{E}[\mathsf{R}_{\mathsf{lin}}(\widetilde{\mathsf{h}}_{\mathsf{ols}})] - \bar{\sigma}^2 \leqslant \mathbb{E}[(\langle \boldsymbol{x}, (\boldsymbol{X}^\top \boldsymbol{X})^{-1} \boldsymbol{X}^\top \boldsymbol{Y} \rangle - \langle \boldsymbol{x}, \boldsymbol{\theta}_\star \rangle)^2]$$
$$=: T_1 + T_2,$$

where

$$T_1 := \mathbb{E}[(\langle \boldsymbol{x}, (\boldsymbol{X}^\top \boldsymbol{X})^{-1} \boldsymbol{X}^\top \mathsf{p}(\boldsymbol{X}\boldsymbol{\theta}_\star)\rangle - \langle \boldsymbol{x}, \boldsymbol{\theta}_\star \rangle)^2]$$
$$= \mathbb{E}[\langle \boldsymbol{x}, (\boldsymbol{X}^\top \boldsymbol{X})^{-1} \boldsymbol{X}^\top [\mathsf{p}(\boldsymbol{X}\boldsymbol{\theta}_\star) - \boldsymbol{X}\boldsymbol{\theta}_\star]\rangle^2]$$
$$\leqslant \mathbb{E}[\|\boldsymbol{X}(\boldsymbol{X}^\top \boldsymbol{X})^{-1}\Sigma(\boldsymbol{X}^\top \boldsymbol{X})^{-1}\boldsymbol{X}^\top\|_{\mathsf{op}} \cdot \|[\mathsf{p}(\boldsymbol{X}\boldsymbol{\theta}_\star) - \boldsymbol{X}\boldsymbol{\theta}_\star]\|_2^2]$$
$$\overset{(i)}{\leqslant} \sqrt{\mathbb{E}[\mathrm{trace}((\boldsymbol{X}^\top \boldsymbol{X})^{-1}\Sigma(\boldsymbol{X}^\top \boldsymbol{X})^{-1}\Sigma)]} \cdot \sqrt{\mathbb{E}[\|[\mathsf{p}(\boldsymbol{X}\boldsymbol{\theta}_\star) - \boldsymbol{X}\boldsymbol{\theta}_\star]\|_2^4]} \overset{(ii)}{\leqslant} \frac{1}{m^2} \leqslant \frac{\bar{\sigma}^2}{m^2}$$

and

$$T_2 := \mathbb{E}[(\langle \boldsymbol{x}, (\boldsymbol{X}^\top \boldsymbol{X})^{-1} \boldsymbol{X}^\top \mathcal{E}\rangle)^2]$$
$$= \bar{\sigma}^2 \mathbb{E}[\mathrm{trace}((\boldsymbol{X}^\top \boldsymbol{X})^{-1}\Sigma)] \overset{(iii)}{=} \bar{\sigma}^2 \frac{d}{m-d-1}.$$

Here, step (i) uses Cauchy-Schwarz inequality, step (ii) and (iii) follow from claim (35) and (36) and the choice of $B$. Combining the bounds on $T_1, T_2$ yields the upper bound $\mathbb{E}[\mathsf{R}_{\mathsf{lin}}(\widetilde{\mathsf{h}}_{\mathsf{ols}})] - \bar{\sigma}^2 \leqslant c\bar{\sigma}^2 \frac{d}{m-d-1}$.

To establish the lower bound, since $\mathbb{E}[a^2] \geqslant \mathbb{E}[b^2] + \mathbb{E}[(a-b)^2] - 2\sqrt{\mathbb{E}[(a-b)^2]} \cdot \sqrt{\mathbb{E}[b^2]}$, it follows that

$$
\begin{aligned}
\mathbb{E}[\mathsf{R}_{\mathsf{lin}}(\widetilde{\mathsf{h}}_{\mathsf{ols}})] - \bar{\sigma}^2 &= \mathbb{E}[(\mathsf{p}(\langle \boldsymbol{x}, (\boldsymbol{X}^\top \boldsymbol{X})^{-1} \boldsymbol{X}^\top \boldsymbol{Y}\rangle) - \mathsf{p}(\langle \boldsymbol{x}, \boldsymbol{\theta}_\star \rangle))^2] \\
&= \mathbb{E}[(\mathsf{p}(\langle \boldsymbol{x}, \boldsymbol{\theta}_\star + (\boldsymbol{X}^\top \boldsymbol{X})^{-1} \boldsymbol{X}^\top \mathcal{E}\rangle) - \mathsf{p}(\langle \boldsymbol{x}, \boldsymbol{\theta}_\star \rangle))^2] \\
&\geqslant T_3 - (T_4 + T_5),
\end{aligned}
$$

where

$$
T_3 = \mathbb{E}[(\langle \boldsymbol{x}, \boldsymbol{\theta}_\star + (\boldsymbol{X}^\top \boldsymbol{X})^{-1} \boldsymbol{X}^\top \mathcal{E}\rangle - \langle \boldsymbol{x}, \boldsymbol{\theta}_\star \rangle))^2] = \mathbb{E}[\mathrm{trace}((\boldsymbol{X}^\top \boldsymbol{X})^{-1}\Sigma)] = \bar{\sigma}^2 \frac{d}{m-d-1}.
$$

$T_4 = 2\sqrt{T_3}\sqrt{T_5}$,

$$
T_5 := \mathbb{E}[[(\mathsf{p}(\langle \boldsymbol{x}, \boldsymbol{\theta}_\star + (\boldsymbol{X}^\top \boldsymbol{X})^{-1} \boldsymbol{X}^\top \mathcal{E}\rangle) - \langle \boldsymbol{x}, \boldsymbol{\theta}_\star + (\boldsymbol{X}^\top \boldsymbol{X})^{-1} \boldsymbol{X}^\top \mathcal{E}\rangle) - (\mathsf{p}(\langle \boldsymbol{x}, \boldsymbol{\theta}_\star \rangle) - \langle \boldsymbol{x}, \boldsymbol{\theta}_\star \rangle)]^2]
$$

$$
\leqslant \mathbb{E}[[\mathsf{p}(\langle \boldsymbol{x}, \boldsymbol{\theta}_\star + (\boldsymbol{X}^\top \boldsymbol{X})^{-1} \boldsymbol{X}^\top \mathcal{E}\rangle) - \langle \boldsymbol{x}, \boldsymbol{\theta}_\star + (\boldsymbol{X}^\top \boldsymbol{X})^{-1} \boldsymbol{X}^\top \mathcal{E}\rangle]^2] + \bar{\sigma}^2 \frac{1}{m^2},
$$

where the inequality uses claim (36). To find a further upper bound of $T_4, T_5$, we first note that $(\boldsymbol{\theta}_\star + (\boldsymbol{X}^\top \boldsymbol{X})^{-1} \boldsymbol{X}^\top \mathcal{E})$ is independent of $\boldsymbol{x}$, and

$$
\|\boldsymbol{\theta}_\star + (\boldsymbol{X}^\top \boldsymbol{X})^{-1} \boldsymbol{X}^\top \mathcal{E}\|_2^2 \leqslant 2B_{\boldsymbol{\theta}}^2 + 2\|(\boldsymbol{X}^\top \boldsymbol{X})^{-1} \boldsymbol{X}^\top \mathcal{E}\|_2^2 \leqslant c\bar{\sigma}^2 \frac{d}{m} + v,
$$

where $v$ is some zero-mean $c\bar{\sigma}^2$-sub-Exponential variable by Theorem 1 in [16]. Under our choice of $B$, following the proof of claim (36) and integrating over the sub-Exponential variable $v$, it can be verified that (when choosing the absolute constant in $B$ sufficiently large) $T_5 \leqslant 2\bar{\sigma}^2/m^2$. Putting the bounds on $T_3, T_5$ (and hence $T_4$) together, we conclude that $\mathbb{E}[\mathsf{R}_{\mathsf{lin}}(\widetilde{\mathsf{h}}_{\mathsf{ols}})] - \bar{\sigma}^2 \geqslant c\bar{\sigma}^2 \frac{d}{m-d-1}$ for some absolute constant $c > 0$.

**Proof of claim** (35) **and** (36). Claim (35) follows directly from properties of the inverse Wishart distribution [43]. For Claim (36), since each coordinate of $\boldsymbol{X}\boldsymbol{\theta}_\star$ are i.i.d. $\mathcal{N}(0, \|\boldsymbol{\theta}_\star\|_2^2)$, w.l.o.g., it suffice to show

$$
\mathbb{E}[|\mathsf{p}(z) - z|^4] \leqslant c\exp(-B^2/c).
$$

for $z \sim \mathcal{N}(0, 1)$. Note that this follows immediately since

$$
\mathbb{E}[|\mathsf{p}(z) - z|^4] \leqslant c\int_B^\infty s^4 \exp(-s^2/2)ds \leqslant cs^3 \exp(-s^2/2) \leqslant c\exp(-s^2/c).
$$

$\square$

## D.5 Proof of Theorem 7

We prove Eq. (14) and (15) in Appendix D.5.1 and D.5.2, respectively.

### D.5.1 Proof of Eq. (14)

It suffices to apply Theorem 4 to the setup in Theorem 7. With a slight abuse of notation, we use both one-hot vectors in $\cup_{i=1}^S \{e_i\}$ and integers in $[S]$ to represent the augmented views $\boldsymbol{z}$ and do not distinguish them in the proof. We also occasionally omit the subscripts in $\mathbb{P}_{\mathcal{Y}}, \mathbb{P}_c$ when the meaning is clear from the context.

We claim that

$$
\frac{\mathbb{P}(\boldsymbol{z}^{(1)}, \boldsymbol{z}^{(2)})}{\mathbb{P}(\boldsymbol{z}^{(1)})\mathbb{P}(\boldsymbol{z}^{(2)})} = \frac{1}{2} \cdot \sum_{y=1}^M \frac{\mathbb{P}_c(y|\boldsymbol{z}^{(1)}) \cdot \mathbb{P}_c(y|\boldsymbol{z}^{(2)})}{\mathbb{P}_{\mathcal{Y}}(y)} + \frac{S}{2} \cdot \mathbb{1}_{\{\boldsymbol{z}^{(1)} = \boldsymbol{z}^{(2)}\}}. \tag{37}
$$

We will prove this claim momentarily. With this claim at hand, we have

**Approximation error.** Let

$$f_\star(z) := \frac{1}{\sqrt{2}}\Big(\frac{\mathbb{P}_c(y = 1|x^{c_1} = z)}{\sqrt{\mathbb{P}_\mathcal{Y}(y = 1)}}, \ldots, \frac{\mathbb{P}_c(y = M|x^{c_1} = z)}{\sqrt{\mathbb{P}_\mathcal{Y}(y = M)}}, \sqrt{S}z^\top\Big)^\top.$$

It can be verified that the parameter $(W_\star, w_\star)$ corresponding to $f_\star$ lies in $\Gamma$. Therefore, the approximation error $\inf_{f\in\mathcal{F}} R_{\chi^2}(\mathsf{S}_f) - R_{\chi^2}(\mathsf{S}_\star) = 0$ since $\mathsf{S}_\star$ is realized by $f_\star$ and the link function $\tau(x) = x$.

**Generalization error.** Let $\mathcal{W} := \{W \in \mathbb{R}^{M\times S}, w \in \mathbb{R}, \|W\|_{2,\infty} \vee |w/\sqrt{S}| \leqslant B_W\}$ and define the metric $\|(W_1, w_1) - (W_2, w_2)\| := \|W_1 - W_2\|_{2,\infty} \vee |(w_1 - w_2)/\sqrt{S}|$ on $\mathcal{W}$.

First, for $f_i(z) = ((W_i z)^\top, w_i \cdot z^\top)^\top$ $(i = 1, 2)$, simple calculation shows $\|f_1 - f_2\|_{2,\infty} \leqslant 2(|w_1 - w_1| \vee \|W_1 - W_2\|_{\mathrm{op}})$, and therefore

$$\log\mathcal{N}(u, \|\cdot\|_{2,\infty}, \mathcal{F}) \leqslant \log\mathcal{N}\Big(\frac{u}{2}, \|\cdot\|, \mathcal{W}\Big) \leqslant \log\mathcal{N}\Big(\frac{u}{2}, \|\cdot\|_{2,\infty}, \mathcal{W}_1\Big) + \log\mathcal{N}\Big(\frac{u\sqrt{S}}{2}, |\cdot|, \mathcal{W}_2\Big)$$

$$\leqslant SM \cdot \log\Big(1 + \frac{4B_W}{u}\Big) + \log\Big(1 + \frac{4B_W}{u}\Big),$$

where $\mathcal{W}_1 := \{W \in \mathbb{R}^{M\times S}, \|W\|_{2,\infty} \leqslant B_W\}, \mathcal{W}_2 := \{w \in \mathbb{R}, |w| \leqslant \sqrt{S}B_W\}$ and the last inequality follows from the upper bound of the covering number of the unit ball (see e.g., Example 5.8 in [44]) and the assumption that $M \leqslant S$. In addition, it is readily verified that $\mathsf{S}_f(z^{(1)}, z^{(2)}) \in [-\bar{B}_{\mathsf{S}}, \bar{B}_{\mathsf{S}}]$ with $\bar{B}_{\mathsf{S}} = 4B_W^2 S = 4M^2 S$ for all $z^{(1)}, z^{(2)}$. Consequently,

$$B_\tau \int_0^{B_W} \sqrt{\log\mathcal{N}(u, \|\cdot\|_{2,\infty}, \mathcal{F})}du$$

$$\leqslant c\Big(\int_0^{B_W}\sqrt{SM\cdot\log\Big(1 + \frac{4B_W}{u}\Big)}du + \int_0^{B_W}\sqrt{\log\Big(1 + \frac{4B_W}{u}\Big)}du\Big)$$

$$\leqslant c\sqrt{SM}B_W = c\sqrt{SM^3}.$$

Combining the result on the approximation error and the generalization error and applying Theorem 4 yields the desired result.

Proof of claim (37). For $z^{(1)} \neq z^{(2)}$, by Bayes' formula, we have

$$\frac{\mathbb{P}(z^{(1)}, z^{(2)})}{\mathbb{P}(z^{(1)})\mathbb{P}(z^{(2)})} = \sum_x \frac{\mathbb{P}(z^{(2)}|x) \cdot \mathbb{P}(x|z^{(1)})}{\mathbb{P}(z^{(2)})} = \sum_x \frac{\mathbb{P}(x|z^{(2)}) \cdot \mathbb{P}(x|z^{(1)})}{\mathbb{P}(x)}$$

$$\overset{(i)}{=} 2\frac{\mathbb{P}(x = (z^{(1)}, z^{(2)})|z^{(2)}) \cdot \mathbb{P}(x = (z^{(1)}, z^{(2)})|z^{(1)})}{\mathbb{P}(x = (z^{(1)}, z^{(2)}))}, \tag{38}$$

where step (i) follows from symmetry between $z^{(1)}, z^{(2)}$. Moreover,

$$\mathbb{P}(x = (z^{(1)}, z^{(2)})|z^{(1)}) = \frac{1}{2}\mathbb{P}(x^{c_2} = z^{(2)}|x^{c_1} = z^{(1)}) = \frac{1}{2}\sum_{y=1}^M \mathbb{P}_c(x^{c_2} = z^{(2)}|y) \cdot \mathbb{P}_c(y|x^{c_1} = z^{(1)})$$

$$= \frac{1}{2}\sum_{y=1}^M \frac{\mathbb{P}_c(y|x^{c_2} = z^{(2)}) \cdot \mathbb{P}_c(y|x^{c_1} = z^{(1)})}{\mathbb{P}_\mathcal{Y}(y)} \cdot \mathbb{P}_c(x^{c_2} = z^{(2)})$$

$$= \frac{1}{2}\sum_{y=1}^M \frac{\mathbb{P}_c(y|z^{(2)}) \cdot \mathbb{P}_c(y|z^{(1)})}{\mathbb{P}_\mathcal{Y}(y)} \cdot \mathbb{P}_c(z^{(2)}), \tag{39a}$$

$$\mathbb{P}_c(z) \overset{(ii)}{=} \mathbb{P}(z), \quad \text{and} \tag{39b}$$

$$\mathbb{P}(z^{(1)}, z^{(2)}) \overset{(iii)}{=} 2\mathbb{P}((z^{(1)}, z^{(2)}), x = (z^{(1)}, z^{(2)})) = \frac{1}{2}\mathbb{P}(x = (z^{(1)}, z^{(2)})), \tag{39c}$$

where step (ii) follows from the generation process of the augmented views $(\boldsymbol{z}^{(1)}, \boldsymbol{z}^{(2)})$, and step (iii) follows from symmetry between $\boldsymbol{z}^{(1)}, \boldsymbol{z}^{(2)}$. Substituting Eq. (39a) into Eq. (38), we find

$$
\begin{aligned}
\frac{\mathbb{P}(\boldsymbol{z}^{(1)}, \boldsymbol{z}^{(2)})}{\mathbb{P}(\boldsymbol{z}^{(1)})\mathbb{P}(\boldsymbol{z}^{(2)})} &= \frac{1}{2}\Big(\sum_{y=1}^{M} \frac{\mathbb{P}_c(y|\boldsymbol{z}^{(2)}) \cdot \mathbb{P}_c(y|\boldsymbol{z}^{(1)})}{\mathbb{P}_{\mathcal{Y}}(y)}\Big)^2 \cdot \frac{\mathbb{P}_c(\boldsymbol{z}^{(1)})\mathbb{P}_c(\boldsymbol{z}^{(2)})}{\mathbb{P}(\boldsymbol{x} = (\boldsymbol{z}^{(1)}, \boldsymbol{z}^{(2)}))} \\
&= \frac{1}{4}\Big(\sum_{y=1}^{M} \frac{\mathbb{P}_c(y|\boldsymbol{z}^{(2)}) \cdot \mathbb{P}_c(y|\boldsymbol{z}^{(1)})}{\mathbb{P}_{\mathcal{Y}}(y)}\Big)^2 \cdot \frac{\mathbb{P}(\boldsymbol{z}^{(1)})\mathbb{P}(\boldsymbol{z}^{(2)})}{\mathbb{P}(\boldsymbol{z}^{(1)}, \boldsymbol{z}^{(2)})},
\end{aligned} \tag{40}
$$

where the second equality uses Eq. (39b) and (39c). Reorganizing Eq. (40), we obtain

$$
\frac{\mathbb{P}(\boldsymbol{z}^{(1)}, \boldsymbol{z}^{(2)})}{\mathbb{P}(\boldsymbol{z}^{(1)})\mathbb{P}(\boldsymbol{z}^{(2)})} = \frac{1}{2}\Big(\sum_{y=1}^{M} \frac{\mathbb{P}_c(y|\boldsymbol{z}^{(2)}) \cdot \mathbb{P}_c(y|\boldsymbol{z}^{(1)})}{\mathbb{P}_{\mathcal{Y}}(y)}\Big) = \frac{1}{2}\frac{\mathbb{P}_c(\boldsymbol{z}^{(1)}, \boldsymbol{z}^{(2)})}{\mathbb{P}_c(\boldsymbol{z}^{(1)})\mathbb{P}_c(\boldsymbol{z}^{(2)})}, \tag{41}
$$

where we recall $\mathbb{P}_c(\cdot)$ is the marginal distribution of $\boldsymbol{x}^{c_1}$ (or $\boldsymbol{x}^{c_2}$) and the second equality follows from Bayes' formula and the fact that $\boldsymbol{x}^{c_1} \perp\!\!\!\perp \boldsymbol{x}^{c_2}|\boldsymbol{y}$.

For $\boldsymbol{z}^{(1)} = \boldsymbol{z}^{(2)} = z$, using Eq. (39b) and properties of conditional distribution, we have

$$
\sum_{z' \in [S]} \frac{\mathbb{P}_c(\boldsymbol{z}^{(1)} = z, \boldsymbol{z}^{(2)} = z')}{\mathbb{P}_c(\boldsymbol{z}^{(1)} = z)\mathbb{P}_c(\boldsymbol{z}^{(2)} = z')} = \frac{1}{\mathbb{P}_c(\boldsymbol{z}^{(2)} = z)} = \frac{1}{\mathbb{P}(\boldsymbol{z}^{(2)} = z)} = \sum_{z' \in [S]} \frac{\mathbb{P}(\boldsymbol{z}^{(1)} = z, \boldsymbol{z}^{(2)} = z')}{\mathbb{P}(\boldsymbol{z}^{(1)} = z)\mathbb{P}(\boldsymbol{z}^{(2)} = z')}.
$$

Combining this with Eq. (41) for all $\boldsymbol{z}^{(2)} \neq \boldsymbol{z}^{(1)}$ and noting that the marginal $\mathbb{P}_c(\cdot)$ is the uniform distribution on $[S]$, we obtain

$$
\begin{aligned}
\frac{\mathbb{P}(\boldsymbol{z}^{(1)} = z, \boldsymbol{z}^{(2)} = z)}{\mathbb{P}(\boldsymbol{z}^{(1)} = z)\mathbb{P}(\boldsymbol{z}^{(2)} = z)} &= \frac{1}{2} \cdot \frac{\mathbb{P}_c(\boldsymbol{x}^{c_1} = z, \boldsymbol{x}^{c_2} = z)}{\mathbb{P}_c(\boldsymbol{x}^{c_1} = z)\mathbb{P}_c(\boldsymbol{x}^{c_2} = z)} + \frac{1}{2}\sum_{z' \in [S]} \frac{\mathbb{P}_c(\boldsymbol{x}^{c_1} = z, \boldsymbol{x}^{c_2} = z')}{\mathbb{P}_c(\boldsymbol{x}^{c_1} = z)\mathbb{P}_c(\boldsymbol{x}^{c_2} = z')} \\
&= \frac{1}{2} \cdot \frac{\mathbb{P}_c(\boldsymbol{x}^{c_1} = z, \boldsymbol{x}^{c_2} = z)}{\mathbb{P}_c(\boldsymbol{x}^{c_1} = z)\mathbb{P}_c(\boldsymbol{x}^{c_2} = z)} + \frac{S}{2} \\
&= \frac{1}{2} \cdot \sum_{y=1}^{M} \frac{\mathbb{P}_c(y|z) \cdot \mathbb{P}_c(y|z)}{\mathbb{P}_{\mathcal{Y}}(y)} + \frac{S}{2}.
\end{aligned}
$$

### D.5.2 Proof of Eq. (15)

Write $\boldsymbol{z} = g(\boldsymbol{x})$. By a standard risk decomposition, we have

$$
\begin{aligned}
\mathsf{R}_{\mathrm{cls}}(\mathsf{h}_{\widehat{\boldsymbol{\Gamma}}}) &= \mathbb{E}[\widehat{\mathsf{R}}_{\mathrm{cls}}(\mathsf{h}_{\widehat{\boldsymbol{\Gamma}}})] - \mathbb{E}[\widehat{\mathsf{R}}_{\mathrm{cls}}(\mathbb{P}_{\boldsymbol{y}|\boldsymbol{x}}(\cdot|\boldsymbol{x}))] \\
&= \mathbb{E}[\widehat{\mathsf{R}}_{\mathrm{cls}}(\mathsf{h}_{\widehat{\boldsymbol{\Gamma}}})] - \inf_{\mathsf{h}} \mathbb{E}[\widehat{\mathsf{R}}_{\mathrm{cls}}(\mathsf{h})] \\
&= \underbrace{\inf_{\boldsymbol{\Gamma}: \|\boldsymbol{\Gamma}_w\|_{\mathrm{op}} \vee \|\boldsymbol{\Gamma}_b\|_2 \leqslant B_{\boldsymbol{\Gamma}}} \mathbb{E}_{\boldsymbol{x},g}[\mathsf{D}_{\mathrm{KL}}(\mathbb{P}_{\boldsymbol{y}|\boldsymbol{x}}(\cdot|\boldsymbol{x}) \| \bar{\mathsf{h}}_{\boldsymbol{\Gamma}}(f(\boldsymbol{z})))]}_{\text{approximation error}} \\
&\quad + \underbrace{\mathbb{E}[\widehat{\mathsf{R}}_{\mathrm{cls}}(\mathsf{h}_{\widehat{\boldsymbol{\Gamma}}})] - \inf_{\boldsymbol{\Gamma}: \|\boldsymbol{\Gamma}_w\|_{\mathrm{op}} \vee \|\boldsymbol{\Gamma}_b\|_2 \leqslant B_{\boldsymbol{\Gamma}}} \mathbb{E}[\widehat{\mathsf{R}}_{\mathrm{cls}}(\mathsf{h}_{\boldsymbol{\Gamma}})]}_{\text{generalization error}}.
\end{aligned}
$$

We will prove that for some absolute constant $c > 0$,

1.

$$
\text{approximation error} \leqslant c\Big(\epsilon_{\mathcal{G}}^{\mathsf{cls}} + \frac{S\exp(B)}{\sigma_{\boldsymbol{E}_\star}^2} \cdot (R_{\mathsf{f}}(\mathsf{S}_{\widehat{f}_{\mathsf{aug}}}) - R_{\mathsf{f}}(\mathsf{S}_\star))\Big), \tag{42a}
$$

and

2. with probability at least $1 - \delta$,

$$
\text{generalization error} \leqslant \frac{cB}{\sqrt{m}}\Big[\sqrt{\log(1/\delta)} + M(\sqrt{\log B_{\boldsymbol{\Gamma}}} + \sqrt{B})\Big]. \tag{42b}
$$

**Approximation error.** Let $\boldsymbol{E}_\star \in \mathbb{R}^{M \times S}$ be the representation where

$$\boldsymbol{E}_{\star,\cdot j} = \frac{1}{\sqrt{2}}\Big(\frac{\mathbb{P}_c(\boldsymbol{y}=1|\boldsymbol{x}^{c_1}=j)}{\sqrt{\mathbb{P}_{\mathcal{Y}}(\boldsymbol{y}=1)}},\ldots,\frac{\mathbb{P}_c(\boldsymbol{y}=M|\boldsymbol{x}^{c_1}=j)}{\sqrt{\mathbb{P}_{\mathcal{Y}}(\boldsymbol{y}=M)}}\Big)^\top$$

for $j \in [S]$ and let $\boldsymbol{E}_\star(\boldsymbol{z})$ denote the $\boldsymbol{z}$-th column of $\boldsymbol{E}_\star$. Let $\widehat{\boldsymbol{E}} := \big(\widehat{f}(1)\ \cdots\ \widehat{f}(S)\big) \in \mathbb{R}^{M \times S}$.

Given a representation $\widehat{f}(\boldsymbol{z})$, consider the classifier

$$\bar{\mathsf{h}}_{\boldsymbol{\Gamma}}(\widehat{f}(\boldsymbol{z})) = \mathrm{softmax}(\log\mathrm{trun}(\boldsymbol{\Gamma}_w\widehat{f}(\boldsymbol{z})+\boldsymbol{\Gamma}_b)),\quad\text{where}$$

$$\boldsymbol{\Gamma}_w := \sqrt{2}\boldsymbol{P}_{\mathcal{Y}}^{1/2}(\boldsymbol{E}_\star\boldsymbol{E}_\star^\top)^{-1}\boldsymbol{E}_\star(\mathrm{I}_S - \mathcal{P}_{\boldsymbol{1}_S})\widehat{\boldsymbol{E}}^\top,\quad \boldsymbol{\Gamma}_b := \frac{1}{\sqrt{2}}\boldsymbol{P}_{\mathcal{Y}}^{1/2}(\boldsymbol{E}_\star\boldsymbol{E}_\star^\top)^{-1}\boldsymbol{E}_\star\boldsymbol{1}_S,$$

$$\tag{43}$$

and $\boldsymbol{P}_{\mathcal{Y}} := \mathrm{diag}\{\mathbb{P}_{\mathcal{Y}}(\boldsymbol{y}=1),\ldots,\mathbb{P}_{\mathcal{Y}}(\boldsymbol{y}=M)\}$. It can be verified that $\|\boldsymbol{\Gamma}_w\|_{\mathrm{op}} \leqslant 2\sqrt{S}M/\sigma_{\boldsymbol{E}_\star} \leqslant B_\Gamma$ and $\|\boldsymbol{\Gamma}_b\|_2 \leqslant \sqrt{S}/\sigma_{\boldsymbol{E}_\star} \leqslant B_\Gamma$. Moreover, we have by Lemma 5 that

$$\mathbb{E}_{\boldsymbol{x},g}[\mathsf{D}_{\mathrm{KL}}(\mathbb{P}_{\boldsymbol{y}|\boldsymbol{x}}(\boldsymbol{y}|\boldsymbol{x})||\bar{\mathsf{h}}_{\boldsymbol{\Gamma}}(\widehat{f}(\boldsymbol{z})))] \leqslant 2\mathbb{E}_{\boldsymbol{x},g}[\mathsf{D}_{\mathrm{KL}}(\mathbb{P}_{\boldsymbol{y}|\boldsymbol{x}}(\cdot|\boldsymbol{x})||\mathbb{P}_{\boldsymbol{y}|\boldsymbol{z}}(\cdot|\boldsymbol{z})] + \mathbb{E}_{\boldsymbol{x},g}[\mathsf{D}_2(\mathbb{P}_{\boldsymbol{y}|\boldsymbol{z}}(\cdot|\boldsymbol{z})||\bar{\mathsf{h}}_{\boldsymbol{\Gamma}}(\widehat{f}(\boldsymbol{z})))]$$

$$\leqslant 2\epsilon_{\mathcal{G}}^{\mathsf{cls}} + \mathbb{E}_{\boldsymbol{x},g}[\mathsf{D}_2(\mathbb{P}_{\boldsymbol{y}|\boldsymbol{z}}(\cdot|\boldsymbol{z})||\bar{\mathsf{h}}_{\boldsymbol{\Gamma}}(\widehat{f}(\boldsymbol{z})))].$$

Therefore, it remains to prove

$$\mathbb{E}_{\boldsymbol{x},g}[\mathsf{D}_2(\mathbb{P}_{\boldsymbol{y}|\boldsymbol{z}}(\cdot|\boldsymbol{z})||\bar{\mathsf{h}}_{\boldsymbol{\Gamma}}(\widehat{f}(\boldsymbol{z})))] \leqslant \frac{cS\exp(B)}{\sigma_{\boldsymbol{E}_\star}^2}\cdot(R_{\mathrm{f}}(\mathsf{S}_{\widehat{f}_{\mathsf{aug}}}) - R_{\mathrm{f}}(\mathsf{S}_\star)). \tag{44}$$

Since

$$\mathbb{E}_{\boldsymbol{x},g}[\mathsf{D}_2(\mathbb{P}_{\boldsymbol{y}|\boldsymbol{z}}(\cdot|\boldsymbol{z})||\bar{\mathsf{h}}_{\boldsymbol{\Gamma}}(\widehat{f}(\boldsymbol{z})))] \leqslant \mathbb{E}_{\boldsymbol{x},g}\Big[\mathbb{E}_{y\sim\mathbb{P}_{y|\boldsymbol{z}}(\cdot|\boldsymbol{z})}\frac{\mathbb{P}_{\boldsymbol{y}|\boldsymbol{z}}(y|\boldsymbol{z}) - \bar{\mathsf{h}}_{\boldsymbol{\Gamma}}(\widehat{f}(\boldsymbol{z}))_y}{\bar{\mathsf{h}}_{\boldsymbol{\Gamma}}(\widehat{f}(\boldsymbol{z}))_y}\Big]$$

$$= \mathbb{E}_{\boldsymbol{x},g}\Big[\sum_{y\in[M]}\frac{(\mathbb{P}_{\boldsymbol{y}|\boldsymbol{z}}(y|\boldsymbol{z}) - \bar{\mathsf{h}}_{\boldsymbol{\Gamma}}(\widehat{f}(\boldsymbol{z}))_y)^2}{\bar{\mathsf{h}}_{\boldsymbol{\Gamma}}(\widehat{f}(\boldsymbol{z}))_y}\Big]$$

$$\leqslant \exp(B)\cdot\mathbb{E}_{\boldsymbol{x},g}\Big[\sum_{y\in[M]}(\mathbb{P}_{\boldsymbol{y}|\boldsymbol{z}}(y|\boldsymbol{z}) - \bar{\mathsf{h}}_{\boldsymbol{\Gamma}}(\widehat{f}(\boldsymbol{z}))_y)^2\Big]$$

$$= \exp(B)\cdot\mathbb{E}_{\boldsymbol{x},g}\Big[\sum_{y\in[M]}(\mathbb{P}_c(y|\boldsymbol{x}^{c_1}=\boldsymbol{z}) - \bar{\mathsf{h}}_{\boldsymbol{\Gamma}}(\widehat{f}(\boldsymbol{z}))_y)^2\Big], \tag{45}$$

where the third line uses the definition of $\mathrm{trun}$ and claim (46) in the proof of Lemma 6, and the last line uses the fact that $\mathbb{P}_c(\boldsymbol{y}=y|\boldsymbol{x}^{c_1}=j) = \mathbb{P}_{\boldsymbol{y}|\boldsymbol{z}}(\boldsymbol{y}=y|\boldsymbol{z}=j)$ for all $y\in[M], j\in[S]$. Eq. (44) follows immediately from Lemma 6 which gives an upper bound on the term in Eq. (45).

**Generalization error.** The proof follows from a standard analysis of empirical process similar to the proof of Eq. (29) in the proof of Theorem 4. Thus, we only provide a sketch of the proof here.

Let $\Gamma := \{\boldsymbol{\Gamma} : \|\boldsymbol{\Gamma}_w\|_{\mathrm{op}} \vee \|\boldsymbol{\Gamma}_b\|_2 \leqslant B_\Gamma\}$ and define the norm $\|\boldsymbol{\Gamma} - \widetilde{\boldsymbol{\Gamma}}\| := \|\boldsymbol{\Gamma}_w - \widetilde{\boldsymbol{\Gamma}}_w\|_{\mathrm{op}} \vee \|\boldsymbol{\Gamma}_b - \widetilde{\boldsymbol{\Gamma}}_b\|_2$. First, by a triangle inequality, the fact that $\|\log\mathsf{h}_{\boldsymbol{\Gamma}}\|_\infty \leqslant 2B$ (which follows from the definition of $\mathrm{trun}$), and Corollary 2.21 in Wainwright [44] for functions with bounded differences, we have

$$\text{generalization error} \leqslant 2\mathbb{E}\Big[\sup_{\boldsymbol{\Gamma}\in\Gamma}|\widehat{\mathsf{R}}_{\mathrm{cls}}(\mathsf{h}_{\boldsymbol{\Gamma}}) - \mathbb{E}[\widehat{\mathsf{R}}_{\mathrm{cls}}(\mathsf{h}_{\boldsymbol{\Gamma}})]|\Big] + 2B\frac{\sqrt{\log(1/\delta)}}{\sqrt{m}}$$

with probability at least $1-\delta$. Let $X_{\boldsymbol{\Gamma}} := \widehat{\mathsf{R}}_{\mathrm{cls}}(\mathsf{h}_{\boldsymbol{\Gamma}}) - \mathbb{E}[\widehat{\mathsf{R}}_{\mathrm{cls}}(\mathsf{h}_{\boldsymbol{\Gamma}})]$. Then we have

$$\mathbb{E}\Big[\sup_{\boldsymbol{\Gamma}\in\Gamma}|\widehat{\mathsf{R}}_{\mathrm{cls}}(\mathsf{h}_{\boldsymbol{\Gamma}}) - \mathbb{E}[\widehat{\mathsf{R}}_{\mathrm{cls}}(\mathsf{h}_{\boldsymbol{\Gamma}})]|\Big] \leqslant \mathbb{E}[|X_{\boldsymbol{\Gamma}_0}|] + \mathbb{E}[\sup_{\boldsymbol{\Gamma},\widetilde{\boldsymbol{\Gamma}}\in\Gamma}|X_{\boldsymbol{\Gamma}} - X_{\widetilde{\boldsymbol{\Gamma}}}|] \leqslant \frac{2B}{\sqrt{m}} + \mathbb{E}[\sup_{\boldsymbol{\Gamma},\widetilde{\boldsymbol{\Gamma}}\in\Gamma}|X_{\boldsymbol{\Gamma}} - X_{\widetilde{\boldsymbol{\Gamma}}}|].$$

Moreover, the process $\{X_{\boldsymbol{\Gamma}}\}_{\boldsymbol{\Gamma}\in\Gamma}$ is a zero-mean sub-Gaussian process with respect to the metric $\rho_X(\boldsymbol{\Gamma},\widetilde{\boldsymbol{\Gamma}}) := 2\|\log\bar{\mathsf{h}}_{\boldsymbol{\Gamma}} - \log\bar{\mathsf{h}}_{\widetilde{\boldsymbol{\Gamma}}}\|_\infty/\sqrt{m}$ since $X_{\boldsymbol{\Gamma}}$ is the average of i.i.d. random variables bounded by

$$2\sup_{i\in[m]}|\langle e_{\boldsymbol{y}_i},\log\bar{\mathsf{h}}_{\boldsymbol{\Gamma}}(\widehat{f}(\boldsymbol{z}_i))\rangle - \langle e_{\boldsymbol{y}_i},\log\bar{\mathsf{h}}_{\widetilde{\boldsymbol{\Gamma}}}(\widehat{f}(\boldsymbol{z}_i))\rangle|$$

$$\leqslant 2\| \log \bar{\mathsf{h}}_{\boldsymbol{\Gamma}}(\widehat{f}(\boldsymbol{z}_i)) - \log \bar{\mathsf{h}}_{\widetilde{\boldsymbol{\Gamma}}}(\widehat{f}(\boldsymbol{z}_i))\|_\infty \leqslant \rho_X(\boldsymbol{\Gamma}, \widetilde{\boldsymbol{\Gamma}}) \cdot \sqrt{m}, \text{ and moreover}$$

$$\rho_X(\boldsymbol{\Gamma}, \widetilde{\boldsymbol{\Gamma}}) \overset{(i)}{\leqslant} c\| \log \mathsf{trun}(\boldsymbol{\Gamma}_w \widehat{f}(\boldsymbol{z}) + \boldsymbol{\Gamma}_b) - \log \mathsf{trun}(\widetilde{\boldsymbol{\Gamma}}_w \widehat{f}(\boldsymbol{z}) + \widetilde{\boldsymbol{\Gamma}}_b)\|_\infty / \sqrt{m},$$

$$\overset{(ii)}{\leqslant} c \exp(B) \cdot \|\boldsymbol{\Gamma} - \widetilde{\boldsymbol{\Gamma}}\| / \sqrt{m} =: \bar{B}\|\boldsymbol{\Gamma} - \widetilde{\boldsymbol{\Gamma}}\| / \sqrt{m},$$

where step (i) uses $\|\log \mathrm{softmax}(\boldsymbol{u}) - \log \mathrm{softmax}(\boldsymbol{v})\|_\infty \leqslant 2\|\boldsymbol{u} - \boldsymbol{v}\|_\infty$ and step (ii) follows from Taylor expansion of $s(x) = \log x$, the assumption that $\|\widehat{f}(\boldsymbol{z})\|_2 \leqslant B_{\boldsymbol{W}} = M$. Therefore, we have by Dudley's integral bound (see e.g., Theorem 5.22 in Wainwright [44]) that

$$\mathbb{E}[\sup_{\boldsymbol{\Gamma}, \widetilde{\boldsymbol{\Gamma}} \in \Gamma} |X_{\boldsymbol{\Gamma}} - X_{\widehat{\boldsymbol{\Gamma}}}|] \leqslant c \int_0^{cB/\sqrt{m}} \sqrt{\log \mathcal{N}(u, \rho_X, \{X_{\boldsymbol{\Gamma}}, \boldsymbol{\Gamma} \in \Gamma\})} du \leqslant c \int_0^{cB/\sqrt{m}} \sqrt{\log \mathcal{N}\left(u, \frac{\bar{B}}{\sqrt{m}}\|\cdot\|, \Gamma\right)} du$$

$$\leqslant c \int_0^{cB/\sqrt{m}} \sqrt{\log \mathcal{N}\left(\frac{\sqrt{m} \cdot u}{\bar{B}}, \|\cdot\|, \Gamma\right)} du$$

$$\leqslant c \int_0^{cB/\sqrt{m}} \left(\sqrt{\log \mathcal{N}\left(\frac{\sqrt{m} \cdot u}{\bar{B}}, \|\cdot\|_{\mathrm{op}}, \Gamma_w\right)} + \sqrt{\log \mathcal{N}\left(\frac{\sqrt{m} \cdot u}{\bar{B}}, \|\cdot\|_2, \Gamma_b\right)}\right) du$$

$$\leqslant c \int_0^{cB/\sqrt{m}} \sqrt{M^2 \cdot \log\left(1 + 4\frac{B_\Gamma \bar{B}}{\sqrt{m}u}\right)} du \leqslant c \frac{BM \log^{1/2}(B_\Gamma \bar{B})}{\sqrt{m}} \leqslant c \frac{BM(\log^{1/2} B_\Gamma + \sqrt{B})}{\sqrt{m}},$$

where $\Gamma_w := \{\boldsymbol{\Gamma}_w \in \mathbb{R}^{M \times M} : \|\boldsymbol{\Gamma}_w\|_{\mathrm{op}} \leqslant B_\Gamma\}$ and $\Gamma_b := \{\boldsymbol{\Gamma}_b \in \mathbb{R}^M : \|\boldsymbol{\Gamma}_b\|_2 \leqslant B_\Gamma\}$, and the last line uses the covering number bound of unit balls. Putting pieces together yields the desired bound.

### D.6 An auxiliary lemma

**Lemma 6** (Upper bound on the term in Eq. (45))**.** *Let the assumptions in Theorem 3 and the notations in its proof in Appendix D.5.2 hold. Assume $R_{\mathrm{f}}(\mathsf{S}_{\widehat{f}_{\mathrm{aug}}}) - R_{\mathrm{f}}(\mathsf{S}_\star) \leqslant c\sigma_{\boldsymbol{E}_\star}^2/(S^2 M)$ for some absolute constant $c > 0$, then*

$$\mathbb{E}_{\boldsymbol{x}, g}\left[\sum_{y \in [M]} (\mathbb{P}_c(y|\boldsymbol{x}^{c_1} = \boldsymbol{z}) - \bar{\mathsf{h}}_{\boldsymbol{\Gamma}}(\widehat{f}(\boldsymbol{z}))_y)^2\right] \leqslant \frac{c'S}{\sigma_{\boldsymbol{E}_\star}^2} \cdot (R_{\mathrm{f}}(\mathsf{S}_{\widehat{f}_{\mathrm{aug}}}) - R_{\mathrm{f}}(\mathsf{S}_\star))$$

*for some absolute constant $c' > 0$.*

*Proof of Lemma 6.* The proof consists of two steps. First, we plug the definition of $\bar{\mathsf{h}}_{\boldsymbol{\Gamma}}$ into Eq. (6) and simplify the expression. Then, we demonstrate that the simplified expression can be further bounded using the excess risk $R_{\mathrm{f}}(\mathsf{S}_{\widehat{f}_{\mathrm{aug}}}) - R_{\mathrm{f}}(\mathsf{S}_\star)$ of the learned encoder $\widehat{f}_{\mathrm{aug}}$.

Step 1: simplify the notation. Since

$$\|\nabla_{\boldsymbol{u}} \mathrm{softmax}(\log \boldsymbol{u})\|_{\mathrm{op}} = \|\frac{1}{\|\boldsymbol{u}\|_1} \mathrm{I}_M - \frac{\boldsymbol{u}}{\|\boldsymbol{u}\|_1} \mathbf{1}_M^\top\|_{\mathrm{op}} \leqslant \frac{1}{\|\boldsymbol{u}\|_1} + 1$$

for any $\boldsymbol{u} \in \mathbb{R}_{>0}^M$, we have

$$\mathbb{E}_{\boldsymbol{x}, g}\left[\sum_{y \in [M]} (\mathbb{P}_c(y|\boldsymbol{x}^{c_1} = \boldsymbol{z}) - \bar{\mathsf{h}}_{\boldsymbol{\Gamma}}(\widehat{f}(\boldsymbol{z}))_y)^2\right] \leqslant c\mathbb{E}_{\boldsymbol{x}, g}\left[\sum_{y \in [M]} (\mathbb{P}_c(y|\boldsymbol{x}^{c_1} = \boldsymbol{z}) - \mathsf{trun}(\boldsymbol{\Gamma}_w \widehat{f}(\boldsymbol{z}) + \boldsymbol{\Gamma}_b)_y)^2\right]$$

$$\leqslant c\mathbb{E}_{\boldsymbol{x}, g}\left[\sum_{y \in [M]} (\mathbb{P}_c(y|\boldsymbol{x}^{c_1} = \boldsymbol{z}) - (\boldsymbol{\Gamma}_w \widehat{f}(\boldsymbol{z}) + \boldsymbol{\Gamma}_b)_y)^2\right],$$

where in the first inequality we use the claim that

$$|1 - \|\mathsf{trun}(\boldsymbol{\Gamma}_w \widehat{f}(\boldsymbol{z}) + \boldsymbol{\Gamma}_b)\|_1| \leqslant 1/2. \tag{46}$$

The proof of this claim is deferred to the end of the proof of the lemma. The second inequality follows from a Taylor expansion of $s(x) = \log x$, the boundedness assumption that $\mathbb{P}_c(y|\boldsymbol{x}^{c_1} = \boldsymbol{z}) \in$

$[\exp(-B), 1]$, and noting the truncation $\mathsf{trun}(\cdot)$ reduces the $\ell_2$ error. Moreover, for any $\boldsymbol{z} \in [S]$, by the definition of $(\boldsymbol{\Gamma}_w, \boldsymbol{\Gamma}_b)$ in Eq. (43)

$$
\begin{aligned}
&\boldsymbol{\Gamma}_w \widehat{f}(\boldsymbol{z}) + \boldsymbol{\Gamma}_b \\
&= \sqrt{2} \boldsymbol{P}_{\mathcal{Y}}^{1/2} (\boldsymbol{E}_\star \boldsymbol{E}_\star^\top)^{-1} \boldsymbol{E}_\star [(\mathrm{I}_S - \mathcal{P}_{\mathbf{1}_S}) \widehat{\boldsymbol{E}}^\top \widehat{f}(\boldsymbol{z}) + \mathbf{1}_S/2] \\
&= \sqrt{2} \boldsymbol{P}_{\mathcal{Y}}^{1/2} (\boldsymbol{E}_\star \boldsymbol{E}_\star^\top)^{-1} \boldsymbol{E}_\star \boldsymbol{E}_\star^\top \boldsymbol{E}_\star(\boldsymbol{z}) \\
&\quad + \sqrt{2} \boldsymbol{P}_{\mathcal{Y}}^{1/2} (\boldsymbol{E}_\star \boldsymbol{E}_\star^\top)^{-1} \boldsymbol{E}_\star [(\mathrm{I}_S - \mathcal{P}_{\mathbf{1}_S}) \widehat{\boldsymbol{E}}^\top \widehat{f}(\boldsymbol{z}) + \mathbf{1}_S/2 - \boldsymbol{E}_\star^\top \boldsymbol{E}_\star(\boldsymbol{z})] \\
&= \sqrt{2} \boldsymbol{P}_{\mathcal{Y}}^{1/2} \boldsymbol{E}_\star(\boldsymbol{z}) + \sqrt{2} \boldsymbol{P}_{\mathcal{Y}}^{1/2} (\boldsymbol{E}_\star \boldsymbol{E}_\star^\top)^{-1} \boldsymbol{E}_\star [(\mathrm{I}_S - \mathcal{P}_{\mathbf{1}_S}) \widehat{\boldsymbol{E}}^\top \widehat{f}(\boldsymbol{z}) + \mathbf{1}_S/2 - \boldsymbol{E}_\star^\top \boldsymbol{E}_\star(\boldsymbol{z})].
\end{aligned}
$$

Since $\sqrt{2} \boldsymbol{P}_{\mathcal{Y}}^{1/2} \boldsymbol{E}_\star(\boldsymbol{z}) = (\mathbb{P}_c(y | \boldsymbol{x}^{c_1} = \boldsymbol{z}))_{y \in [M]}$ and $\boldsymbol{z} \overset{d}{=} \boldsymbol{x}^{c_1}$ follows the uniform distribution on $[S]$ by assumption, it follows that

$$
\begin{aligned}
&\mathbb{E}_{\boldsymbol{x}, g} \Big[ \sum_{y \in [M]} (\mathbb{P}_c(y | \boldsymbol{x}^{c_1} = \boldsymbol{z}) - (\boldsymbol{\Gamma}_w \widehat{f}(\boldsymbol{z}) + \boldsymbol{\Gamma}_b)_y)^2 \Big] \\
&\leqslant 2 \mathbb{E}_{\boldsymbol{z}} [\|(\boldsymbol{E}_\star \boldsymbol{E}_\star^\top)^{-1} \boldsymbol{E}_\star [(\mathrm{I}_S - \mathcal{P}_{\mathbf{1}_S}) \widehat{\boldsymbol{E}}^\top \widehat{f}(\boldsymbol{z}) + \mathbf{1}_S/2 - \boldsymbol{E}_\star^\top \boldsymbol{E}_\star(\boldsymbol{z})]\|_2^2] \\
&\leqslant \frac{2}{\sigma_{\boldsymbol{E}_\star}^2} \mathbb{E}_{\boldsymbol{z}} [\|[(\mathrm{I}_S - \mathcal{P}_{\mathbf{1}_S}) \widehat{\boldsymbol{E}}^\top \widehat{f}(\boldsymbol{z}) + \mathbf{1}_S/2 - \boldsymbol{E}_\star^\top \boldsymbol{E}_\star(\boldsymbol{z})]\|_2^2] \\
&\leqslant \frac{2}{S \sigma_{\boldsymbol{E}_\star}^2} \|(\mathrm{I}_S - \mathcal{P}_{\mathbf{1}_S}) \widehat{\boldsymbol{E}}^\top \widehat{\boldsymbol{E}} + \mathbf{1}_S \mathbf{1}_S^\top/2 - \boldsymbol{E}_\star^\top \boldsymbol{E}_\star\|_{\mathrm{fro}}^2 \\
&= \frac{2}{S \sigma_{\boldsymbol{E}_\star}^2} \|(\mathrm{I}_S - \mathcal{P}_{\mathbf{1}_S}) \widehat{\boldsymbol{E}}^\top \widehat{\boldsymbol{E}} - (\mathrm{I}_S - \mathcal{P}_{\mathbf{1}_S}) \boldsymbol{E}_\star^\top \boldsymbol{E}_\star\|_{\mathrm{fro}}^2,
\end{aligned}
\tag{47}
$$

where the last equality follows since $\boldsymbol{E}_\star^\top(\boldsymbol{z}^{(1)}) \boldsymbol{E}_\star(\boldsymbol{z}^{(2)}) = \frac{\mathbb{P}_c(\boldsymbol{z}^{(1)}, \boldsymbol{z}^{(2)})}{2 \mathbb{P}_c(\boldsymbol{z}^{(1)}) \mathbb{P}_c(\boldsymbol{z}^{(2)})}$ for any $(\boldsymbol{z}^{(1)}, \boldsymbol{z}^{(2)}) \in [S]$, and $\frac{1}{S} \sum_{\boldsymbol{z}^{(2)} \in [S]} \frac{\mathbb{P}_c(\boldsymbol{z}^{(1)}, \boldsymbol{z}^{(2)})}{\mathbb{P}_c(\boldsymbol{z}^{(1)}) \mathbb{P}_c(\boldsymbol{z}^{(2)})} = 1$ for all $\boldsymbol{z}^{(1)} \in [S]$.

Step 2: bound the expression by excess risk. We claim that for some absolute constant $c > 0$,

$$
\begin{aligned}
&\|(\mathrm{I}_S - \mathcal{P}_{\mathbf{1}_S}) \widehat{\boldsymbol{E}}^\top \widehat{\boldsymbol{E}} - (\mathrm{I}_S - \mathcal{P}_{\mathbf{1}_S}) \boldsymbol{E}_\star^\top \boldsymbol{E}_\star\|_{\mathrm{fro}}^2 \\
&\leqslant c \|(\mathrm{I}_S - \mathcal{P}_{\mathbf{1}_S})(\widehat{\boldsymbol{E}}^\top \widehat{\boldsymbol{E}} + \widehat{w} \mathrm{I}_S) - (\mathrm{I}_S - \mathcal{P}_{\mathbf{1}_S})(\boldsymbol{E}_\star^\top \boldsymbol{E}_\star + S \cdot \mathrm{I}_S/2)\|_{\mathrm{fro}}^2, \quad \text{and} \tag{48a} \\
&\|(\mathrm{I}_S - \mathcal{P}_{\mathbf{1}_S})(\widehat{\boldsymbol{E}}^\top \widehat{\boldsymbol{E}} + \widehat{w} \mathrm{I}_S) - (\mathrm{I}_S - \mathcal{P}_{\mathbf{1}_S})(\boldsymbol{E}_\star^\top \boldsymbol{E}_\star + S \cdot \mathrm{I}_S/2)\|_{\mathrm{fro}}^2 \\
&\leqslant S^2 \cdot (R_{\mathrm{f}}(\mathsf{S}_{\widehat{f}_{\mathrm{aug}}}) - R_{\mathrm{f}}(\mathsf{S}_\star)). \tag{48b}
\end{aligned}
$$

Combining claim (48a) and (48b) and bound (47) yields the desired bound. Now, it remains to prove these two claims.

**Proof of claim (48a).** Adopt the shorthand notation $\Delta = (\widehat{\boldsymbol{E}}^\top \widehat{\boldsymbol{E}} + \widehat{w} \mathrm{I}_S) - (\boldsymbol{E}_\star^\top \boldsymbol{E}_\star + S \cdot \mathrm{I}_S/2)$. First, by the triangle inequality, it suffices to show

$$
\|(\mathrm{I}_S - \mathcal{P}_{\mathbf{1}_S})(\widehat{w} - S/2)\|_{\mathrm{fro}}^2 \leqslant c \|(\mathrm{I}_S - \mathcal{P}_{\mathbf{1}_S}) \Delta\|_{\mathrm{fro}}^2
$$

for some absolute constant $c > 0$. Note that $\mathsf{rank}(\widehat{\boldsymbol{E}}^\top \widehat{\boldsymbol{E}} - \boldsymbol{E}_\star^\top \boldsymbol{E}_\star) \leqslant 2M$, therefore, there are at least $S/2$ singular values of $\Delta$ which equal $|\widehat{w} - S|/2$. As a result, we have

$$
\begin{aligned}
\|(\mathrm{I}_S - \mathcal{P}_{\mathbf{1}_S}) \Delta\|_{\mathrm{fro}}^2 &= \mathrm{trace}(\Delta(\mathrm{I}_S - \mathcal{P}_{\mathbf{1}_S}) \Delta) = \|\Delta\|_{\mathrm{fro}}^2 - \frac{1}{S} \mathbf{1}_S^\top \Delta^2 \mathbf{1}_S \\
&\geqslant \|\Delta\|_{\mathrm{fro}}^2 - \|\Delta\|_{\mathrm{op}}^2 \geqslant \frac{1}{4} \|(\widehat{w} - S/2) \mathrm{I}_S\|_{\mathrm{fro}}^2 \geqslant \frac{1}{4} \|(\mathrm{I}_S - \mathcal{P}_{\mathbf{1}_S})(\widehat{w} - S/2)\|_{\mathrm{fro}}^2.
\end{aligned}
$$

**Proof of claim (48b).** Adpot the shorthands $\mathsf{S}_{\widehat{f}_{\mathrm{aug}}}^{\mathsf{m}} := \left(\mathsf{S}_{\widehat{f}_{\mathrm{aug}}}(\boldsymbol{z}^{(1)}, \boldsymbol{z}^{(2)})\right)_{\boldsymbol{z}^{(1)}, \boldsymbol{z}^{(2)} \in [S]} \in \mathbb{R}^{S \times S}$ and $\mathsf{S}_\star^{\mathsf{m}} := \left(\mathsf{S}_\star(\boldsymbol{z}^{(1)}, \boldsymbol{z}^{(2)})\right)_{\boldsymbol{z}^{(1)}, \boldsymbol{z}^{(2)} \in [S]} \in \mathbb{R}^{S \times S}$, where $\mathsf{S}_\star(\boldsymbol{z}^{(1)}, \boldsymbol{z}^{(2)}) = \frac{\mathbb{P}(\boldsymbol{z}^{(1)}, \boldsymbol{z}^{(2)})}{\mathbb{P}_{\boldsymbol{z}}(\boldsymbol{z}^{(1)}) \mathbb{P}_{\boldsymbol{z}}(\boldsymbol{z}^{(2)})}$. Since we

assume $\boldsymbol{z} \stackrel{d}{=} \boldsymbol{x}^{c_1}$ follows the uniform distribution on $[S]$, by the definition of $\widehat{f}_{\mathsf{aug}}$ and claim (37) in the proof of Eq. (14)

$$\|(\mathrm{I}_S - \mathcal{P}_{\mathbf{1}_S})(\widehat{\boldsymbol{E}}^\top \widehat{\boldsymbol{E}} + \widehat{w}\mathrm{I}_S) - (\mathrm{I}_S - \mathcal{P}_{\mathbf{1}_S})(\boldsymbol{E}_\star{}^\top \boldsymbol{E}_\star + S \cdot \mathrm{I}_S/2)\|_{\mathrm{fro}}^2$$
$$= \|(\mathrm{I}_S - \mathcal{P}_{\mathbf{1}_S})(\mathsf{S}_{\widehat{f}_{\mathsf{aug}}}^{\mathsf{m}} - \mathsf{S}_\star{}^{\mathsf{m}})\|_{\mathrm{fro}}^2$$
$$= S^2 \cdot T_1,$$

where

$$T_1 := \mathbb{E}_{\boldsymbol{z}^{(1)}, \boldsymbol{z}^{(2)} \sim \mathbb{P}_{\boldsymbol{z}} \times \mathbb{P}_{\boldsymbol{z}}}[((\mathsf{S}_{\widehat{f}_{\mathsf{aug}}} - \mathsf{S}_\star)(\boldsymbol{z}^{(1)}, \boldsymbol{z}^{(2)}) - \mathbb{E}_{\boldsymbol{z}^{(2)} \sim \mathbb{P}_{\boldsymbol{z}}}[(\mathsf{S}_{\widehat{f}_{\mathsf{aug}}} - \mathsf{S}_\star)(\boldsymbol{z}^{(1)}, \boldsymbol{z}^{(2)})])^2].$$

Finally, by a second-order Taylor expansion of $R_{\mathrm{f}}(\mathsf{S})$ at $\mathsf{S}_\star$, we have

$$R_{\mathrm{f}}(\mathsf{S}_{\widehat{f}_{\mathsf{aug}}}) - R_{\mathrm{f}}(\mathsf{S}_\star) = T_1.$$

Combining the two equalities yields the claim.

**Proof of claim** (46). Note that for any $\boldsymbol{z} \in [S]$,

$$|1 - \|\mathsf{trun}(\boldsymbol{\Gamma}_w \widehat{f}(\boldsymbol{z}) + \boldsymbol{\Gamma}_b)\|_1| \leqslant \sum_{y \in [M]} |\mathbb{P}_c(y|\boldsymbol{x}^{c_1} = \boldsymbol{z}) - \mathsf{trun}(\boldsymbol{\Gamma}_w \widehat{f}(\boldsymbol{z}) + \boldsymbol{\Gamma}_b)_y|$$
$$\leqslant \sum_{y \in [M]} |\mathbb{P}_c(y|\boldsymbol{x}^{c_1} = \boldsymbol{z}) - (\boldsymbol{\Gamma}_w \widehat{f}(\boldsymbol{z}) + \boldsymbol{\Gamma}_b)_y|$$
$$\leqslant \sqrt{MS} \cdot \sqrt{\mathbb{E}_{\boldsymbol{x},g}\Big[ \sum_{y \in [M]} (\mathbb{P}_c(y|\boldsymbol{x}^{c_1} = \boldsymbol{z}) - (\boldsymbol{\Gamma}_w \widehat{f}(\boldsymbol{z}) + \boldsymbol{\Gamma}_b)_y)^2 \Big]},$$

where the last line follows from the assumption that $\boldsymbol{x}^{c_1}$ (and hence $\boldsymbol{z}$) follows the uniform distribution on $[S]$. Thus, combining Eq. (47), claim (48a) and (48b) yields

$$|1 - \|\mathsf{trun}(\boldsymbol{\Gamma}_w \widehat{f}(\boldsymbol{z}) + \boldsymbol{\Gamma}_b)\|_1| \leqslant c \frac{S\sqrt{M}}{\sigma_{\boldsymbol{E}_\star}} \cdot \sqrt{R_{\mathrm{f}}(\mathsf{S}_{\widehat{f}_{\mathsf{aug}}}) - R_{\mathrm{f}}(\mathsf{S}_\star)} \leqslant \frac{1}{2}.$$

$\square$

# E   Additional experiments

We also conducted small-scale experiments in the CLIP setting (language-image pretraining, [31]) to compare the contrastive learning losses. Namely, we use the CLIP model (RN50-quickgelu, which consists of a ResNet-50 image encoder and 12-layer Transformer text encoder) on a 100K subsample of the cc3m-wds dataset [33] using both KL (i.e., InfoNCE) and $\chi^2$-contrastive losses (Eq. 3 and 10). The original dataset contains about 3.3M image-text pairs, but due to limited compute, we trained on the subsample for 32 epochs.

We evaluated the models based on their zero-shot classification performance on the ImageNet-1k validation set (1000 classes, 500 images per class). For KL and $\chi^2$-contrastive losses, we set the link functions $\tau(x)$ to $x/t$ and $e^{x/t}$, respectively, with trainable temperature $t$ initialized to 1. We used a batch size of 128 and the AdamW optimizer with weight decay 0.02, and selected the best learning rate via grid search from $\{3e{-}5, 1e{-}4, 3e{-}4, 1e{-}3\}$. The optimal learning rate for both losses is $3 \times 10^{-4}$.

Table 1: Top-5 zero-shot classification accuracy on ImageNet-1k.

| Method | Accuracy (%) |
|---|---|
| InfoNCE | $7.5 \pm 0.3$ |
| Chi-squared | $9.4 \pm 0.1$ |

We repeated the experiments three times and report the top-5 accuracy on the ImageNet-1k validation set. From Table 1, we observe that in this small-scale experiment, the model trained with $\chi^2$-contrastive loss achieves zero-shot performance comparable to that of InfoNCE. We do not claim that the $\chi^2$-contrastive loss is superior, as both methods could benefit from further hyperparameter tuning (e.g., initial temperature) or larger datasets. However, we note that $\chi^2$-contrastive loss is able to learn representations that are useful for downstream tasks, which is consistent with our theoretical findings. We leave more extensive experiments in the CLIP setting to future work.

