# OpenReview forum: "A Statistical Theory of Contrastive Learning via Approximate Sufficient Statistics"
_NeurIPS.cc/2025/Conference — NeurIPS 2025 poster_

### Official Review · Reviewer_DTT6 · 2025-06-18

**Clarity:** 2
**Significance:** 3
**Originality:** 2
**Rating:** 4
**Confidence:** 3

**Summary:**

The paper develops a statistical framework for augmentation based contrastive learning such as SimCLR through the concept of approximate sufficient statistics. Three equivalent definitions of approximate sufficient statistics for general f-divergences are proposed. The authors show that minimizing a finite-batch InfoNCE loss yields encoders whose sufficiency gap is controlled by the excess empirical risk.
They then derive excess risk bounds for downstream regression and classification that depend on the encoder's sufficiency and the augmentation error.

**Questions:**

I would consider increasing the score after the following questions are at least partly addressed.
1. Could you explain the technical gap or challenges to Oko et al. 2025 when dealing with a self-supervised contrastive learning for a single modality data?
1. The paper introduces the augmentation error term $\epsilon_G$. Do you have any upper bounds for any real augmentation such as random cropping, color jitter, or blur?
1. As a curiosity, how does the temperature parameter in the InfoNCE loss affect convergence rate?

**Ethical Concerns:**

["NO or VERY MINOR ethics concerns only"]

**Final Justification:**

I am changing my score to Borderline Accept (4), as the rebuttal meets the criteria I laid out in my Questions. In particular, it resolves my novelty concern of the difference from Oko et al. (2025).

On $\epsilon_{\mathcal{G}}$ for real augmentations, I understand that general bounds are difficult, but there is still no concrete bound even for a simplified version of a canonical augmentation (e.g., random cropping). This remaining gap is why my recommendation is borderline rather than a clear accept.

**Limitations:**

yes

**Paper Formatting Concerns:**

no major formatting issues found

**Quality:**

3

**Strengths And Weaknesses:**

**Strengths**

1. Solid theoretical analysis for self supervised contrastive learning using data augmentation.
1. Establishes a clear link between downstream risk and approximate sufficient statistics, and provides learning-theoretic bounds for those statistics.

**Weaknesses**

1. Their second claimed contribution (line 44) sounds vague. They assert that random transformations in SimCLR "introduce additional challenges for theoretical analysis," but they do not identify the challenges of applying Oko et al. 2025 to the contrastive learning for a single modality data.
At the same time, their first claimed contribution (line 39) of extending approximate sufficient statistics from the KL divergence to any f divergence and proving three equivalent forms remains a modest generalization of Oko et al. 2025.

1. SimCLR defines specific data augmentation schemes that are central to its success, yet the paper did not derive the resulting augmentation bias $\epsilon_{\mathcal{G}}$ for any of these transforms.

---

> ### Author Rebuttal · Authors · 2025-07-28
>
> We are grateful to Reviewer DTT6 for their thoughtful review of our submission. Below, we respond to each question  raised in their review.
>
> ---
> > Q1. Could you explain the technical gap or challenges to Oko et al. 2025 when dealing with a self-supervised contrastive learning for a single modality data?
>
> A1. A main technical challenge of extending Oko et al. (2025) [1] to single-modality data (especially to SimCLR) is handling the error induced by random augmentation. Note that random augmentation is not performed in CLIP considered in [1], and therefore the downstream error can be controlled directly via the sufficiency of the encoder (see, e.g., Propositions 2, 3, and 4 in [1]). In our work, we introduce a novel augmentation error term $\epsilon\_{\mathcal{G}}$ (or $\epsilon^{\mathsf{cls}}\_{\mathcal{G}}$), and use it to establish downstream error bounds for any encoder $f$, by applying triangle inequalities for $\ell_2$ error in regression (Theorem 2) and for KL divergence in classification (Theorem 3). Moreover, in the regression case (Theorem 2 and the subsequent discussion), we show that the same bound holds for the minimum error $\widetilde\epsilon\_{\mathcal{G}}$, suggesting the tightness of our bounds for encoders $f$ with low sufficiency.
>
> ---
>
> > Q2. The paper introduces the augmentation error term $\epsilon_{\mathcal{G}}$. Do you have any upper bounds for any real augmentation such as random cropping, color jitter, or blur?
>
> A2. We thank the reviewer for this practical question. In general, it is challenging to derive upper bounds for real-world augmentations such as cropping, color jitter, or blur, as such bounds depend on concrete assumptions about the augmentations and oracle knowledge of the data. For example, in image classification tasks, the augmentation error of random cropping depends on the proportion of pixels dropped and the true label probabilities of the cropped images—which are typically unknown. Nevertheless, in practice, one may use the predicted label probabilities of the original and cropped images from a highly accurate image classifier as a surrogate for the true label probabilities, and use them to empirically estimate the augmentation error $\epsilon^{\mathsf{cls}}_{\mathcal{G}}$.
>
>  Theoretically, we emphasize that our main results hold for arbitrary data augmentation $g$, with the downstream error bounds depending on $g$ only through the augmentation error $\epsilon\_{\mathcal{G}}$ (or $\epsilon^{\mathsf{cls}}\_{\mathcal{G}}$). Explicit bounds on  $\epsilon\_{\mathcal{G}}$ (or $\epsilon^{\mathsf{cls}}\_{\mathcal{G}}$) require additional assumptions on the augmentation. For example, in the context of Theorem 6, we have $\epsilon\_{\mathcal{G}}=0$ if $g$ is the projection onto some random subspace that contains $\theta_\star$.
>
> ---
> > Q3. As a curiosity, how does the temperature parameter in the InfoNCE loss affect convergence rate?
>
> A3. In our work, the temperature parameter (denoted by $t$) enters the InfoNCE loss through the link function $\tau(x)$ in Eq. (2) when defining  $\tau(x):=x/t$. Theoretically, the temperature affects our bound on sufficiency (Theorem 1) through affecting the upper bound on the score $B_{\mathsf{S}}$. Qualitatively, a smaller $t$ results in a larger constant $C$ in the bound in Eq. (4), which implies a slower convergence speed (up to constant factors).
>
>
>
>
>
>
>
>
> ---
>
>
> [1]. Oko, K., Lin, L., Cai, Y., & Mei, S. (2025). A statistical theory of contrastive pre-training and multimodal generative ai. arXiv preprint arXiv:2501.04641.

---

> ### Comment · Reviewer_DTT6 · 2025-08-06
>
> Thank you for the detailed response.
>
> Your clarification on the SimCLR-specific challenge and the resulting augmentation error decomposition addressed my main concern about novelty. Accordingly, I am raising my score.
>
> While I appreciate that deriving explicit bounds for the augmentation error $\epsilon_{\mathcal{G}}$ is difficult in general, I strongly recommend including at least one concrete form or upper bound in the camera-ready, even under simplified assumptions.

---

> > ### Author Response · Authors · 2025-08-06
> >
> > Thanks for your helpful comments and for raising the score! We will make sure to include concrete examples on the augmentation error $\epsilon_{\mathcal{G}}$ in the paper.

---

### Official Review · Reviewer_RiyN · 2025-07-01

**Clarity:** 2
**Significance:** 3
**Originality:** 4
**Rating:** 4
**Confidence:** 3

**Summary:**

This paper develops a new theoretical framework for data augmentation-based contrastive learning, focusing on SimCLR. It generalizes the concept of approximate sufficient statistics from prior work on CLIP to general f-divergences and establishes that minimizing contrastive losses yields encoders that are approximately sufficient. The authors show that minimizing contrastive losses (like InfoNCE) yields encoders that are approximately sufficient, and that the sufficiency of these encoders, together with the error induced by data augmentation, governs their adaptability and performance on downstream tasks. The paper provides concrete examples in linear regression and topic classification to illustrate the applicability of the theory.

**Questions:**

- Is it straightforward to extend your theoretical results to multimodal contrastive learning settings (e.g. image-text), in line with identifiability analyse such as in Daunhawer et al. ICLR 2024 https://arxiv.org/pdf/2303.09166, IDENTIFIABILITY RESULTS FOR MULTIMODAL CONTRASTIVE LEARNING
- How does your approximate sufficiency framework encompass or relate to alternative contrastive/self-supervised objectives such as VICReg, Barlow Twins, SwAV, and BYOL, DINO, where some of them do not explicitly use negative samples, and yet surpasses SimCLR in most natural imaging experiments ?
- Could you clarify the practical validity of your assumptions (e.g. bounded score functions, invertibility and Lipschitz continuity of the link function), particularly in the context of deep neural network encoders?
- Could you clarify the impact of the encoder architecture on your generalization bounds ? Does it impact the approximation error ?
- Do you have experimental results comparing the empirical performance of encoders trained with different f-divergences (e.g. KL vs. chi-squared contrastive losses) on standard datasets?
- Could you discuss how your present work connects/relates to identifiability ?

**Ethical Concerns:**

["NO or VERY MINOR ethics concerns only"]

**Final Justification:**

Authors have globally responded to my points but I wish to keep my score as it is.

**Limitations:**

Yes.

**Paper Formatting Concerns:**

Nothing to report.

**Quality:**

3

**Strengths And Weaknesses:**

Strengths:
- The paper significantly extends the notion of approximate sufficiency from its prior, more limited form (KL-divergence for CLIP) to a general framework encompassing various f-divergences and equivalent mathematical formulations. This generalization is both non-trivial and valuable for the broader contrastive learning community.
- The definitions of sufficiency (information loss, variational form, conditional Bregman) are carefully presented and shown to be equivalent, providing a solid mathematical foundation.
- In Sec. 3.1, the paper establishes a sufficiency bound for the ERM estimator in and shows that when f minimizes the ERM, the bound is a function of a generalization error (which goes to 0 as the number of samples tend to infinity) and a constant approximation error.
- Then, in Sec. 3.2, the paper establishes that the downstream performance of learned encoders on linear regression and topic classification depends on both their sufficiency and the augmentation-induced error, providing a theoretical explanation on previous work focusing on how the data augmentions affect the performances of SimCLR's encoder.
- In Sec 3.3, the authors generalizes to other loss terms than SimCLR, by generalizing their result of other type of f using general f-sufficiency. In Eq. 9, they notably extend to the chi-squared sufficiency. This generalization is appreciated and is a strength of the paper.

Weaknesses:
- The paper is primarily theoretical; while it provides illustrative examples, it lacks empirical experiments that would demonstrate the practical impact of the theoretical bounds and insights in real-world settings. Notbaly, it would have been interesting to study the use on real dataset of various f, such as teh chi-squared contrastive loss for ex. Besides, we would have expected an experimental comparison between empirical risk and theoretical risk, using either real-world dataset (if applicable), or a synthetic one.
- Overall, the paper is hard to read, and the use of the term "encoder" to both talk about the neural network and the f of the f-MI makes it confusing.
- While the paper discuss how the sufficiency of the encoder impact the downstream performances, it eludes past work about identifiability (Self-Supervised Learning with Data Augmentations Provably Isolates Content from Style, von Kugelgen et al. NeurIPS 2021, https://openreview.net/pdf?id=4pf_pOo0Dt), even though the connection between identifiability and encoder-sufficiency appears intuitively close.
- While the theoretical bounds are elegant, they rely on assumptions such as boundedness of the score function and invertibility/Lipschitz continuity of the link function. In practical neural networks, these conditions may not strictly hold. Discussing these assumptions would be beneficial for the paper. Moreover, the theory abstracts away from architectural details (e.g., depth, parameter count), which are known to affect the performance of learned representations. A discussion on how architecture choices influence sufficiency and generalization would strengthen the practical relevance of the results.

---

> ### Author Rebuttal · Authors · 2025-07-28
>
> We sincerely appreciate Reviewer RiyN for their thorough review and constructive feedback on our work. Below, we hope to address all your comments.
>
> > Q1 and Q6. generalization to multimodal contrastive learning and connection with identifiability [1,2]
>
> A1. We thank the reviewer for this insightful question and pointing out the connection with the identifiability results.
>
> **Generalization to multimodal contrastive learning.**  Yes, it is straightforward to generalize our theoretical results to multimodal contrastive learning. The main modification is to choose $X,Y$ as a pair of samples $(x,y)$ from two modalities (e.g, an image and its associated text) instead of two augmented views in the definition of sufficiency (see line 99). As a consequence, similar downstream bounds as in [4] can be derived based on the generalized $\mathsf{f}$-sufficiency.
>
> **Connection with identifiability.** The identifiability results in contrastive learning [1,2] consider a setting where paired views (either from augmentations or across modalities)  share some common latent factors, and show that a representation that minimizes the expected alignment loss can identify these latent factors. Under their assumptions, it can be shown that the learned representation is equivalent to an encoder with zero sufficiency in our work. In contrast, we derive general downstream error bounds for encoders with arbitrary sufficiency, even though the encoders cannot exactly identify the hidden factors when the sufficiency measure is strictly positive.
>
>
> ---
>
> > Q2. How does your approximate sufficiency framework encompass or relate to alternative contrastive/self-supervised objectives such as VICReg, Barlow Twins, SwAV, and BYOL, DINO?
>
> A2.  In our work, the approximate sufficiency framework applies to any encoder $f$, including those obtained from alternative contrastive/self-supervised objectives mentioned by the reviewer, with the downstream error bounds depending on the sufficiency of $f$. However, while the excess risk provides a direct upper bound on the sufficiency for the InfoNCE loss (line 166), for the alternative contrastive/self-supervised objectives mentioned above, it is not yet clear how their respective excess losses relate to our sufficiency measure. We leave establishing the connections between these excess losses and the sufficiency measure for future exploration.
>
> ---
>
> > Q3. Could you clarify the practical validity of your assumptions (e.g. bounded score functions, invertibility and Lipschitz continuity of the link function), particularly in the context of deep neural network encoders?
>
> A3. In practice, the link function $\tau(x)$ is often chosen as a linear function, i.e., $\tau(x) = x/t$, where $t > 0$ is the temperature parameter. Thus, the invertibility and Lipschitz continuity of the link function are naturally satisfied. In SimCLR [3], the encoder $f$ is normalized before computing the score; therefore, the score is bounded by $1/t$, where the temperature $t \in \\{0.1, 0.5, 1\\}$ in their experiments.
>
> ---
>
> > Q4. Could you clarify the impact of the encoder architecture on your generalization bounds ? Does it impact the approximation error ?
>
> A4. The encoder architecture affects the generalization bounds by influencing the covering number in the second term of the generalization error (see line 236). By a standard parameter-counting argument (e.g., Theorem 6 in [4]), the covering number can be upper bounded by $\mathcal{O}(d)$ for transformer-based models. The approximation error, on the other hand, depends on the encoder class $\mathcal{F}$. A richer and more complex encoder class (or architecture) has a smaller approximation error but potentially larger generalization error.
>
> ---
>
> > Q5. Do you have experimental results comparing the empirical performance of encoders trained with different f-divergences (e.g. KL vs. chi-squared contrastive losses) on standard datasets?
>
> A5. We conduct synthetic experiments to learn data representations via contrastive learning with a two-layer neural network, and evaluate them on downstream linear regression.
>
>
> In the contrastive learning phase, we generate $n$ i.i.d. samples $x_i \sim \mathcal{N}(0, I_d)$. The augmentation $g$ adds i.i.d. $\mathcal{N}(0, \sigma_1^2)$ noise to the first $s < d$ coordinates of $x_i$, and replaces the remaining coordinates with i.i.d. $\mathcal{N}(0, 1)$ noise. We apply KL and $\chi^2$-contrastive learning (Eq 3 and 10) with link function $\tau(x) = x$, and encoder $f(x)$ being a two-layer ReLU neural network mapping $\mathbb{R}^d$ to $\mathbb{R}^s$. We set $s = 10,d = 100,n=500$, hidden dimension 64, and batch size $K = 64$. The encoder is trained using Adam (learning rate 0.001) for 300 epochs, after which the training loss is observed to converge.
>
>
> For downstream regression, we generate $m$ i.i.d. samples $(x_i, y_i)\_{i=1}^m$, where $x_i \sim \mathcal{N}(0, I_d)$ and $y_i = \langle x_i, \theta_\star \rangle + \epsilon_i$, with $\epsilon_i \sim \mathcal{N}(0, \sigma^2)$ independent of $x_i$. We choose $\theta_\star = (\mathbf{1}^\top_s/\sqrt{s},\mathbf{0}^\top_{d-s})^\top$ and $\sigma=1$. Using the learned representation $\widehat{f}(x_i) \in \mathbb{R}^s$ from KL (or $\chi^2$)-contrastive learning, we fit a downstream linear model to predict $y_i$. We define the excess risk of any predictor $h$ as $\mathbb{E}[(y_i - h(x_i))^2] - \sigma^2$, and evaluate the excess risk of the linear model trained on $(\widehat{f}(x_i),y_i)\_{i=1}^m$ using 50000 test samples. For comparison, we also report the excess risk of a linear model trained directly on the original samples $(x_i,y_i)\_{i=1}^m$. Results for various downstream sample sizes $m$ and the standard deviation over 10 runs are shown below.
>
> > **Table: Excess risk  for various downstream sample sizes m**
> >
> > | m    | InfoNCE         | Chi-squared     | Direct LR       |
> > |------|-----------------|-----------------|-----------------|
> > | 150 | 0.106 ± 0.040 | 0.120 ± 0.028 | 2.066 ± 0.594 |
> > | 500 | 0.060 ± 0.015 | 0.070 ± 0.013 | 0.243 ± 0.032 |
> > | 1000 | 0.052 ± 0.012 | 0.063 ± 0.012 | 0.114 ± 0.021 |
> > | 2000 | 0.046 ± 0.011 | 0.058 ± 0.013 | 0.055 ± 0.012 |
> > | 5000 | 0.042 ± 0.011 | 0.055 ± 0.013 | 0.021 ± 0.005 |
> > | 10000 | 0.040 ± 0.012 | 0.054 ± 0.012 | 0.011 ± 0.004 |
>
>
> From the table, we observe that InfoNCE and Chi-squared achieve comparable excess risks (differences about one standard deviation), and both are substantially lower than that of direct linear regression when the sample size $m$ is relatively small (e.g., $m = 150, 500$). This suggests that both KL and $\chi^2$-contrastive learning can learn a “good” low-dimensional representation for the downstream task. As the sample size increases, the excess risk of direct linear regression converges to zero, while those of InfoNCE and Chi-squared converge to non-zero constants. This is consistent with our theoretical results, which attribute the excess risk to the non-zero sufficiency of $\widehat{f}$ and the augmentation error $\epsilon_{\mathcal{G}}$.
>
> Due to limited computational resources, we did not evaluate the performance of $\chi^2$-contrastive learning on large-scale real-world datasets and leave this for future investigation.
>
> ---
>
> > Other comments: use $f$ for both encoder and $f$-divergence.
>
> We currently use different fonts for $f$ when referring to the encoder $f$ and the $\mathsf{f}$-divergence. We will clarify this distinction more explicitly in the paper to avoid confusion.
>
>
>
>
>
>
>
>
>
>
>
>
>
>
>
>
>
>
>
>
>
>
>
>
> ---
>
> [1]. Von Kügelgen, Julius, et al. "Self-supervised learning with data augmentations provably isolates content from style." Advances in neural information processing systems 34 (2021): 16451-16467.
>
> [2]. Daunhawer, Imant, et al. "Identifiability results for multimodal contrastive learning." arXiv preprint arXiv:2303.09166 (2023).
>
> [3]. Chen, Ting, et al. "A simple framework for contrastive learning of visual representations." International conference on machine learning. PmLR, 2020.
>
> [4]. Oko, K., Lin, L., Cai, Y., & Mei, S. (2025). A statistical theory of contrastive pre-training and multimodal generative ai. arXiv preprint arXiv:2501.04641.

---

> > ### Comment · Reviewer_RiyN · 2025-08-04
> >
> > I thank the authors for the detailed responses to my questions. My score remains unchanged.

---

### Official Review · Reviewer_sYsE · 2025-07-02

**Clarity:** 2
**Significance:** 3
**Originality:** 3
**Rating:** 4
**Confidence:** 2

**Summary:**

This paper proposes a theoretical framework for analyzing data augmentation-based contrastive learning, particularly SimCLR, by means of approximate sufficient statistics, which extends a prior work using KL-divergence to general f-divergences. Specifically, this work shows that minimizing a contrastive loss produces encoders approximately sufficient, and they can be adapted to downstream regression and classification tasks, where downstream performance depends on their sufficiency and the error induced by data augmentation in contrastive learning.

**Questions:**

Please address concerns in Weaknesses above.

**Ethical Concerns:**

["NO or VERY MINOR ethics concerns only"]

**Final Justification:**

The authors successfully addressed my concerns. They also reported an additional experiment in a toy setting in their rebuttal, and further added another one in the CLIP setting around the end of the rebuttal period, though the experimental result in the CLIP setting seems somewhat weak.

**Limitations:**

Different from the checklist, limitations are not provided.

**Paper Formatting Concerns:**

nothing special

**Quality:**

3

**Strengths And Weaknesses:**

**Strengths**

+ The idea of analyzing contrastive learning via approximate sufficient statistics sounds interesting.

+ The claim is supported by comprehensive theoretical derivation.

+ Concrete examples on regression and classification are provided, though they are limited to linear models.


**Weaknesses**

- There is no experimental results, raising a concern on the practicality of the proposed idea. At least the prior work by Oko et al. [27] conducted some experiments with CLIP. If the theoretical results deviate from practical nonlinear models, then their contribution is limited accordingly. To address this concern, the authors could first conduct toy experiments with linear models as in the examples in Section 4, and then extend it to nonlinear models and see if their theoretical results empirically hold, or how much they are deviated from practical models.

- L252: It is not clear how "end-to-end theoretical guarantees for the downstream performance of encoders obtained by minimizing general f-contrastive losses" is drawn by combining the results from Sections 3.3.1 and 3.3.2.

- There are many "some constants" throughout the paper. How are they determined, at least in examples? How much are bounds tight?

- The error on the downstream task induced by data augmentation epsilon_G is an important variable in their analysis but it is not specified well. For example, it would be useful to know how we can measure epsilon_G in practice, how much its value deviates by taking different augmentation strategies, and so on.

- Formatting issue: generally speaking, any citation should not be appeared in Abstract.

- Typo: f-sufficieny in the title of Section 3.3.1

---

> ### Author Rebuttal · Authors · 2025-07-28
>
> We thank Reviewer sYsE for their careful review and valuable feedback. We hope to address all questions below.
>
> ---
> > Q1. There is no experimental results, raising a concern on the practicality of the proposed idea. At least the prior work by Oko et al. [27] conducted some experiments with CLIP. If the theoretical results deviate from practical nonlinear models, then their contribution is limited accordingly. To address this concern, the authors could first conduct toy experiments with linear models as in the examples in Section 4, and then extend it to nonlinear models and see if their theoretical results empirically hold, or how much they are deviated from practical models.
>
>  A1. We conduct synthetic experiments to learn data representations via contrastive learning with a two-layer neural network, and evaluate them on downstream linear regression.
>
>
> In the contrastive learning phase, we generate $n$ i.i.d. samples $x_i \sim \mathcal{N}(0, I_d)$. The augmentation $g$ adds i.i.d. $\mathcal{N}(0, \sigma_1^2)$ noise to the first $s < d$ coordinates of $x_i$, and replaces the remaining coordinates with i.i.d. $\mathcal{N}(0, 1)$ noise. We apply KL and $\chi^2$-contrastive learning (Eq 3 and 10) with link function $\tau(x) = x$, and encoder $f(x)$ being a two-layer ReLU neural network mapping $\mathbb{R}^d$ to $\mathbb{R}^s$. We set $s = 10,d = 100,n=500$, hidden dimension 64, and batch size $K = 64$. The encoder is trained using Adam (learning rate 0.001) for 300 epochs, after which the training loss is observed to converge.
>
>
> For downstream regression, we generate $m$ i.i.d. samples $(x_i, y_i)\_{i=1}^m$, where $x_i \sim \mathcal{N}(0, I_d)$ and $y_i = \langle x_i, \theta_\star \rangle + \epsilon_i$, with $\epsilon_i \sim \mathcal{N}(0, \sigma^2)$ independent of $x_i$. We choose $\theta_\star = (\mathbf{1}^\top_s/\sqrt{s},\mathbf{0}^\top_{d-s})^\top$ and $\sigma=1$. Using the learned representation $\widehat{f}(x_i) \in \mathbb{R}^s$ from KL (or $\chi^2$)-contrastive learning, we fit a downstream linear model to predict $y_i$. We define the excess risk of any predictor $h$ as $\mathbb{E}[(y_i - h(x_i))^2] - \sigma^2$, and evaluate the excess risk of the linear model trained on $(\widehat{f}(x_i),y_i)\_{i=1}^m$ using 50000 test samples. For comparison, we also report the excess risk of a linear model trained directly on the original samples $(x_i,y_i)\_{i=1}^m$. Results for various downstream sample sizes $m$ and the standard deviation over 10 runs are shown below.
>
> > **Table: Excess risk  for various downstream sample sizes m**
> >
> > | m    | InfoNCE         | Chi-squared     | Direct LR       |
> > |------|-----------------|-----------------|-----------------|
> > | 150 | 0.106 ± 0.040 | 0.120 ± 0.028 | 2.066 ± 0.594 |
> > | 500 | 0.060 ± 0.015 | 0.070 ± 0.013 | 0.243 ± 0.032 |
> > | 1000 | 0.052 ± 0.012 | 0.063 ± 0.012 | 0.114 ± 0.021 |
> > | 2000 | 0.046 ± 0.011 | 0.058 ± 0.013 | 0.055 ± 0.012 |
> > | 5000 | 0.042 ± 0.011 | 0.055 ± 0.013 | 0.021 ± 0.005 |
> > | 10000 | 0.040 ± 0.012 | 0.054 ± 0.012 | 0.011 ± 0.004 |
>
>
> From the table, we observe that InfoNCE and Chi-squared achieve comparable excess risks (differences about one standard deviation), and both are substantially lower than that of direct linear regression when the sample size $m$ is relatively small (e.g., $m = 150, 500$). This suggests that both KL and $\chi^2$-contrastive learning can learn a “good” low-dimensional representation for the downstream task. As the sample size increases, the excess risk of direct linear regression converges to zero, while those of InfoNCE and Chi-squared converge to non-zero constants. This is consistent with our theoretical results, which attribute the excess risk to the non-zero sufficiency of $\widehat{f}$ and the augmentation error $\epsilon_{\mathcal{G}}$.
>
> Due to limited computational resources, we did not evaluate the performance of $\chi^2$-contrastive learning on large-scale real-world datasets and leave this for future investigation.
>
> ---
>
> > Q2. L252: It is not clear how "end-to-end theoretical guarantees for the downstream performance of encoders obtained by minimizing general f-contrastive losses" is drawn by combining the results from Sections 3.3.1 and 3.3.2.
>
> A2. In Section 3.3.1, we show that the sufficiency of encoders can be bounded by the excess risk of general f-contrastive losses  (line 218), with concrete calculations for $\chi^2$-sufficiency in Theorem 4. In Section 3.3.2, Proposition 5 shows that the downstream guarantees in Theorem 2 and 3 remain valid when  Eq. (13) is satisfied. Together, these results imply (at least for $\chi^2$-contrastive learning) that an encoder $\widehat{f}$ minimizing the empirical $\mathsf{f}$-contrastive loss can achieve small downstream errors, provided the pretraining sample size $n$ is sufficiently large and the augmentation error $\epsilon_{\mathcal{G}}$ is small.
> Nevertheless, we agree the statement may be somewhat overstated, as we focus our analysis on $\chi^2$-contrastive learning, leaving other contrastive losses for future investigation (lines 224–227). We will revise the statement in the paper.
>
> ---
> > Q3. There are many "some constants" throughout the paper. How are they determined, at least in examples? How much are bounds tight?
>
> A3. Throughout the paper, we use $c > 0$ to denote absolute constants independent of any parameters (e.g., $c = 1, 2, 10$). We use $C > 0$ to denote constants that depend polynomially on certain parameters (e.g., $B_{\mathsf{S}}$ in Assumption 1), and we explicitly state the dependencies when introducing the constants. However, tracking the exact polynomial dependence (e.g., $C = \mathcal{O}(B_{\mathsf{S}}^2)$) can be cumbersome in analysis, so we use $C$ to simplify the presentation and improve clarity without losing too much in the bounds. Please let us know if this addresses your question.
>
> ---
>
> > Q4. The error on the downstream task induced by data augmentation epsilon_G is an important variable in their analysis but it is not specified well. For example, it would be useful to know how we can measure epsilon_G in practice, how much its value deviates by taking different augmentation strategies, and so on.
>
> A4. We thank the reviewer for this practical question. In general, it is challenging to measure $\epsilon\_{\mathcal{G}}$ (or $\epsilon^{\mathsf{cls}}\_{\mathcal{G}}$) for real-world augmentations, as their values depend on concrete assumptions about the augmentations and oracle knowledge of the data. For example, in image classification tasks, the augmentation error $\epsilon^{\mathsf{cls}}\_{\mathcal{G}}$ of random cropping depends on the proportion of pixels dropped and the true label probabilities of the cropped images—which are typically unknown. Nevertheless, in practice, one may use the predicted label probabilities of the original and cropped images from a highly accurate image classifier as a surrogate for the true label probabilities, and use them to empirically estimate the augmentation error $\epsilon^{\mathsf{cls}}_{\mathcal{G}}$.
>
>
>
> ---
>
> > Q5 and 6: formatting issue and typo.
>
> A5. Thanks for the suggestions. We will fix them in the paper.

---

> ### Comment · Reviewer_sYsE · 2025-08-05
>
> Thank you for your response. While I still believe that the experiment presented during the rebuttal period is somewhat limited, and that extending the experiment to a more realistic setting, e.g., using CLIP, would further strengthen the contribution, I do not consider this is a critical reason to reject the paper. I have no significant concerns at this point.

---

> > ### Author Response · Authors · 2025-08-08
> >
> > Thanks for the comments and suggestions. Over the past week, we have conducted small-scale experiments in the CLIP setting (language-image pretraining). Namely, we trained the CLIP model (RN50-quickgelu, which consists of a ResNet-50 image encoder and 12-layer Transformer text encoder) on a 100K subsample of the cc3m-wds dataset using both InfoNCE and $\chi^2$-contrastive losses. The original dataset contains about 3.3M image-text pairs, but due to limited compute, we trained on the 100K subsample for 32 epochs.
> >
> > We evaluated the models based on their zero-shot classification performance on the ImageNet-1k validation set (1000 classes, 500 images per class). For InfoNCE and $\chi^2$-contrastive loss, we set the link functions $\tau(x)$ to be $x/t$ and $e^{x/t}$, respectively, with the trainable temperature $t$ initialized to 1. We chose the batch size $K=128$ and used the AdamW optimizer with weight decay 0.02. The learning rate was selected via grid search over  $\\{3\mathrm{e}{-5}, 1\mathrm{e}{-4}, 3\mathrm{e}{-4}, 1\mathrm{e}{-3}\\}$. For both losses, the selected optimal learning rate was $3\mathrm{e}{-4}$.
> >
> > We repeated the experiments three times and report the top-5 accuracy (%) on the ImageNet-1k validation set:
> >
> > > | InfoNCE         | Chi-squared     |
> > > |--------------------|--------------------|
> > > | 7.53 ± 0.25   | 9.43 ± 0.11  |
> >
> > We can see that in this small-scale experiment, the model trained with $\chi^2$-contrastive loss achieves comparable zero-shot performance to that trained with InfoNCE. We do not claim that the $\chi^2$-contrastive loss is better than InfoNCE, as both methods could benefit from further hyperparameter tuning (e.g., initial temperature) or larger datasets. However, we do believe that $\chi^2$-contrastive loss can learn useful representations of the data, aligning with our theoretical findings. We hope this addresses your question regarding the practical applicability of our method.

---

### Official Review · Reviewer_8N6b · 2025-07-03

**Clarity:** 3
**Significance:** 3
**Originality:** 2
**Rating:** 4
**Confidence:** 3

**Summary:**

This paper provides a statistical theory of contrastive learning via the approximate sufficient statistics, following~\cite{oko2025statistical} for contrastive language-image pretraining (CLIP) using KL-divergence. Specifically, the authors generalize the concept of the approximate sufficient statistics to equivalent forms and general $f$-divergence, and illustrate that minimizing SimCLR and other contrastive losses yields encoders that are approximately sufficient. The key factors influencing these near-sufficient encoders are their sufficiency and the error induced by data augmentation in contrastive learning. Examples, including linear regression and topic classification, are given to illustrate the applicability of results.

**Questions:**

1. Do the provided bounds in Eq.(4) in Theorem 1, Eq.(7a) and Eq.(7b) in Theorem 2, and Eq.(8) in Theorem 3 tight? Please give more discussions.
2. Please give experimental results to support the validity of the theoretical results in Theorem 6.

**Ethical Concerns:**

["NO or VERY MINOR ethics concerns only"]

**Final Justification:**

The authors have addressed most of my concerns, including the tightness of the bounds, additional challenges compared with~\cite{oko2025statistical}, and experimental results to support the validity of the theoretical results in Theorem 6. The remaining concern is the reason why using KL divergence instead of common metrics (e.g., 0-1 loss) to evaluate the classification performance.
Thus, I keep the score.

**Limitations:**

Please see the weaknesses section.

**Quality:**

3

**Strengths And Weaknesses:**

# Strengths
1. Overall, this paper is well-written. The related work and comparison with this work are discussed, especially the work~\cite{oko2025statistical}.
2. The claims about the contributions are supported. Specifically, (1) it generalizes the concept of the approximate sufficient statistics in~\cite{oko2025statistical} for contrastive language-image pretraining (CLIP) using KL-divergence; (2) it provides the theoretical analysis of data augmentation-based contrastive learning following the SimCLR framework, demonstrating the downstream performance of the learned encoder depends on its sufficiency and the error induced by the random transformation.
3. The proofs seem right, although I have not checked the proof details line by line.

# Weaknesses
1. Technically, the novelty seems a little limited due to its similarity to the work~\cite{oko2025statistical}. More details about the additional challenges are suggested to be provided.
2. The tightness of the bounds in Eq.(4) in Theorem 1, Eq.(7a) and Eq.(7b) in Theorem 2, and Eq.(8) in Theorem 3,  are not discussed.
Besides, for the classification problem of Theorem 3, why use KL divergence instead of common metrics (e.g., 0-1 loss) to evaluate the classification performance?
2. There are no experimental results to support the theoretical results. Are the provided bounds non-vacuous or meaningful? Synthetic experiments on examples of linear regression or classification can be conducted to illustrate the validity of theoretical results.

---

> ### Author Rebuttal · Authors · 2025-07-28
>
> We appreciate the helpful comments and constructive feedback from Reviewer 8N6b. Below, we respond to the questions and comments in a point-by-point manner.
>
> ---
> > Q1. Do the provided bounds in Eq.(4) in Theorem 1, Eq.(7a) and Eq.(7b) in Theorem 2, and Eq.(8) in Theorem 3 tight? Please give more discussions.
>
>
> A1. For Eq. (4), we expect that the polynomial dependence of the constant $C$ on $B_{\mathsf{S}}$ is necessary to obtain a non-vacuous bound for arbitrary batch size $K > 2$. However, we suspect that the constant $C$ in Eq. (5a) may be improved to depend polynomially on $\log(B_{\mathsf{S}})$. We hope to derive a more precise characterization of the (log-)polynomial dependence in Eq (4) and (5a) in future work.
>
> For Eq. (7a), as discussed after Theorem 2 (line 195-197), the error $\epsilon_{\mathcal{G}}$ on the right-hand side can be replaced by the minimum error $\widetilde\epsilon_{\mathcal{G}}$, and is therefore tight (up to constant factors) when the sufficiency is sufficiently small.
>
> For Eq. (7b) and (8b), the current results are not tight due to the gap between between $\epsilon_{\mathcal{G}}$ and the minimum error $\widetilde\epsilon_{\mathcal{G}}$. We leave the derivation of matching lower bounds to future work.
>
> ---
>
> > W1: Technically, the novelty seems a little limited due to its similarity to the work~\cite{oko2025statistical}. More details about the additional challenges are suggested to be provided.
>
> A2. A main technical challenge of extending Oko et al. (2025) [1] to the SimCLR setting (or single-modal contrastive learning) is handling the error induced by random augmentation. Note that random augmentation is not performed in CLIP considered in [1], and therefore the downstream error can be controlled directly via the sufficiency of the encoder (see, e.g., Propositions 2, 3, and 4 in [1]). In our work, we introduce a novel augmentation error term $\epsilon_{\mathcal{G}}$ (or $\epsilon^{\mathsf{cls}}_{\mathcal{G}}$), and use it to establish downstream error bounds for any encoder $f$, by applying triangle inequalities for $\ell_2$ error in regression (Theorem 2) and for KL divergence in classification (Theorem 3).
>
>
> ---
>
>
> > W2. Besides, for the classification problem of Theorem 3, why use KL divergence instead of common metrics (e.g., 0-1 loss) to evaluate the classification performance?
>
> A3. We use KL divergence instead of 0-1 loss in the multi-class classification setting (Theorem 3) because it is a standard and more informative metric for quantifying the difference between the predicted and true label distributions.
>
> ---
>
>
> > Q2 and W3. Please give experimental results to support the validity of the theoretical results in Theorem 6.
>
> A4. We conduct synthetic experiments to learn data representations via contrastive learning with a two-layer neural network, and evaluate them on downstream linear regression.
>
>
> In the contrastive learning phase, we generate $n$ i.i.d. samples $x_i \sim \mathcal{N}(0, I_d)$. The augmentation $g$ adds i.i.d. $\mathcal{N}(0, \sigma_1^2)$ noise to the first $s < d$ coordinates of $x_i$, and replaces the remaining coordinates with i.i.d. $\mathcal{N}(0, 1)$ noise. We apply KL and $\chi^2$-contrastive learning (Eq 3 and 10) with link function $\tau(x) = x$, and encoder $f(x)$ being a two-layer ReLU neural network mapping $\mathbb{R}^d$ to $\mathbb{R}^s$. We set $s = 10,d = 100,n=500$, hidden dimension 64, and batch size $K = 64$. The encoder is trained using Adam (learning rate 0.001) for 300 epochs, after which the training loss is observed to converge.
>
>
> For downstream regression, we generate $m$ i.i.d. samples $(x_i, y_i)\_{i=1}^m$, where $x_i \sim \mathcal{N}(0, I_d)$ and $y_i = \langle x_i, \theta_\star \rangle + \epsilon_i$, with $\epsilon_i \sim \mathcal{N}(0, \sigma^2)$ independent of $x_i$. We choose $\theta_\star = (\mathbf{1}^\top_s/\sqrt{s},\mathbf{0}^\top_{d-s})^\top$ and $\sigma=1$. Using the learned representation $\widehat{f}(x_i) \in \mathbb{R}^s$ from KL (or $\chi^2$)-contrastive learning, we fit a downstream linear model to predict $y_i$. We define the excess risk of any predictor $h$ as $\mathbb{E}[(y_i - h(x_i))^2] - \sigma^2$, and evaluate the excess risk of the linear model trained on $(\widehat{f}(x_i),y_i)\_{i=1}^m$ using 50000 test samples. For comparison, we also report the excess risk of a linear model trained directly on the original samples $(x_i,y_i)\_{i=1}^m$. Results for various downstream sample sizes $m$ and the standard deviation over 10 runs are shown below.
>
> > **Table: Excess risk  for various downstream sample sizes m**
> >
> > | m    | InfoNCE         | Chi-squared     | Direct LR       |
> > |------|-----------------|-----------------|-----------------|
> > | 150 | 0.106 ± 0.040 | 0.120 ± 0.028 | 2.066 ± 0.594 |
> > | 500 | 0.060 ± 0.015 | 0.070 ± 0.013 | 0.243 ± 0.032 |
> > | 1000 | 0.052 ± 0.012 | 0.063 ± 0.012 | 0.114 ± 0.021 |
> > | 2000 | 0.046 ± 0.011 | 0.058 ± 0.013 | 0.055 ± 0.012 |
> > | 5000 | 0.042 ± 0.011 | 0.055 ± 0.013 | 0.021 ± 0.005 |
> > | 10000 | 0.040 ± 0.012 | 0.054 ± 0.012 | 0.011 ± 0.004 |
>
>
> From the table, we observe that InfoNCE and Chi-squared achieve comparable excess risks (within one standard deviation except for $m=10000$), and both are substantially lower than that of direct linear regression when the sample size $m$ is relatively small (e.g., $m = 150, 500$). This suggests that both KL and $\chi^2$-contrastive learning can learn a “good” low-dimensional representation for the downstream task. As the sample size increases, the excess risk of direct linear regression converges to zero, while those of InfoNCE and Chi-squared converge to non-zero constants. This is consistent with our theoretical results, which attribute the excess risk to the non-zero sufficiency of $\widehat{f}$ and the augmentation error $\epsilon_{\mathcal{G}}$.
>
> Due to limited computational resources, we did not evaluate the performance of $\chi^2$-contrastive learning on large-scale real-world datasets and leave this for future investigation.
>
> ---
>
> [1]. Oko, K., Lin, L., Cai, Y., & Mei, S. (2025). A statistical theory of contrastive pre-training and multimodal generative ai. arXiv preprint arXiv:2501.04641.

---

> > ### Comment · Reviewer_8N6b · 2025-08-07
> >
> > Thank you for the detailed response. It has addressed most of my concerns, and I will keep the score.

---

### Comment · Area_Chair_sidy · 2025-08-04
**Author-reviewer discussion period ends soon**

Dear reviewers,

Author-reviewer discussion period ends soon. Please check the rebuttals and take an appropriate action.

AC

---

### Decision · Program_Chairs · 2025-09-17

**Decision:**

Accept (poster)

**Comment:**

The paper develops a statistical framework for contrastive learning with data augmentation (e.g., SimCLR), extending prior work on CLIP with KL-divergence to general f-divergences. It formalizes the concept of approximate sufficient statistics, and shows that minimizing contrastive losses yields encoders that are approximately sufficient.

Reviewers commented strengths such as theoretical contribution, mathematical rigor and timely topic. While three reviewers were originally worried of lack of experiments, they came out with positive comments after successful rebuttal. At the end, all have unanimously provided review score of 4. Given the theoretical nature of the contribution, an acceptance is recommended.